# Profiling protein–protein interactions to predict the efficacy of B-cell-lymphoma-2-homology-3 mimetics for acute myeloid leukaemia

Changju Chun [1,6], Ja Min Byun [2,6], Minkwon Cha[1,4,6], Hongwon Lee [3], Byungsan Choi[3], Hyunwoo Kim[3], Saem Hong[3], Yunseo Lee[3], Hayoung Park[3,5], Youngil Koh [2] ✉ & Tae-Young Yoon [1,3] ✉

B-cell-lymphoma-2 (BCL2) homology-3 (BH3) mimetics are inhibitors of protein–protein interactions (PPIs) that saturate anti-apoptotic proteins in the BCL2 family to induce apoptosis in cancer cells. Despite the success of the BH3-mimetic ABT-199 for the treatment of haematological malignancies, only a fraction of patients respond to the drug and most patients eventually develop resistance to it. Here we show that the efficacy of ABT-199 can be predicted by profiling the rewired status of the PPI network of the BCL2 family via single-molecule pull-down and co-immunoprecipitation to quantify more than 20 types of PPI from a total of only $1.2 \times 10^6$ cells per sample. By comparing the obtained multidimensional data with BH3-mimetic efficacies determined ex vivo, we constructed a model for predicting the efficacy of ABT-199 that designates two complexes of the BCL2 protein family as the primary mediators of drug effectiveness and resistance, and applied it to prospectively assist therapeutic decision-making for patients with acute myeloid leukaemia. The characterization of PPI complexes in clinical specimens opens up opportunities for individualized protein-complex-targeting therapies.

Large protein complexes play pivotal roles in orchestrating the transition between critical cellular states while reflecting diverse cellular states[1,2]. Examples include regulation by mammalian target of rapamycin (mTOR) complexes in metabolism[3], inflammasomes in innate immunity[4] and B-cell-lymphoma-2 (BCL2) family complexes in apoptotic cell death[5]. Despite the extensive range of proteins associated with these complexes and their rapid turnover, the connectivity among several core components still defines critical setpoints in the alteration of cellular states. For example, across possible combinations of the BCL2-family proteins, the formation of proteinaceous pores by BCL2-like protein 4 (BAX) and BCL2 homologous antagonist/killer (BAK) leads to mitochondria outer membrane permeabilization making a point of no return from apoptosis initiation[6–8]. The survival and apoptotic pressures accrued in cells are reflected by the expression of anti- and pro-apoptotic proteins, tipping the scale towards survival or apoptosis, respectively. While BCL2, B-cell-lymphoma-extra-large

[1]School of Biological Sciences and Institute for Molecular Biology and Genetics, Seoul National University, Seoul, South Korea. [2]Department of Internal Medicine, Seoul National University College of Medicine, Seoul National University Hospital, Seoul, South Korea. [3]Department of Biomarker Discovery, PROTEINA Co., Ltd, Seoul, South Korea. [4]Present address: Department of Physics, Pohang University of Science and Technology (POSTECH), Pohang, South Korea. [5]Present address: School of Biological Sciences and Institute for Molecular Biology and Genetics, Seoul National University, Seoul, South Korea. [6]These authors contributed equally: Changju Chun, Ja Min Byun, Minkwon Cha. ✉e-mail: go01@snu.ac.kr; tyyoon@snu.ac.kr

(BCLxL) and induced myeloid leukaemia cell differentiation protein (MCL1) make up the group of anti-apoptotic proteins that sequester BAX/BAK to prevent pore formation[5,9–11], pro-apoptotic proteins (such as BCL2-like protein 11 (BIM), BCL2-associated agonist of cell death (BAD) and BH3-interacting-domain death agonist (BID)) share an alpha-helical BCL2 homology-3 (BH3) domain and interact with either anti-apoptotic proteins or BAX/BAK[12–14], thereby triggering BAX/BAK-mediated pore formation[15,16].

In diseased states, the dysregulation of cellular function is frequently attributed to chronic or spurious formation of these protein complexes, as seen with mTOR complexes found in various cancers or type 2 diabetes[17,18] and inflammasomes in irritable bowel syndrome or neurodegenerative diseases[19,20]. Likewise, many cancers exhibit upregulated protein–protein interactions (PPIs) of the anti-apoptotic proteins driven by genomic and epigenomic alterations to suppress the activity of BAX/BAK, thus averting apoptosis initiation[10,16,21–23]. Consequently, extensive efforts have been made to develop PPI inhibitors that can disintegrate disease-causing protein complexes[24]. In particular, BH3 mimetics, small-molecule drugs that structurally emulate the BH3 domain, are a notable example in these endeavours[25,26]. While sparing pro-apoptotic proteins, BH3 mimetics are expected to bind selectively to the binding grooves of the anti-apoptotic proteins to shut down their interactions with pro-apoptotic proteins and BAX/BAK[27]. By introducing an azaindole moiety to ABT-263 (Navitoclax) that binds to both BCL2 and BCLxL[28], ABT-199 (Venetoclax) has been developed to specifically bind to BCL2 with a sub-nM affinity[29]. ABT-199 was approved for the treatment of chronic lymphoid leukaemia (CLL) in 2016 (NCT01889186)[30] and acute myeloid leukaemia (AML) in 2018 (NCT02287233, NCT02203773)[31,32], which stimulated the development of other BH3 mimetics that target other anti-apoptotic proteins beyond BCL2 (refs. 27,33).

Consistent with other targeted therapies such as BH3 mimetics, either as monotherapies or in combination, a precision-medicine approach could support the personalized use of PPI inhibitors, utilizing the distinctive molecular data of individual patients[34]. Despite ABT-199's favourable therapeutic outcomes in treating CLL[35,36], over 30% of patients with AML still fail to respond[32,37,38] and many with initial responses eventually develop resistance[39,40]. These facts underscore unmet needs where diverse treatment options could benefit patients if initial responses or resistance development could be accurately predicted. However, multiple factors, including post-translational modifications, protein conformational changes and cellular localizations, may hinder the direct translation of genomic and proteomic information into accurate PPI strength predictions[5,41,42]. Genomic and proteomic profiling have thus found limited use in this technical niche[43–45], and correlations between total expression levels of BCL2 and other anti-apoptotic proteins, and response to BH3 mimetics have not been robust enough for clinical application[46–49]. While the BH3 profiling method offers usefulness in predicting BH3 mimetics, it is not routinely used in clinical settings, probably due to the need for sample viability maintenance[50–52].

In this work, we hypothesized that the responsiveness of individual cancers to a PPI inhibitor would be determined by the compositions of target protein complexes, which continuously change alongside the clonal evolution of the cancers. Specifically, in the case of BH3 mimetics, the complexes between anti-apoptotic and pro-apoptotic proteins and their intricate balances may primarily mediate the efficacy of BH3 mimetics in given cancers. We propose that a predictive algorithm may be trained by extensively profiling BCL2-family protein complexes in as many cancers with known responses as possible. To enable such profiling with minimal sample consumption and increased throughput, we adopted the single-molecule pull-down and co-immunoprecipitation (SMPC) technique, directly immobilizing target protein complexes on the imaging plane of a single-molecule fluorescence microscope and counting the number of complexes[53–56]. This approach allowed us to

determine the populations of 20 different types of protein complex (and protein level) using only ~30,000 cells per complex type, revealing the latest rewiring status of the BCL2-family network in individual clinical samples.

We found that the introduction of bacterial toxin and BH3 mimetics caused significant changes in the composition of BCL2 protein complexes, with only minor alterations in corresponding protein levels. This finding underscores the importance of directly profiling protein complexes rather than merely analysing protein levels. By correlating these multidimensional data with the ex vivo determined efficacies of BH3 mimetics, we were able to train analytical models that identified the protein and PPI parameters determining the efficacy of BH3 mimetics. For ABT-199, we found that the counts of two specific complexes, BCL2-BAX and BCLxL-BAK, served as critical contrasting analysis parameters, with the former driving ABT-199's efficacy and the latter contributing to resistance development. Following a similar procedure, we constructed an efficacy model for an MCL1-targeting inhibitor. Subsequently, we applied the developed analytical model to guide therapeutic decisions for patients with AML in a prospective fashion, confirming the model's predictive power. Our work showcases the utility of the SMPC technique for extensive and accurate profiling of protein complexes and presents a path towards precision medicine in therapies targeting protein complexes.

## Results

### Profiling BCL2-family protein complexes using SMPC

To assess BCL2-family protein complexes and gain insights into how apoptosis is suppressed in individual cancers, we utilized the SMPC technique to characterize the PPI network among BCL2-family proteins[53,54,56,57]. We initiated surface immunoprecipitation (IP) of one of the primary anti-apoptotic proteins (that is, BCL2, BCLxL or MCL1) from crude cell extracts (Fig. 1a and Extended Data Fig. 1a–c), followed by the addition of a monoclonal antibody that binds to the interaction partner complexed with the surface-immobilized bait. A fluorescently labelled secondary antibody was then used, completing an immunoassay to detect specific protein complexes (CPXs) (Fig. 1b, left). In addition, we employed a detection antibody that binds to an epitope directly on the surface bait to determine the total amount of surface-immobilized baits (protein total level, LV) (Fig. 1b, right and Extended Data Fig. 1d–f).

To enhance the sensitivity and reproducibility in this protein complex detection method, we optimized the lysis protocol[57,58]. By using HEK293T cells co-expressing BCL2 and the BH3 domain of BIM (BIM$_{BH3}$), we found that mild lysis with cholesterol-like detergents, such as glycol-diosgenin (GDN) and digitonin, minimized dissociation of the model BCL2-BIM complexes (Extended Data Fig. 1g). Furthermore, this lysis protocol did not induce the formation of spurious complexes even in dense cell extracts (that is, at a total protein concentration of 5–10 mg ml$^{-1}$), confirming that all detected protein complexes were formed in cellular environments (Fig. 1c). Conformational changes of the BCL2-family proteins during their activation are crucial in their function[16,59]. For instance, the pore formers, BAX and BAK, expose their α1 helices in the activated state, which are otherwise buried in the core structure. By assessing the active status of BAX using an anti-α1 helix antibody, we found that the detergent condition substantially affected BAX's conformational state[2,60]. Triton X-100, often used in conventional co-IP methods, induced spurious conformational changes[61], while GDN essentially preserved BAX's active conformations induced by the addition of a bacterial toxin, staurosporine[62] (Fig. 1d). Similarly, we experimentally confirmed the ability to assess the activation status of BAK[63,64], by employing an antibody that specifically binds to the α1 helix of BAK (Extended Data Fig. 1h,i). Finally, we experimentally confirmed that a single freeze–thaw cycle did not seriously affect the single-molecule counts of BCL2 protein complexes (Extended Data Fig. 1j–l).

To minimize non-specific adsorption to the surface, we placed a high-density polyethylene glycol (PEG) layer between the IP antibodies

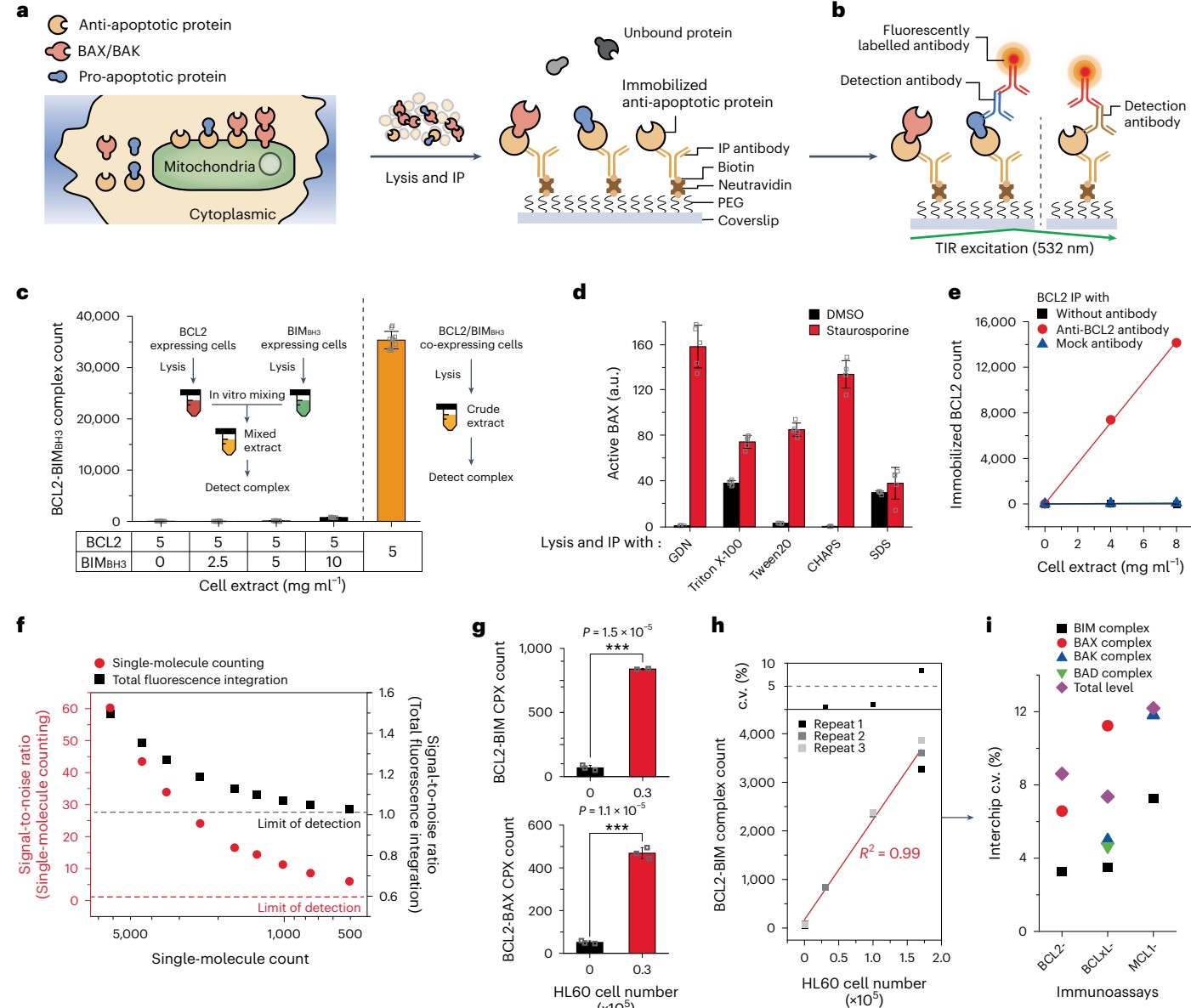

**Fig. 1 | Profiling BCL2-family protein complexes using the SMPC technique.**
**a**, Schematic for the surface IP of BCL2-family proteins. Anti-apoptotic proteins in cell extracts were immobilized onto the surface of the reaction chamber. **b**, Schematic of immunoassay selectively detecting protein complexes (left) and the total level of surface bait protein (right). The fluorescence signals from detection antibodies were imaged under total internal reflection (TIR) excitation. **c**, Comparison of the formation of model BCL2-BIM$_{BH3}$ complex in an intracellular environment by BCL2-mCherry/BIM$_{BH3}$-eGFP co-expression (right) and in vitro mixing of two individual extracts with each expressing BCL2-mCherry and BIM$_{BH3}$-eGFP (left) ($n = 10$ independent images). **d**, Changes in active BAX level measured for staurosporine-treated HL60 cells after lysis and IP with different

detector types ($n = 5$ independent images). **e**, Single-molecule count of surface-immobilized BCL2-eGFP from the crude extract with indicated IP antibodies ($n = 10$ independent images). **f**, Comparison of signal-to-noise ratio between single-molecule counting analysis and integration of total fluorescence signals. **g**, Counts of endogenous BCL2 protein complexes from the 30,000 HL60 cells by using immunoassay ($n = 3$ biological replicates) (two-sided two-sample $t$-test, ***$P = 1.5 \times 10^{-5}$, $1.1 \times 10^{-5}$). **h**, Counts of endogenous BCL2-BIM complexes and interchip c.v.s from the fixed numbers of HL60 cells ($n = 3$ biological replicates). **i**, Interchip c.v.s obtained from independent interchip measurement for all immunoassays and cell numbers ($n = 3$ biological replicates). Data represent means ± s.d.

and the glass coverslip[57,58] (Fig. 1a). This PEG layer proved essential in enabling direct surface IP of the target protein complexes from the crude extracts, reducing background signals to a minimal level on our single-molecule fluorescence imaging (Fig. 1e). By effectively removing background signals through signal processing, we could identify single-molecule fluorescence spots and directly count their numbers (Extended Data Fig. 2a–c)[55,65]. Compared with conventional approaches that integrate total fluorescence signals over a field of view, this single-molecule counting method maintained signal-to-noise ratios higher than 5 when there were ~1,000 or fewer

individual single-molecule spots per imaging area (Fig. 1f and Extended Data Fig. 2).

Through the combination of the three technological factors of optimized lysis protocol, anti-fouling PEG layer and direct single-molecule counting, we achieved statistically significant counts for endogenous BCL2-BIM or BCL2-BAX CPXs using only ~30,000 AML cells (Fig. 1g,h). We also demonstrated a similar capability using a non-small-cell lung cancer cell line (PC9 cells), where we detected 8 different BCLxL- and MCL1 protein complexes using less than 50,000 PC9 cells each (Fig. 1i and Extended Data Fig. 3). Three independent

interchip measurements yielded coefficients of variation (c.v.s) of less than 15% for all assays and cell numbers tested, confirming the high stability of the established SMPC technique (Fig. 1h,i and Extended Data Fig. 3).

### Monitoring responses to apoptotic stress using SMPC

To investigate whether the developed SMPC technique could monitor alterations within the BCL2-family PPI network, HL60 cells were treated with 2 μM of staurosporine. The crude extracts were obtained from these cells at various time points during the treatment (Extended Data Fig. 4a). Through flow cytometry analysis, the early apoptosis population, characterized by high Annexin V but low propidium iodide staining, was observed to surge at 1 h of staurosporine treatment, reducing the healthy cell population (Extended Data Fig. 4b). The late apoptosis population, stained with both Annexin V and propidium iodide, increased after 4 h. Necrosis populations, marked by high propidium iodide and low Annexin V staining, remained negative throughout the study, signifying that the cells underwent apoptosis rather than necrosis (Extended Data Fig. 4a,b).

We observed that the counts of active BAX and BAK dramatically increased as apoptosis progressed (Fig. 2a). The count of BAX-BAK complexes, a more direct surrogate marker of mitochondria outer membrane permeabilization[66,67], began to rise simultaneously with the active BAX and BAK counts but continued to escalate into the late apoptosis stage. We then quantified the total levels of anti-apoptotic proteins (Fig. 2b). Through careful calibration of immunolabeling efficiencies (Extended Data Fig. 5), we were able to directly compare the quantities of the primary anti-apoptotic proteins and their protein complexes (Fig. 2b–d). We found that BCL2 proteins were more abundant by 4 to 5 times compared with BCLxL and MCL1, aligning with the known dependence of HL60 on BCL2 (Fig. 2b). As apoptosis progressed, BCLxL and MCL1 levels gradually diminished, while BCL2 protein levels exhibited a slight yet significant increase.

Contrary to the minimal change in BCL2 protein level, the complexes formed between BCL2 and pro-apoptotic proteins displayed dynamic changes throughout the apoptosis stages. We discovered that BCL2 protein complexes outnumbered those involving BCLxL and MCL1, further confirming HL60's BCL2 dependence (Fig. 2c,d). Notably, the two protein complexes exhibited contrasting behaviours. While the counts of the BCL2-BIM complex gradually decreased over time, the BCL2-BAX complex counts surged for the first 2 h following staurosporine treatment (Fig. 2c,d). Collectively, our comprehensive profiling of the BCL2-family protein complexes revealed that BCL2 proteins mainly functioned to sequester the active BAX proteins during staurosporine-induced apoptotic pressure in HL60 while simultaneously releasing some previously held BIM proteins. Presumably, under the applied staurosporine treatment conditions, the active BAX outnumbered BCL2's sequestration capacity, leading to the HL60 cells succumbing to apoptotic stress and initiating mitochondria outer membrane permeabilization (Fig. 2a,d).

### Assessment of potential PPIs using protein interactor probes

Our findings reinforce the understanding that BCL2-family proteins, akin to many cell signalling proteins, function by binding to (or sequestering) interaction partners[4]. During elevated survival pressure in HL60 cells induced by staurosporine treatment, the BCL2 proteins consistently formed additional BCL2-BAX complexes to counteract apoptotic pressure (Fig. 2d). This led us to conceive additional assays to evaluate the capacity of BCL2-family proteins to form further protein complexes beyond current complex populations.

To evaluate the affinities of potential PPIs, we employed the same SMPC approach to immobilize anti-apoptotic proteins, and various pro-apoptotic proteins tagged with enhanced green fluorescent proteins (eGFPs) were then introduced into the reaction chamber (we call this modality as probe binding assay, PBA) (Fig. 2e).

These PPI probes were designed to engage with a pool of unoccupied anti-apoptotic proteins, potentially facilitating the formation of new protein complexes (Fig. 2e). Indeed, the PBA effectively replicated the dissociation constants for the binding interactions between anti- and pro-apoptotic proteins, and faithfully reproduced known biochemical features including a higher affinity of BCL2 for BAD compared with BIM and negligible interactions for BCL2-NOXA and MCL1-BAD pairs (Extended Data Fig. 6a–f and Supplementary Table 1)[4]. We additionally confirmed that addition of these PPI probes did not cause disruption of the pre-existing BCL2-BIM complexes. This indicates that the PPI probe selectively binds to unoccupied anti-apoptotic proteins without causing the disintegration of existing endogenous complexes (Extended Data Fig. 6g).

Using PBA, we analysed BCL2 proteins extracted from the staurosporine-treated HL60 cells (Extended Data Fig. 4b). By employing the BH3 domain of BIM (BIM$_{BH3}$-eGFP) as a PPI probe, BCL2 exhibited a significantly higher PBA count in comparison with BCLxL and MCL1 (Fig. 2f). Given the higher affinities of the BIM$_{BH3}$ domain for BCLxL and MCL1 than for BCL2 (Extended Data Fig. 6a,e), this elevated PBA count suggests that BCL2 works as a larger reservoir in sequestering pro-apoptotic proteins compared with BCLxL or MCL1 in HL60 (Fig. 2f). Notably, as apoptosis progressed, the PBA count of BIM$_{BH3}$ declined, despite the rise in total BCL2 levels, possibly indicating the exhaustion of available BCL2 proteins through the formation of BCL2-BAX complexes (Fig. 2b–f). Furthermore, we inspected the composition of the BCL2 protein complexes in four AML cell lines (Fig. 2g). When immobilizing an equivalent number of total BCL2 proteins per view (Fig. 2g, right), NB4 and U937 cells were found to contain substantially more BCL2-BIM and BCL2-BAX complexes, the two main species of BCL2 protein complex species, compared with HL60 (Fig. 2h). This observation corresponded with the BIM$_{BH3}$ PBA count being more than halved for NB4 and U937 relative to HL60 (Fig. 2i), reflecting increased BCL2 occupancy and thereby reduced pools of BCL2 proteins available for potential PPIs in these cells (Fig. 2j). It was noteworthy that the combined totals of the PBA and protein complex counts accounted for 60 to 80% of the total BCL2 proteins in the cell lines we investigated. This suggests that a substantial proportion of BCL2 proteins may be complexed with interaction partners not accounted for in our study (Fig. 2j).

Finally, we investigated whether different PPI probes target the same pool of BCL2 proteins or distinct, separate BCL2 pools (that is, each pool accessible only by specific PPI probes). To answer this question, we introduced multiple PPI probes simultaneously and found that these probes directly competed with one another, as well as with ABT-199 (Fig. 2k and Extended Data Fig. 6h,i). In addition, the consistency of the PBA counts across different cell lines and PPI probes indicated the existence of a singular, common BCL2 protein pool, rather than multiple, fragmented pools of BCL2 proteins (compare Fig. 2i with Extended Data Fig. 6j–l).

### Tracing changes in the BCL2-family protein complexes provoked by BH3 mimetics

Leveraging the two assays we established (namely, single-molecule immunoassay and PBA), we set out to explore how BH3 mimetics instigated apoptotic pressure in AMLs. We exposed HL60 cells to 100 nM ABT-199 for 24 h, resulting in a modest ~30% increase in the total BCL2 level, while BCLxL and MCL1 levels remained minimal (Fig. 3a). Conversely, the amounts of active BAX and BAK, along with BAX-BAK complexes, surged 5-fold after ABT-199 treatment, signalling active apoptosis induction in HL60 (Fig. 3b). Although total BAX and BAK populations were largely consistent (Fig. 3c), the majority of the BCL2-BIM and BCL2-BAX complexes dissociated due to ABT-199 exposure (Fig. 3d). These findings highlight that BAX, once released from BCL2, mainly contributed to the steep increases in active BAX and BAK populations. In addition, ABT-199's liberation of BCL2 proteins led to

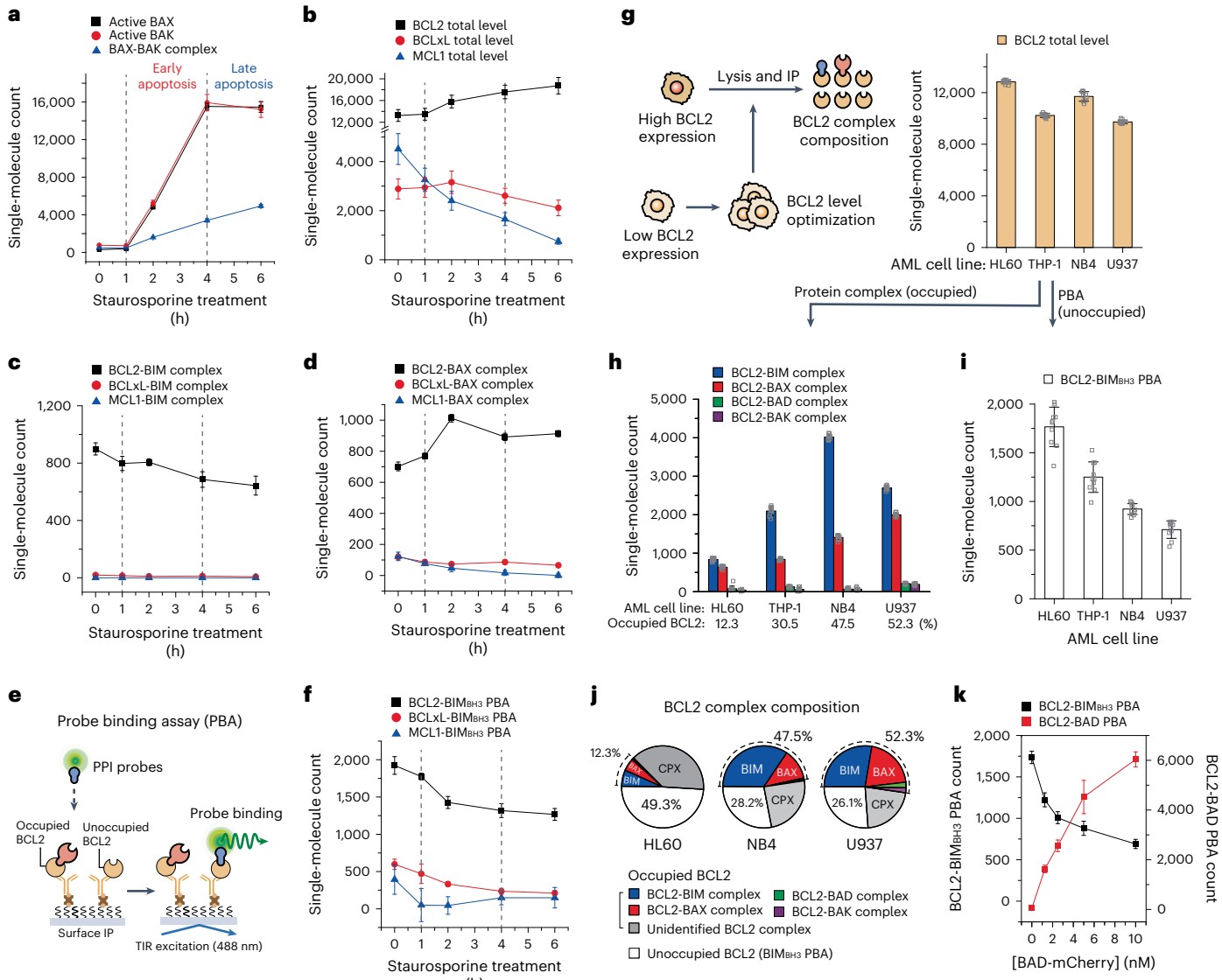

**Fig. 2 | Tracking the PPI profile changes to intrinsic apoptotic stress and assessment of potential PPIs with PBA. a–d**, Changes in endogenous BCL2-family PPI profiles of HL60 cells through apoptosis progression by 2 μM of staurosporine. Active BAX/BAK level (**a**), total levels of anti-apoptotic proteins (**b**), BIM complexes (**c**), BAX complexes ($n$ = 10 independent images) (**d**). **e**, Schematic of the PBA to measure the unoccupied populations of surface-immobilized anti-apoptotic proteins. **f**, Changes in $BIM_{BH3}$ PBAs for anti-apoptotic proteins from HL60 cells through apoptosis progression by staurosporine ($n$ = 10 independent images). **g**, Schematic of the comparison of the BCL2 protein complex compositions in different AML cell lines (left).

The density of surface-immobilized BCL2 was constantly maintained by optimizing the total protein concentration of crude cell extracts in all AML cell lines (right) ($n$ = 10 independent images). **h–j**, Compositions of the BCL2 PPI profiles in AML cell lines. BCL2 protein complexes (**h**), $BCL2-BIM_{BH3}$ PBA ($n$ = 10 independent images) (**i**), integrated BCL2 complex compositions (**j**). **k**, Changes in $BCL2-BIM_{BH3}$ PBA and BCL2-BAD PBA with increasing amounts of BAD probe ($n$ = 10 independent images). $BIM_{BH3}$ probe was presented at 10 nM. The single-molecule counts were rescaled to account for the labelling efficiencies of the immunoassays calculated in Extended Data Fig. 5. Data represent means ± s.d.

an elevated PBA count—a direct contrast to the staurosporine-treated case where the PBA count gradually diminished over the treatment period (Fig. 3e).

To delineate the dynamics of BCL2-BIM and BCL2-BAX complexes, especially at early ABT-199 treatment stages, we administered a higher 300 nM concentration of ABT-199 to HL60 cells and conducted SMPC and PBA profiling at each hourly interval following initial treatment (Extended Data Fig. 4c). The two major BCL2 complexes showed opposing results. The BCL2-BAX complex began to unravel instantly after ABT-199 administration, while the number of BCL2-BIM complexes largely persisted (Fig. 3f). The kinetics of BCL2-BAX dissociation synchronized with the activation of BAX/BAK conformations and BAX-BAK complexes (Fig. 3g). These data suggest that the dissociation of BCL2-BAX

complexes, without necessitating BCL2-BIM dissociation or additional BAX expression, probably sufficed to invoke pro-apoptotic pressure and initiate apoptosis (Fig. 3f–i)[43,68–70].

We further investigated whether similar complex dynamics were observable in MCL1 inhibitor-responsive cells. In reaction to AZD-5591 (ref. 71), U937 cells exhibited augmented populations of active BAX and BAK proteins, along with BAX-BAK complexes, consistent with U937's susceptibility to the MCL1 inhibitor (Fig. 3j). Mirroring the case with ABT-199-treated HL60 cells, AZD-5591 treatment spurred a conspicuous dissociation of MCL1 protein complexes (MCL1-BIM and MCL1-BAK), evidenced by the pronounced increase in $BIM_{BH3}$ probe binding for MCL1 (Fig. 3k,l). However, unlike the HL60 case where other anti-apoptotic proteins demonstrated minimal alterations, BCLxL

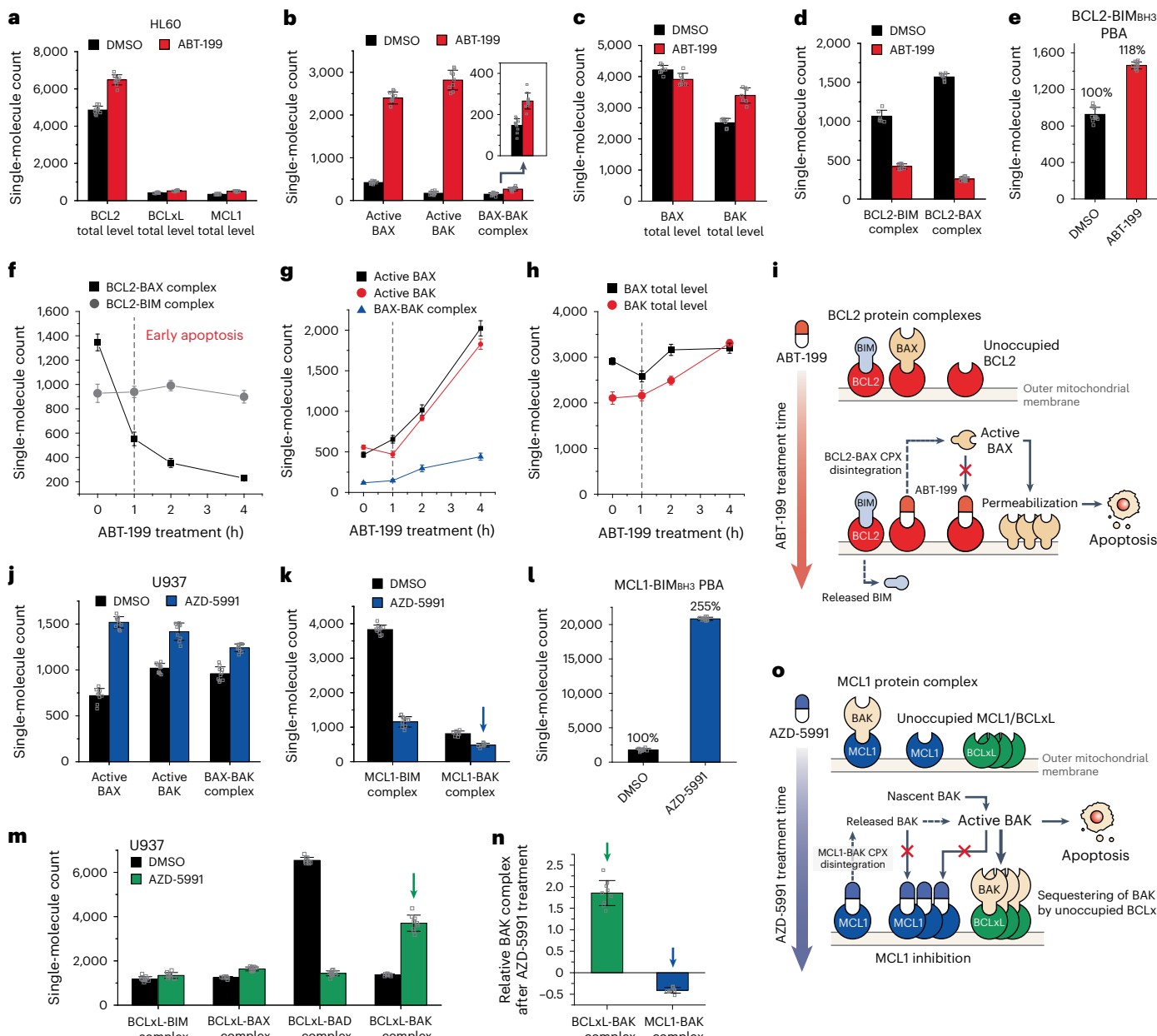

**Fig. 3 | Changes in the BCL2-family protein complexes provoked by BH3 mimetics. a–e,** Changes in BCL2-family PPI profiles in HL60 cells after ABT-199 treatment (100 nM, 24 h). Total levels of anti-apoptotic proteins (**a**), active BAX/BAK level (**b**), total levels of BAX/BAK (**c**), BCL2 complexes (**d**), BCL2-BIM$_{BH3}$ PBA ($n$ = 10 independent images) (**e**). **f–h,** Changes in BCL2-family PPI profile in HL60 cells after higher concentration of ABT-199 treatment (300 nM, 4 h). BCL2 complexes (**f**), active BAX/BAK level (**g**), total levels of BAX/BAK ($n$ = 10 independent images) (**h**). **i,** Schematic illustration of the mode of action of ABT-199. **j–n,** Changes in BCL2-family PPI profiles in U937 cells after AZD-5991

treatment (100 nM, 24 h). Active BAX/BAK level (**j**), MCL1 complexes (**k**), MCL1-BIM$_{BH3}$ PBA (**l**), BCLxL complexes ($n$ = 10 independent images) (**m**), comparison of relative changes in BAK complexes after AZD-5991 treatment (**n**). The relative changes were obtained from the data indicated in **k** and **m** normalized to BCLxL- or MCL1-total level. **o,** Schematic illustration of the mode of action of AZD-5991. The single-molecule counts were rescaled to account for the labelling efficiencies of the immunoassays calculated in Extended Data Fig. 5 (**d,k,m**). Data represent means ± s.d.

proteins in U937 cells exhibited significant surges in their complex counts, with BCLxL-BAK complexes becoming preeminent (Fig. 3m). A thorough calibration of labelling efficiencies allowed us to discern that the observed rise in BCLxL-BAK markedly outpaced the decrease in MCL1-BAK complexes (Fig. 3n). We consequently hypothesized that the majority of new BCLxL-BAK complexes originated from the sequestering of nascent BAK proteins, rather than from capture of those liberated from MCL1 (Fig. 3o). In summary, these data substantiate the feasibility of comprehensive profiling of BCL2-family protein complexes, enhancing our understanding of how apoptotic pressure

is both generated and managed by the BCL2-family protein network within individual cancers.

## Constructing an analysis model for ABT-199 efficacy with multidimensional BCL2 SMPC and PBA profiling

So far, our collective data suggest that BH3 mimetics work through either protein complex disintegration, saturation of anti-apoptotic proteins, or both to achieve their effects. In addition, preliminary findings hint at a single-agent response when cancer cells show a specific dependence on a singular anti-apoptotic protein, with minimal

compensation from other anti-apoptotic proteins (Extended Data Fig. 7). We therefore decided to investigate whether multiparameter data could be used to develop an analytical model to identify the dependency on anti-apoptotic proteins for specific cancers and forecast their responsiveness to BH3-mimetic drugs.

For this purpose, we analysed 32 primary AML samples, of which 30 were gathered from bone marrow mononuclear cells (BMMCs) and 2 from peripheral blood mononuclear cells (PBMCs) (Fig. 4a). The clinical diagnoses included de novo AML ($n = 17$), relapsed AML ($n = 4$) and secondary AML ($n = 11$) (Supplementary Data Set 1). We evaluated the ex vivo effectiveness of ABT-199, determining a normalized area under the response curve (AUC) for each specimen (Fig. 4a, top). Concurrently, the BCL2 SMPC and PBA profiling was performed on the same cohort of samples in a blind fashion, allowing the determination of 22 unique protein/PPI metrics for each sample comprising roughly $1.2 \times 10^6$ cells in total (Fig. 4a, bottom and Supplementary Data Set 2).

Upon comparing the protein/PPI assay data with AUC, four BCL2-related metrics: BCL2-BAD PBA, BCL2-BIM$_{BH3}$ PBA, BCL2-BAX CPX and BCL2 total level, exhibited the highest correlations with ex vivo efficacy (Fig. 4b). Although BCLxL- and MCL1-related metrics also showed appreciable correlations, the correlation values were lower, signalling that most AML cases examined relied on BCL2 for survival (Fig. 4b). Interestingly, the BCL2-BIM complex showed a much lower correlation than the BCL2-BAX complex count, reflecting our observation that the BCL2-BAX complexes' dissociation predominantly mediated the pro-apoptotic effects of ABT-199 in HL60 cells (Fig. 4b–d). At this juncture, it was observed that both the total level of BCL2 and the BCL2-BAX CPXs were not detectable in healthy PBMCs, which implies that BCL2-related complexes are enriched in AML cancer blasts (Extended Data Fig. 8a). However, a simple combination of the four BCL2-related metrics only led to limited improvements in correlation while losing statistical significance (Extended Data Fig. 8b and Supplementary Table 2), possibly due to redundant information concerning the BCL2 protein pool's molecular status (Extended Data Fig. 8c and see below).

To explore metric combinations in a broader parametric space, we divided the 32 AML samples into responsive and non-responsive groups using an ex vivo AUC threshold of 0.61, corresponding to an IC$_{50}$ (half maximal inhibitory concentration) of 100 nM ABT-199 (Extended Data Fig. 8d)[72,73]. We then examined ~10,000 different models, each with varying metric combinations and coefficients, selecting 67 models that made accurate predictions for the training set responsiveness, as validated by the remaining test samples (Extended Data Fig. 8e). Remarkably, the final 67 models consistently identified a subgroup of selected metrics rather than displaying diverse metrics randomly (Fig. 4e). Two metrics, BCLxL-BIM$_{BH3}$ PBA and BCLxL-BAK CPXs, exhibited substantially higher negative correlations with ex vivo AUC values compared with single-metric analysis (Fig. 4b versus Fig. 4e). This presumably reflects a compensatory role for BCLxL in countering the accumulated apoptotic pressure in AMLs.

Upon detailed examination of the 67 models, we noticed a pattern where the information from each metric displayed minimal overlap. For example, while combining the BCL2-BAX CPX and BCL2-BIM$_{BH3}$ PBA data properly (which represented occupied and unoccupied portions of BCL2), neither metric could be combined with the BCL2 total level data (which included both occupied and unoccupied pools), probably due to partial overlap in the information they provided (Extended Data Fig. 8b and Supplementary Table 2). Similarly, the BCLxL-BAK CPX consistently exhibited higher absolute values in its correlations with ex vivo AUC than BCLxL-BAX CPX, possibly because of redundancy in the information concerning BAX that was already considered by the inclusion of the BCL2-BAX CPX metric (Fig. 4e). As a result, a combination of BCL2-BIM$_{BH3}$ PBA, BCL2-BAX CPX and BCLxL-BAK CPX (with a negative coefficient) was selected, showing a slight but substantial enhancement in the correlation with AUC values, with

statistical significance ($P < 0.05$) for all metrics examined (Fig. 4f and Supplementary Table 3). In the receiver operating characteristic (ROC) analysis for binary prediction (responsive versus non-responsive), the resulting AUC showed the highest values for the combined model compared with single-metric models (Fig. 4g and Supplementary Table 4). Moreover, the mean squared error (MSE) was determined to be smallest for the combined model, indicating higher accuracy in estimating ABT-199 efficacy, with only three outliers being included (Extended Data Fig. 8f and Supplementary Table 4).

We tested whether the created model could illustrate changes in the BCL2 PPI network as AML underwent clonal cancer evolution. For instance, BC-7064, a primary AML case, initially responded favourably to combined treatment with ABT-199 and azacitidine but relapsed after discontinuing the treatment (Fig. 4h). We performed BCL2 SMPC and PBA profiling on the initial (BC-7064) and relapsed (BC-7064-R) samples using our technique in a retrospective manner, comparing the three diagnostic PPI metrics in the combined model (Fig. 4h). The initial sample exhibited parameters indicative of strong ABT-199 effectiveness (estimated score of 0.78), composed of upregulated BCL2-BIM$_{BH3}$ PBA and BCL2-BAX CPX counts and a low BCLxL-BAK CPX level (Fig. 4i). Conversely, the two BCL2-related metrics were more than halved for BC-7064-R, signalling a reduced driving force for ABT-199 efficacy. Although the resistance factor BCLxL-BAK complex also decreased for BC-7064-R, the weakened BCL2 dependence led to an estimated score of 0.36, explaining the relapse of this specific blood cancer (Fig. 4h,i).

Finally, we sought to make quantitative comparisons between our SMPC and PBA methods and other established techniques for predicting BH3-mimetic effectiveness, such as BH3 profiling[74,75]. BH3 profiling involves the addition of BH3 peptides derived from BIM, BAD and Harakiri (HRK) to individual AML samples after cell permeabilization, with mitochondrial depolarization monitored using JC-1 dye fluorescence (Extended Data Fig. 9a, top). In our study, HL60 cells demonstrated robust depolarization signals in response to the BAD peptide but not to the HRK peptide, a finding consistent with previous research (Extended Data Fig. 9b,c)[52,76]. We continued by examining 14 primary AML samples from the same cohort (due to limited sample availability) and tested various combinations of the depolarization signals elicited by BIM, BAD and HRK peptides at different concentrations (Supplementary Data Set 3). We found that a combination that examined the difference between BAD and HRK signals (both measured at 10 μM peptides) yielded the highest correlation value of 0.65 with ex vivo ABT-199 efficacy (Extended Data Fig. 9d). Importantly, for this smaller cohort consisting of 14 AML samples, the BCL2 SMPC and PBA profiling essentially maintained the performance parameters examined, including correlation, MSE and AUC from the ROC analysis, which were superior to those determined for the BH3 profiling (Extended Data Fig. 9d–g and Supplementary Table 5). Attempts to use the BCL2 level measured by flow cytometry or western blotting as a predictive marker (Extended Data Fig. 9a) only resulted in muted performance across all evaluated parameters, further emphasizing the effectiveness of our profiling technique in estimating ABT-199 efficacy (Extended Data Fig. 9h–k and Supplementary Table 5).

## Construction of an MCL1 inhibitor efficacy model

Next, we investigated the possibility of training an analysis model for the MCL1 inhibitor AZD-5991, employing a similar process to that which we established for ABT-199. For this purpose, we evaluated the ex vivo efficacy of AZD-5991 on the same primary AML cohort (27 out of 32 total samples), comparing the determined AUC values with the BCL2-family metrics outlined in Fig. 4 (Supplementary Data Set 2). Both the total MCL1 level and the MCL1 PBA count displayed correlation with AZD-5991 efficacy, although the correlations were diminished (Fig. 5a–c). Notably, the BIM$_{BH3}$ PBA assay for BCLxL exhibited a negative correlation with AZD-5991 efficacy (Fig. 5d, see caption). Plotting the BCLxL PBA counts for responsive (ex vivo AUC ≥ 0.61) versus non-responsive

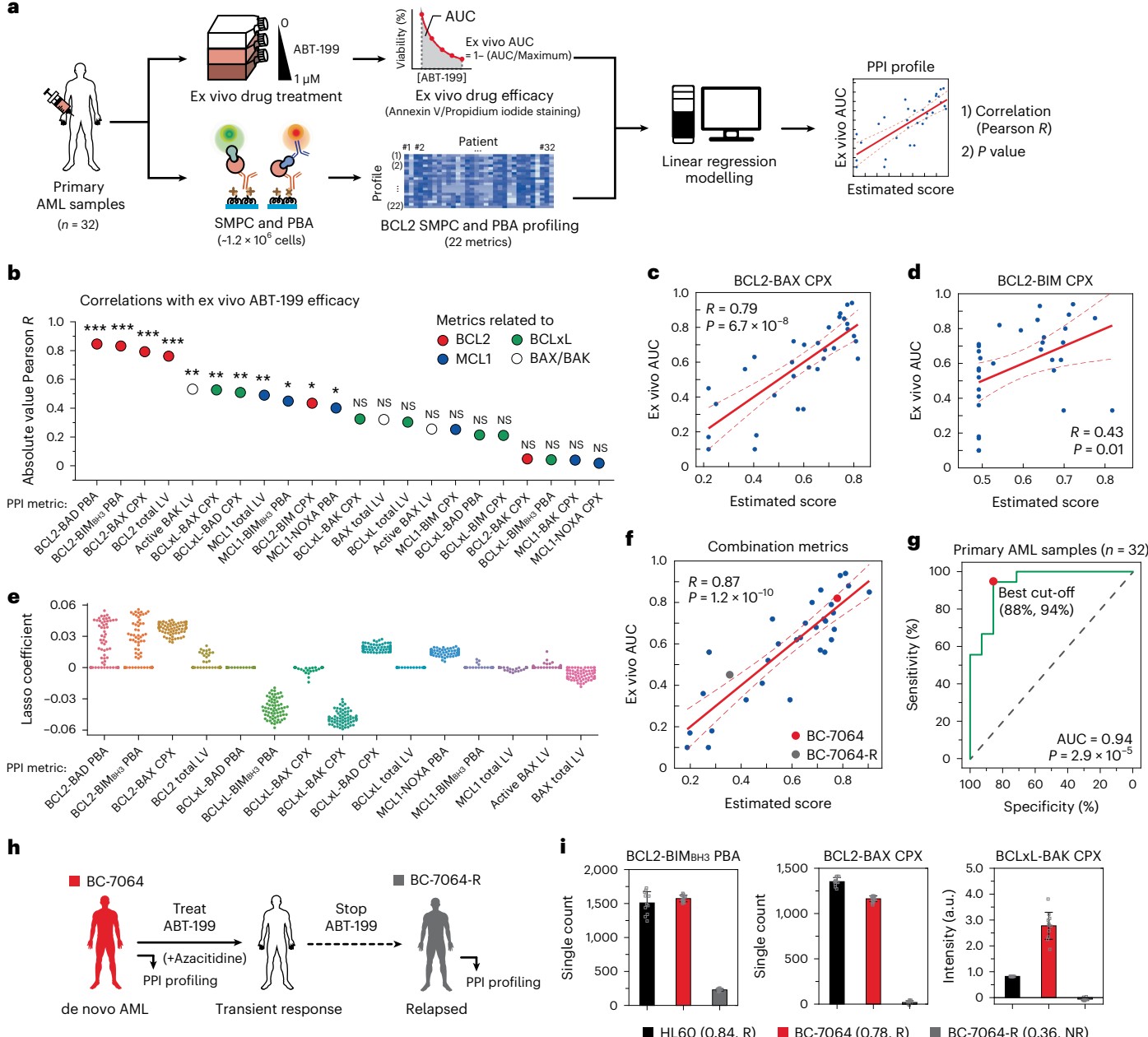

**Fig. 4 | Constructing an analysis model for ABT-199 efficacy with multidimensional BCL2 SMPC and PBA profiling. a**, Schematic for generating linear regression correlation between ex vivo efficacy of ABT-199 and PPI profiles from primary AML samples. The collected AML cells were cultured and treated with 0–1 μM of ABT-199, and the AUCs of cell viability were obtained as ex vivo drug efficacy (top). Approximately $1.2 \times 10^6$ of primary AML cells on the same cohort underwent PPI profiling with the SMPC (bottom). **b**, The absolute Pearson correlations between ex vivo AUC of ABT-199 and PPI metrics for primary AML samples (one-sided $F$-test, *$P < 0.05$, **$P < 0.01$, ***$P < 0.001$, NS, not significant; $P$ values are provided as Source Data) ($n = 32$). **c,d**, Correlations between ex vivo AUC and single BCL2-related PPI metrics. BCL2-BAX CPX (coefficient: 0.157,

$P = 6.7 \times 10^{-8}$) (**c**), BCL2-BIM CPX (coefficient: 0.013, $P = 0.01$) (one-sided $F$-test) (**d**). **e**, Lasso coefficients of PPI profiles correlated with ex vivo AUC of ABT-199 for primary AML samples (67 models). **f**, Correlation between ex vivo AUC and the combination of multiple PPI metrics (BCL2-BIM_BH3 PBA, BCL2-BAX CPX and BCLxL-BAK CPX) (one-sided $F$-test, $P = 1.2 \times 10^{-10}$). **g**, ROC curve between the estimated score and the ex vivo efficacy (two-sided $t$-test, $P = 2.9 \times 10^{-5}$). **h**, Clinical features and the ABT-199 administration history of BC-7064. **i**, Comparison of the PPI profiles and the estimated scores with PPI diagnostic results from the initial and the relapsed BC-7064 samples (R, responsive; NR, non-responsive) ($n = 10$ independent images). Data represent means ± s.d.

samples concerning AZD-5991 revealed that the non-responsive group manifested distinctly higher counts[72,73], indicative of upregulated activities of BCLxL proteins within this group (Fig. 5e and Extended Data Fig. 8g). These collective findings suggest that BCLxL proteins play a substantial compensatory role when MCL1 proteins are saturated with inhibitor molecules. Furthermore, MCL1 protein complexes presented only limited correlations with ex vivo AUC, aligning with our hypothesis

presented in Fig. 3 that AZD-5991 may operate by preventing additional binding of pro-apoptotic proteins to MCL1 (that is, saturation of MCL1 proteins), rather than inducing dissociation of pre-existing MCL1 protein complexes (Fig. 5a).

Integrating multiple parameters, namely, the MCL1 total level, BCLxL-BIM_BH3 PBA and BCLxL-BAK CPX, we formulated a linear regression model that yielded an optimized correlation with ex vivo AZD-5991

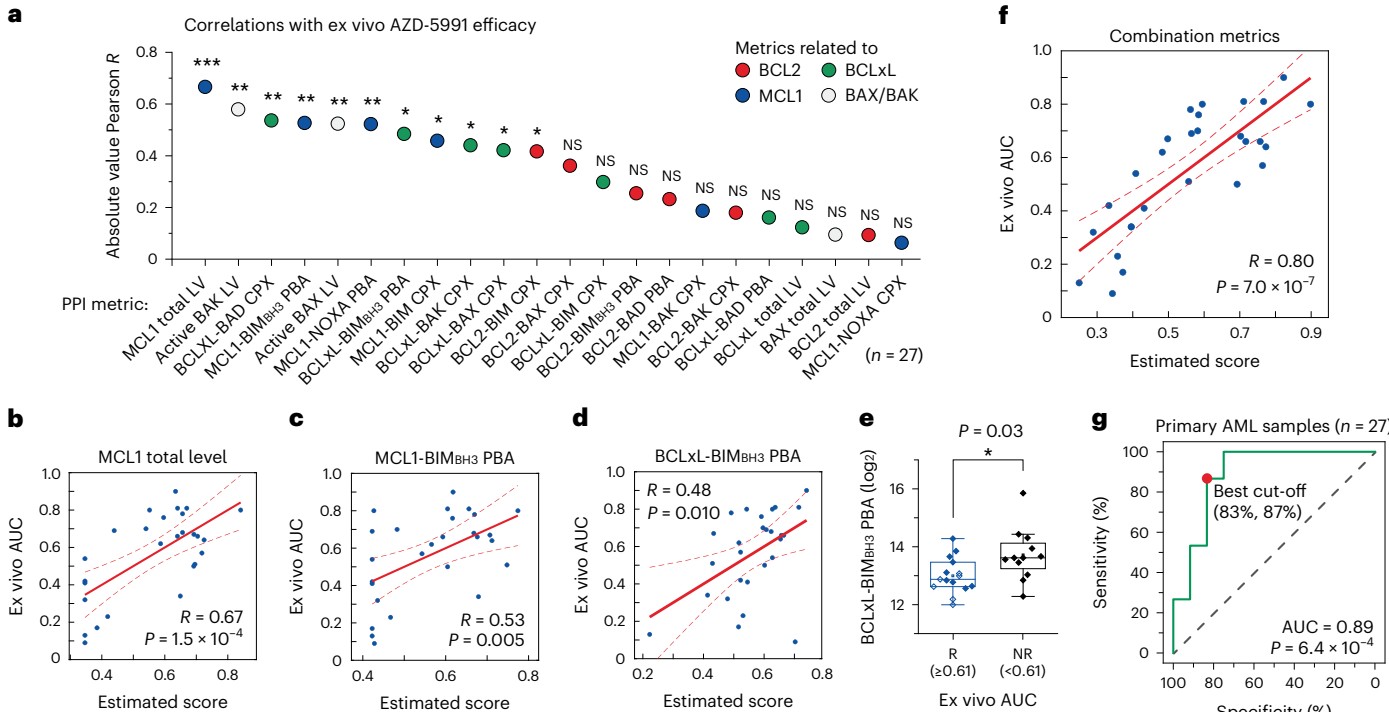

**Fig. 5 | Construction of an MCL1 inhibitor efficacy model. a**, The absolute Pearson correlations between ex vivo AUC of AZD-5991 and PPI profiles for primary AML samples (*$P < 0.05$, **$P < 0.01$, ***$P < 0.001$; $P$ values are provided as Source Data) ($n = 27$). **b–d**, Correlations between ex vivo AUC and single PPI metrics. MCL1 total level (coefficient: 0.05, $P = 1.5 \times 10^{-4}$) (**b**), MCL1-BIM$_{BH3}$ PBA (coefficient: 0.04, $P = 0.005$) (**c**), BCLxL-BIM$_{BH3}$ PBA (coefficient: −0.13, $P = 0.010$) (one-sided $F$-test) (**d**). **e**, Comparison of BCLxL-BIM$_{BH3}$ PBA counts for the AZD-5991 to responsive (ex vivo AUC ≥ 0.61, $n = 12$) and non-responsive

(ex vivo AUC < 0.61, $n = 15$) AML samples (two-sided two-sample $t$-test, $P = 0.03$). For the boxplots, the centre line represents the median, the box limits are the upper and lower quartiles, and the whiskers represent 1.5× interquartile range. **f**, Correlation between ex vivo AUC and the combination of multiple PPI metrics (MCL1 total level, BCLxL-BIM$_{BH3}$ PBA, BCLxL-BAK CPX) (one-sided $F$-test, $P = 7.0 \times 10^{-7}$). **g**, ROC curve between the estimated score and the ex vivo efficacy (two-sided $t$-test, $P = 6.4 \times 10^{-4}$).

efficacy (Fig. 5f and Supplementary Table 6). While several BCL2-related parameters alone already displayed high correlations with ABT-199 efficacy (signalling the singular dependence on BCL2) (Fig. 4b), the MCL1-related metrics required combination with BCLxL-involving metrics to enhance the correlation, pointing to a complex interplay and dynamic between MCL1 and BCLxL protein pools (Fig. 5f and Supplementary Table 6 versus Extended Data Fig. 8h and Supplementary Table 7). The principle of minimal information overlap between metrics appeared to be preserved here as well, given that the final combination encompassed occupied and unoccupied populations of both MCL1 and BCLxL with seemingly minimal overlap. Through the combined model for AZD-5991 effectiveness, we achieved a high accuracy amounting to an AUC of 0.89 and statistically significant discriminative performance for binary prediction with ROC analysis (Fig. 5g). Altogether, our multiparameter PPI data not only facilitate the construction of an analysis model for the efficacy of BH3 mimetics, but also allow us to identify the relative contributions of each assay data and the intricate balances among the protein complexes, potentially shedding light on the drugs' modes of action.

## Using the PPI analysis model as a predictive biomarker for ABT-199 efficacy

We next explored whether BCL2 SMPC and PBA profiling could proactively guide therapeutic decisions. BCL2 SMPC and PBA profiling was performed for ten patients who were treated with ABT-199 through oral administration. Treatment outcomes were evaluated in accordance with European LeukemiaNet (ELN) recommendations[77], and sequential samples were collected throughout the treatment journey (Fig. 6a and Supplementary Table 8)[78].

For example, two patients, BC-6524 and BC-7230, exhibited contrasting patterns in our PPI profiling, mirroring their disparate responses to ABT-199 treatment (Fig. 6b–d). Within the combination analysis model detailed in Fig. 4f, BC-6524 was anticipated to respond to a BCL2-targeted inhibitor with a high estimated score of 0.90, while BC-7230 was predicted as a non-responder with an estimated score of 0.27 (Fig. 6b). Specifically, BC-6524 displayed significantly higher counts for the BCL2-BIM$_{BH3}$ PBA and BCL2-BAX CPX assays compared with BC-7230, while the BCLxL-BAK complex counts were low for both patients (Fig. 6c). Impressively, these diagnostic predictions were strongly corroborated by the in vivo responses. For patient BC-6524, peripheral blast counts began to decrease from day 2, culminating in confirmed complete remission as per bone marrow examination (Fig. 6d, red). Conversely, patient BC-7230 did not respond to ABT-199, as evidenced by increasing peripheral AML blasts from day 3 (Fig. 6d, grey).

When the 10 cases grouped into responsive (including complete remission and partial remission) and non-responsive categories, our predictions for responder (R) and non-responder (NR) with a threshold score of 0.61 aligned remarkably with the in vivo clinical outcomes for the nine patients (out of a total of ten patients), including those for BC-6524 and BC-7230 previously described, achieving 100% sensitivity and 83% specificity (Fig. 6e, Extended Data Fig. 10a–g and Supplementary Table 8). Moreover, the estimated scores for the initial responses between the two groups manifested a significant difference as determined by the Mann–Whitney test, which further increased when we included six additional samples from the same cohort collected after relapse (Fig. 6f and Extended Data Fig. 10h). An intriguing case was BC-7052, where our model initially projected a favourable

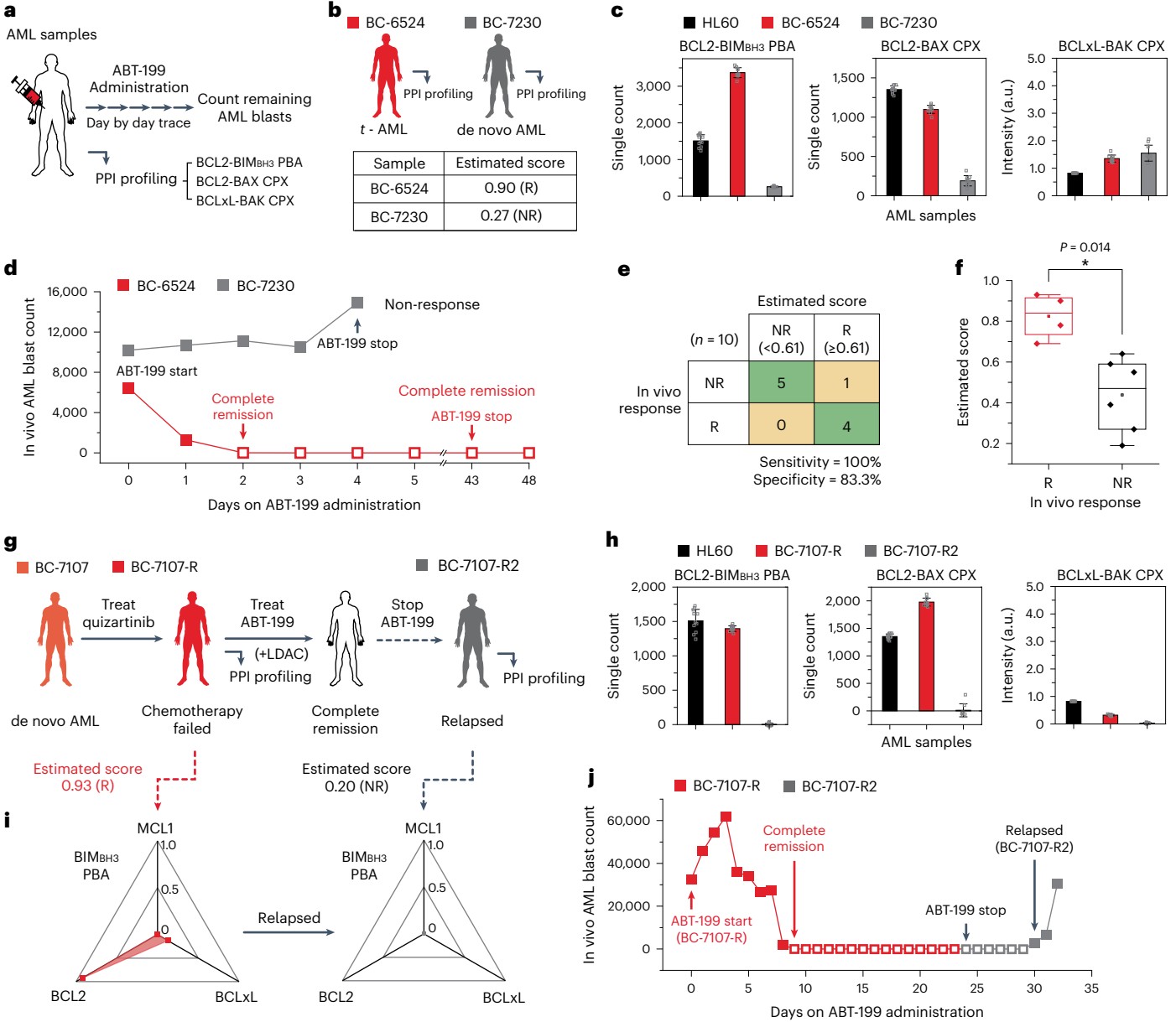

**Fig. 6 | Using the PPI analysis model as a predictive biomarker for in vivo ABT-199 response. a**, Schematic of the study. Complete blood cell count analysis was carried out on a daily basis to count the number of AML blasts in the primary AML samples. **b**–**d**, Comparison of the PPI profiles from the BC-6524 and BC-7230 samples. Clinical features and the estimated scores with PPI diagnostic results for ABT-199 (**b**), comparison of BCL2-family PPI profiles ($n = 10$ independent images) (**c**), changes in in vivo AML blast counts through the days after ABT-199 administration (**d**). **e**, Confusion matrix for the drug response prediction based on the estimated score and the in vivo drug responses ($n = 10$). **f**, Comparison of the estimated scores of patients with AML between responsive ($n = 4$) and non-responsive ($n = 6$) patients for in vivo ABT-199 administration (two-sided

Mann–Whitney test, *$P = 0.014$). For the boxplots, the centre line represents the median, the box limits are the upper and lower quartiles, and the whiskers represent 1.5× interquartile range. **g**, Clinical features and the ABT-199 administration history of BC-7107. **h**, Tracking the changes in BCL2-family PPI profiles after relapse of BC-7107-R ($n = 10$ independent images). **i**, Tracking the changes in BIM$_{BH3}$ PBA profiles and PPI diagnostic results after relapse of BC-7107-R. The data were normalized to the BCL2-BIM$_{BH3}$ PBA level of HL60 cells. **j**, Changes in in vivo AML blast counts from BC-7107-R through the days after ABT-199 administration. The estimated scores were calculated from the model in Fig. 4f (**b**,**g**). Data represent means ± s.d.

score of 0.64 (Extended Data Fig. 10a). After a short initial response, the patient entered partial remission and ceased ABT-199 treatment. Interestingly, in ex vivo efficacy determination, the sample recorded a high AUC value of 0.63, precisely mirroring the prediction by the analysis model (Supplementary Data Set 2). This observation highlights a current model limitation, as the model was trained on ex vivo drug efficacy, and suggests that incorporating factors unique to in vivo cancer evolution may refine the model's predictive capabilities. For example, the BCLxL PBA count, absent from the current model, was

conspicuously high for BC-7052, hinting at a scenario where BCLxL proteins might become upregulated in activity as ABT-199 treatment persisted (Extended Data Fig. 10a, right).

In addition to the analysis of the BC-7064 case presented in Fig. 4h, we conducted a longitudinal tracking study on another case, BC-7107, to gain insights into AML evolution during ABT-199 treatment (Fig. 6g). This patient, who had relapsed refractory disease and previously undergone treatment with another chemotherapeutic agent (Quizartinib), initially exhibited a robust BCL2 dependence (with an estimated score

of 0.93) (Fig. 6g–i). Notably, the patient achieved complete remission upon combined treatment with ABT-199 and LDAC (low-dose cytarabine) (Fig. 6g,j). However, after one month of treatment, relapse occurred (Fig. 6j). Interestingly, PPI profiling after the relapse showed nearly background signals for all three examined anti-apoptotic proteins, suggesting that the cancer might have developed a distinct pathway to evade apoptosis initiation (Fig. 6h–j).

## Discussion

In our pursuit of establishing a precision-medicine system for protein-complex-targeting therapies, we harnessed the capabilities of the SMPC technique to comprehensively profile BCL2-family protein complexes within specific AMLs. This endeavour involved optimizing extraction protocols applicable to clinical samples, ensuring the preservation of most protein complexes and retaining specific BCL2-family protein conformations through gentle lysis techniques. The integration of an anti-fouling polymer layer minimized cross-interactions and non-specific adsorption during surface pull-down procedures. In addition, the direct single-molecule counting of target protein complexes demonstrated sustained sensitivity, effectively working even when only a few hundred protein complexes were immobilized per imaging area (~$10^4$ μm²). These enabled the assessment of over 20 distinct PPI species within clinical samples comprising as few as ~$1.2 \times 10^6$ cells, achieving a level of sensitivity and accuracy amenable to clinical applications.

Given that BH3 mimetics function as PPI inhibitors, conventional genomic and proteomic profiling inherently falls short in capturing the dynamic rewiring of the PPI network and, consequently, predicting the response to specific BH3 mimetics[46–49,79,80]. Indeed, our findings demonstrated that the counts of BCL2-family protein complexes underwent drastic changes of more than 3- to 5-fold, while corresponding protein levels exhibited marginal alterations of approximately less than 30%. Moreover, the effect of the two examined BH3 mimetics, ABT-199 and AZD-5991, on BCL2-family protein complexes appeared distinct, probably attributed to their distinct mechanisms of action (as discussed below). This underscores how relying solely on genomic mutations and proteomic levels, without considering their interconnectedness, may fail to fully elucidate the intricate complexity of BCL2-family biology underlying BH3-mimetic efficacy.

Presently, there are competing hypotheses regarding the primary species of protein complexes that BH3 mimetics target to dissociate for initiating apoptosis[16,81,82]. The activated BAX and BAK proteins, responsible for forming BAX-BAK pores and mediating mitochondria outer membrane permeabilization, can either be newly expressed by cells or released from anti-apoptotic proteins. In the former scenario, BH3 mimetics saturate the target anti-apoptotic proteins, preventing further sequestration of nascent BAX and BAK proteins. In the latter, BH3 mimetics bind to the binding groove of anti-apoptotic proteins, causing the release of previously bound pro-apoptotic proteins. A detailed analysis of changes in relevant protein complexes and protein levels, along with their intricate balances, revealed that ABT-199 primarily acted by disassembling pre-existing BCL2 protein complexes. Conversely, the MCL1 inhibitor AZD-5591 appeared to exert its efficacy by interfering with the binding of newly produced BAK proteins to MCL1. Furthermore, tracing the behaviour of the two types of BCL2 protein complex over time indicated that the dissociation of BCL2-BAX complexes, not BCL2-BIM complexes, closely correlated with the onset of apoptosis in AML cells induced by ABT-199. Our findings are consistent with previous studies that employed fluorescence lifetime imaging microscopy to observe the release of BAD and BID, but not of BIM, from BCL2 and BCLxL by ABT-263 (ref. 83). In addition, these studies noted pro-apoptotic effects triggered by the disruption of BCLxL-BAX complexes through BAD in isolated mouse liver mitochondria[84]. Indeed, in our multiparametric measurement and linear regression analysis, the quantity of BCL2-BAX complexes was more closely tied to the ex vivo efficacy of ABT-199 than to the quantity of BCL2-BIM complexes.

This led us to conclude that, in the specific AML cases studied, the dissociation of BCL2-BAX complexes primarily drove the efficacy of ABT-199.

These data compelled us to hypothesize a particular group of driver protein complexes that powerfully and positively correlated with the efficacy of the BH3 mimetics. We also found protein or PPI metrics associated with the drug efficacy with negative coefficients, which we defined as resistance complexes. The resulting analysis models included both driver and resistance complexes, reflecting the competing roles of different anti-apoptotic proteins in achieving the BH3-mimetic efficacy. In the combined model for ABT-199 effectiveness, BCL2-BIM PBA and BCL2-BAX CPX worked as the driver complexes, while BCLxL-BAK CPX worked as the resistance complex. These resistance complexes, displaying negative correlations, have also been consistently noted in the BH3 profiling method, where the priming effect triggered by the HRK peptide, a specific binder to BCLxL, acts as a detractor for ABT-199 efficacy[52,76]. Similarly, the BCLxL-BAK complex has been identified as a critical negative factor in our efficacy prediction model for MCL1 inhibitors. These observations robustly imply that BCLxL compensates within the anti-apoptotic pathway when either BCL2 or MCL1 is overwhelmed by BH3 mimetics. The coefficient for BCLxL-BAK CPX, however, was much smaller than those of BCL2-related parameters, consistent with the singular dependence of AMLs on BCL2. On the other hand, the analysis model for AZD-5591 efficacy, consisting of MCL1 total level, BCLxL-BAK CPX and BCLxL-BIM$_{BH3}$ PBA, assigned larger coefficients to two BCLxL-involved parameters than it did for the MCL1 total level, suggesting that MCL1 was in direct competition with BCLxL in mediating the effectiveness of AZD-5591. It is important to note that our current suite of assays lacks the capability to detect certain anti- and pro-apoptotic proteins in the BCL2 family, including BCL2-related protein A1 (BCL2A1) and BID. By extending our assay to incorporate these proteins—achievable through the screening of appropriate antibodies for IP and detection—we anticipate a more comprehensive profiling of the BCL2-family PPI network, as well as enhanced accuracy in our analysis model.

Our analysis model bears an intrinsic limitation as it was constructed on the basis of drug efficacy determined ex vivo. In addition, due to the relatively small size of the patient cohort employed in this study, the applicability of the model to larger cohorts remains untested. Notwithstanding these considerations, we took the step of applying the analysis model for prospective stratification of ABT-199 responders. Remarkably, in nine out of ten cases, predictions based on three PPI metrics showed a significant correlation with clinical responses, achieving 100% sensitivity and 83.3% specificity. In addition, our longitudinal tracking approach allowed us to observe how PPI connectivity evolved among the BCL2-family proteins, ultimately contributing to the development of resistance and cancer recurrence. While obtaining frozen BMMC or PBMC samples from haematological malignancies does not present substantial challenges, we expect that the extraction of protein complexes from fixed tissues will be a crucial technical factor for extending this method to solid tumours. Furthermore, ready access to the surface-passivated reaction chip and the single-molecule imaging device used in this study is essential for the widespread adoption of this diagnostic method in clinical decision-making.

In summary, our findings advocate for the combination of the SMPC and PBA techniques as an effective method of scrutinizing multiple protein complexes in clinical samples with heightened sensitivity and precision. Although emerging technologies such as single-protein sequencing primarily serve as discovery tools[85], the technique showcased in this study is tailored to the examination of dozens of protein complexes in clinical specimens. These profiles can be integrated to generate information that is most directly pertinent to drug efficacies, thereby facilitating treatment decisions that would be considerably challenging to achieve through genomic and proteomic profiling alone. Given the rise in protein-complex-targeting therapies, it will be insightful to see whether the methodologies established herein could

be extended to address other disease-associated protein complexes and drugs beyond the BCL2 family and BH3 mimetics.

## Methods

### Primary samples from patients with AML

The study was conducted according to the Declaration of Helsinki and was approved by the Institutional Review Board (IRB) at Seoul National University Hospital (IRB number: H-1910-176-107). Informed consent was taken from all patients before participation in any study-related procedure. No compensation was given for participation. The cryo-preserved BMMCs and PBMCs collected from 35 Korean patients with AMLs undergoing ABT-199 treatment at Seoul National University Hospital were used, regardless of previous medical history. Samples were collected between January 2014 and December 2019. Acute promyelocytic leukaemias and biphenotypic leukaemias were excluded. Mononuclear cells were isolated by Ficoll gradient centrifugation. All samples were frozen in liquid nitrogen before flow cytometry and SMPC analysis. The cell collection method is described in the section 'Cell culture'. Culturing method for flow cytometry assay is described in the section 'Drug treatment and drug efficacy measurement'. Detailed information on each sample is described in Supplementary Data Set 1 and Reporting summary.

### Antibodies

Anti-rabbit immunoglobulin G (IgG) with biotin conjugation (111-065-144, Jackson ImmunoResearch, 1:200) and anti-mouse IgG with biotin conjugation (715-066-151, Jackson ImmunoResearch, 1:200) were used to immobilize the antibodies for surface IP. Anti-RFP antibody with biotin conjugation (ab34771, Abcam, 1:200) and anti-GFP antibody with biotin conjugation (ab6658, Abcam, 1:200) were used to immobilize mCherry- or eGFP-labelled proteins on the surface. Anti-MCL1 (94296S, Cell Signaling Technology, D2W9E, 1:100), BCLxL (MA5-15142, Thermo Fisher, C.85.1, 1:100), BCL2 (4223S, Cell Signaling Technology, D55G8, 1:100), BAX (5023S, Cell Signaling Technology, D2E11, 1:100) and BAK (ab32371, Abcam, Y164, 1:100) antibodies were used to immobilize the corresponding proteins via surface pull-down. Anti-MCL1 (MAB8825, Abnova, C9, 1:100), BCLxL (NBP1-47665, Novus Biologicals, OTI4A9, 1:100), BCL2 (sc-7382, Santa Cruz Biotechnology, C2, 1:100), BAX (MABC1176M, Sigma Aldrich, 6A7, 1:100), BAK (sc-517390, Santa Cruz Biotechnology, AT38E2, 1:100), BIM (sc-374358, Santa Cruz Biotechnology, H5, 1:100), BAD (sc-8044, Santa Cruz Biotechnology, C7, 1:100) and NOXA (sc-56169, Santa Cruz Biotechnology, 114C307, 1:100) antibodies were used to detect the corresponding proteins in the protein total level and the protein complex assays. To measure BCL2-BAX complex level, anti-BAX (5023S, Cell Signaling Technology, D2E11, 1:100) and anti-BCL2 (BMS1028, Invitrogen, Bcl-2/100, 1:100) antibodies were used to immobilize and detect the BCL2-BAX complex, respectively. Anti-rabbit IgG with Cy3 conjugation (111-165-046, Jackson ImmunoResearch, 1:1,000) and anti-mouse IgG with Cy3 conjugation (715-165-151, Jackson ImmunoResearch, 1:1,000) were used to label the detection antibodies for immunoassay. Anti-BCL2 antibody (BMS1028, Invitrogen, Bcl-2/100, 0.5 μg), mouse IgG1 kappa isotype control (554121, BD Biosciences, 0.5 μg) and anti-mouse IgG with PE conjugation (715-116-150, Jackson ImmunoResearch, 1.5 μl) were used to detect the protein levels for quantitative flow cytometry. Anti-BCL2 antibody (sc-7382, Santa Cruz Biotechnology, C2, 1:1,000) and HRP-linked anti-mouse IgG (7076S, Cell Signaling Technology, 1:1,000) were used to detect the protein levels for western blotting.

### Drug reagents

Staurosporine (HY-15141, MedChemExpress), ABT-199 (HY-15531, MedChemExpress) and AZD-5991 (C-1060, Chemgood) were used to treat the blood cancer cell lines or primary AML cells. The drugs were diluted to 10 mM with dimethylsulfoxide (DMSO) and stored at −80 °C.

### Cell culture

HL60, THP-1, U937 and Ramos cells were purchased from Korean Cell Line Bank. HEK293T and SU-DHL8 cells were purchased from the American Type Culture Collection (ATCC). NB4 cells were purchased from the German Collection of Microorganisms and Cell Cultures (DSMZ). PC9 cells were provided by Y. Hayata (Kyushu University Faculty of Medicine, Japan). HEK293T cells were cultured in DMEM medium (D6429, Sigma Aldrich) supplemented with 10% (v/v) fetal bovine serum (FBS; 26140-079, Gibco) and 100 μg ml$^{-1}$ penicillin/streptomycin (15140-122, Gibco). HL60, THP-1, NB4, U937, Ramos, SU-DHL8 and PC9 cells were cultured in RPMI 1640 medium (R8758, Sigma Aldrich) supplemented with 10% (v/v) FBS and 100 μg ml$^{-1}$ penicillin/streptomycin. For THP-1, 0.05 mM of 2-mercaptoethanol (M3148, Sigma Aldrich) was added to culture media. All cells were incubated in a humidified incubator at 37 °C and 5% $CO_2$. HEK293T and PC9 cells were rinsed with cold Dulbecco's (D)PBS (D8537, Sigma Aldrich) before collection. Cells were collected with the scraper (90020, SPL Life Sciences) in 1 ml of cold DPBS. Suspension-type cells were collected by centrifugation (500 $g$, 5 min, 4 °C) and rinsed with 1 ml of cold DPBS. The cell suspensions were centrifuged at 500 $g$ for 5 min at 4 °C. The supernatants were discarded after centrifugation and the cell pellets were stored at −80 °C by snap freezing with liquid nitrogen.

### Fluorescence-labelled protein constructions

All full-length BCL2-family proteins (*BCL2* (HG10195-M, SinoBiological), *BCLxL* (HG10455-M, SinoBiological), *MCL1* (HG10240-M, SinoBiological), *BIM$_{EL}$* (HG13816-G, SinoBiological), *BAD* (HG10020-M, SinoBiological), *BAX* (HG11619-M, SinoBiological), *BAK* (HG10450-M, SinoBiological) and *NOXA* (HG16548-U, SinoBiological)) were isolated from their respective human complementary (c)DNA. cDNA of *BIM$_{BH3}$* (MRQAEPADMRPEIWIAQELRRIGDEFNAYYARR) was isolated from *BIM$_{EL}$* cDNA. cDNA of *BAK fragments* (residues 24–69 ($\alpha_1$ helix), 70–91 ($\alpha_2$–$\alpha_3$ helices), 90–150 ($\alpha_3$–$\alpha_5$ helices) and 146–187 ($\alpha_6$–$\alpha_8$ helices)) were isolated from *BAK* cDNA. All cDNAs were cloned into pCMV vectors with either eGFP or mCherry sequences to generate fluorescence-labelled proteins using Gibson assembly. The fluorescence proteins were fused to the carboxyl end of BCL2-family proteins. The resulting plasmids were introduced into HEK293T cells through transient transfection. Plasmid DNA at 10 μg was mixed with 30 μg of linear polyethyleneimine (23966-100, Polysciences) in 1 ml of serum-free DMEM. The mixture was introduced into ~2 × 10$^6$ HEK293T cells in a 90 mm$^2$ cell culture plate.

### Cell lysis

All collected cell pellets were resuspended with the lysis buffer (0.2% or 1% (v/v) detergent, 50 mM HEPES at pH 7.4, 150 mM NaCl, 10% (v/v) glycerol, 1 mM EDTA, 2% (v/v) protease inhibitor cocktail (P8340, Sigma Aldrich), 2% (v/v) phosphatase inhibitor cocktail 2 (P5726, Sigma Aldrich) and 2% (v/v) phosphatase inhibitor cocktail 3 (P0044, Sigma Aldrich)). Triton X-100 (X100, Sigma Aldrich), glyco-diosgenin (GDN) (GDN101, Anatrace), digitonin (D141, Sigma Aldrich), Tween 20 (P2287, Sigma Aldrich), 3-[(3-cholamidopropyl) dimethylammonio]-1-propanesulfonate hydrate (CHAPS) (C3023, Sigma Aldrich) and sodium dodecyl sulfate (SDS) (L3771, Sigma Aldrich) were used to optimize the lysis protocol. Triton X-100, GDN, digitonin, Tween 20 and CHAPS were used at 1% (v/v) and SDS was used at 0.2% (v/v) in lysis buffer. HEK293T cells expressing probes for the PBA experiments: BIM$_{BH3}$-eGFP, BIM$_{EL}$-eGFP, BAD-eGFP and NOXA-eGFP, were lysed with Triton X-100 lysis buffer. HEK293T cells expressing fluorescence protein labelled BCL2-family proteins (BCL2-mCherry, BCLxL-mCherry, MCL1-mCherry, BCL2-eGFP, BCLxL-eGFP, MCL1-eGFP, BAX-eGFP, BAK-eGFP and BAK fragments-eGFP) were lysed with GDN lysis buffer for immunoassay and PBA. The construct designs and protein expressions in HEK293T cells for fluorescence protein labelled BCL2-family proteins are described above. All cancer cell lines and primary AML cells were lysed with GDN lysis buffer for immunoassay and PBA.

For the cell lysis, the cell pellets were resuspended with the designated lysis buffer and incubated for 30 min at 4 °C. After lysis, the cell suspensions were centrifuged at 15,000 g for 10 min at 4 °C. The supernatants were isolated after centrifugation. The total protein concentration for each supernatant was measured with a DC protein assay kit (5000113, 5000114, 5000115, Bio-Rad) following manufacturer instructions. The total concentration of the fluorescence proteins in each supernatant was measured with a Sense microplate reader (425-301, HIDEX). To quantify the fluorescence proteins, a laser wavelength of 485 nm was used for eGFP excitation and 544 nm was used for mCherry excitation. Serial dilution experiments using eGFP and mCherry were performed to generate the calibration curves. The aforementioned calibration curves were used to quantify the concentrations of fluorescently labelled proteins. The supernatants were then aliquoted and stored at −80 °C, followed by snap freezing with liquid nitrogen.

### Drug treatment and drug efficacy measurement

All blood cancer cells and primary AML cells were precultured in a 25T flask (non-treated surface) for 3 h before drug treatment. All cells were seeded in 4.5 ml of culture media and cultured in a humidified incubator at 37 °C and 5% $CO_2$. Staurosporine, ABT-199 and AZD-5991 were then administered to cells at various concentrations. All drugs were initially diluted with DMSO to 1 mM. For experiments, serial dilution was performed with non-serum RPMI 1640 to reach the target concentration. To determine the efficacy of drugs in blood cancer cells and primary AML cells, an Annexin V/propidium iodide apoptosis assay was performed, where cells were stained with FITC-Annexin V (640906, Biolegend) and propidium iodide (421301, Biolegend). After drug treatment, $5 \times 10^5$ cells were rinsed with cold DPBS and collected by centrifugation (500 g, 5 min, 4 °C). After the supernatants were discarded, the cell pellets were resuspended with 100 µl of cold Annexin V-binding buffer (422201, Biolegend). After resuspension, 10 µl of FITC-Annexin V solution and 5 µl of propidium iodide solution were added into the cell suspensions. The cell suspensions were incubated for 15 min at room temperature, avoiding exposure to light. After that, 400 µl of cold Annexin V-binding buffer was added to avoid overstaining. The stained cells were analysed by flow cytometry (SH800S, Sony) and using the SH800S Cell Sorter Software 2.1.5 (Sony). The proportions of Annexin V−/propidium iodide− cells from each sample were calculated and converted to viability (%). On the basis of the measured viability, the drug efficacy curve was fitted using the 'logistic' function in the data analysis software OriginPro (2022), and the area under the curve (AUC) was measured from the fitted curve.

### Single-molecule pull-down and co-IP for BCL2-family proteins

NeutrAvidin (5 µg ml$^{-1}$, 31000, Thermo Fisher) was loaded into each reaction chamber of the Pi-Chip (PROTEINA) and incubated for 10 min. The imaging chip was washed with Triton X-100 buffer (0.1% (v/v) Triton X-100, 50 mM HEPES at pH 7.4, 150 mM NaCl) to remove unbound NeutrAvidin. Biotin-conjugated IgGs (anti-rabbit IgGs) were loaded to each reaction chamber at 1:200 in Triton X-100 buffer and incubated for 10 min, followed by washing with Triton X-100 buffer. Then, the previously mentioned monoclonal antibodies (anti-MCL1, BCLxL, BCL2, BAX and BAK antibodies for surface IP) were introduced into each reaction chamber at a dilution of 1:100 in Triton X-100 buffer and then incubated for 10 min. The imaging chip was washed with GDN buffer (0.01% (w/v) GDN, 50 mM HEPES at pH 7.4, 150 mM NaCl, 1% (v/v) glycerol, 1 mM EDTA). After washing, crude cell extracts were diluted on the basis of the total protein concentrations and loaded into the reaction chambers. During the process of immunoprecipitating fluorescence-labelled proteins, anti-RFP or anti-GFP antibodies were introduced at a dilution of 1:200 in Triton X-100 buffer. This mixture was then incubated for 20 min as an alternative to using biotin-conjugated IgGs. To measure the total level of BCL2-family proteins, cell extracts at a concentration of 1 mg ml$^{-1}$ were introduced into the reaction chambers.

These chambers were pre-coated with primary antibodies for immunoprecipitation, and the samples were allowed to incubate for 1 h to capture the target proteins. To measure the total level of BAX and BAK, the cell extracts were mixed with the activation buffer (2% (v/v) Triton X-100, 50 mM HEPES at pH 7.4, 150 mM NaCl and 1 mM EDTA) at a ratio of 1:1 for 30 min before pull-down. After activation, the cell extracts were loaded into the reaction chambers and incubated for 1 h. To measure the PPI complex levels, 2 mg ml$^{-1}$ of cell extracts were loaded into each reaction chamber, allowing 2 h of incubation to pull down the target PPI complexes.

After the surface pull-down of target proteins or protein complexes, the previously mentioned primary antibodies were introduced to detect either total protein levels (anti-MCL1, BCLxL, BCL2, BAX and BAK antibodies for immunoassay) or protein complexes (anti-BAX, BAK, BIM, BAD and NOXA antibodies for immunoassay). These antibodies were loaded into the reaction chambers at a dilution of 1:100 in Triton X-100 buffer and then incubated for 1 h. To label the detection antibodies, Cy3-conjugated IgGs (anti-mouse IgGs) were introduced into the reaction chambers at a dilution of 1:1,000 in Triton X-100 buffer, and the chambers were incubated for 20 min. To measure the PBA, 1 mg ml$^{-1}$ of cell extracts was loaded into each reaction chamber and incubated for 1 h. After that, the imaging chip was washed with GDN buffer. The HEK293T cells expressing PPI probes were lysed with Triton X-100 lysis buffer. The crude cell extracts containing the PPI probes were diluted to achieve the target eGFP concentration. Subsequently, they were incubated for 10 min with the immunoprecipitated bait proteins. To ascertain the dissociation constants ($K_d$), we quantified the fractional occupancy of surface baits at equilibrium. This was calculated as the ratio of bound probes to the number of surface baits within our designated field of view. We then measured these fractional occupancy values across a range of probe concentrations and fitted them to the following equation, which enabled the determination of the dissociation constant for a specific PBA reaction.

$$\text{Fractional occupancy} = \frac{[\text{PPI probe}]}{K_d + [\text{PPI probe}]} \qquad (1)$$

### Single-molecule fluorescence imaging

Single-molecule fluorescence signals were collected using a PI-View system (PROTEINA) and assessed using the PI-Analyzer software (v.1, PROTEINA), which allows for single-molecule fluorescence imaging at two different excitation lasers, 488 nm and 532 nm[55,57,65]. The emitted photons from a single fluorophore were collected by employing a sensitive sCMOS (scalable complementary metal-oxide semiconductor) camera. An autofocus system provides high-throughput image acquisition across the whole imaging chip that contains 40 reaction chambers. To determine the single-molecule counts for each assay, ~10 different locations were imaged in a single reaction chamber. Each image was acquired for 5 frames with 100 ms time resolution, and those frames were averaged to generate a single snapshot, which reduced random noise and improved signal-to-noise ratio. Protocols for image processing are detailed in Extended Data Fig. 2. The current imaging system identifies and counts up to 12,000 single-molecule fluorescence spots within a 100 × 100 µm² field of view, an upper limit for the identification of diffraction-limited spots. When the number of such spots exceeds this limit, overlapping begins to occur, compromising the ability to identify individual spots (Extended Data Fig. 2d). To address this limitation, the total fluorescence signals captured across all pixels within the field of view were integrated. Simultaneously, by recording time traces of photobleaching, a relationship between the number of single-molecule spots and the total fluorescence intensity was established, particularly in the range that permitted the identification of individual spots (that is, below the upper limit), which was typically linear (Extended Data Fig. 2e,f). This linear relationship ultimately enabled extrapolation of total fluorescence intensity into

corresponding single-molecule counts in regions exceeding the upper limit (Extended Data Fig. 2g). The computational analysis codes are available on GitHub[86].

### Drug efficacy prediction model analysis

Drug efficacy prediction models for patients with AML were generated using linear regression analysis in MATLAB 2021a. The protein and protein complex data from the primary AML cells were all converted to a $\log_2$ scale. Any instance where the number of single-molecule fluorescence spots was less than 100 counts were fixed to the minimum value of 100 counts for log transformation. For BCLxL-BAX, BCLxL-BAK and BCL2-BAK protein complexes, the total fluorescence signal data were used instead of single-molecule spot data, and fluorescence signals less than $10^7$ were fixed to the minimum value ($10^7$ a.u.) for log transformation. The ex vivo drug efficacies (ex vivo AUCs) of the primary AML cells were obtained from the viability measurement method described in the section 'Drug treatment and drug efficacy measurement'. The AUCs were initially calculated from the fitted curves for viability after BH3-mimetic treatment, and ex vivo drug efficacies were normalized using ex vivo AUC = 1−(AUC/Maximal AUC). The correlation between the PPI profiling data and the ex vivo drug efficacy was calculated through linear regression with the custom analysis code. Each generated model provided the statistical indicators, $P$ values for the linear regression itself, as well as for the intercept. The models were accepted only when their $P$ values were statistically significant ($P < 0.05$). To construct combined metrics models, various randomly selected combinations of PPI profiles were assessed to identify a model that met two criteria: high correlation (indicated by a high $R$ value) and statistical significance ($P < 0.05$). Lasso regression analysis for selecting metrics that are highly correlated with drug response were generated in Python 3.11. The training and test groups were randomly selected from the primary AML sample cohort. The Lasso regression models were trained using the training group and evaluated on the basis of Pearson's $R$ as well as prediction outcomes for the test group, and 67 models were selected from 10,000 different initial models (Extended Data Fig. 8e). The custom analysis codes and raw data utilized for generating the models can be accessed on GitHub[86].

### BH3 profiling

We followed the methods outlined in previous studies for BH3 profiling[74,75]. The MEB buffer (150 mM mannitol (M9647, Sigma Aldrich), 10 mM HEPES at pH 7.4, 50 mM KCl (P9541, Sigma Aldrich), 0.02 mM EGTA (BE004, Biosolution), 0.02 mM EDTA, 0.1% (w/v) BSA (A4737, Sigma Aldrich) and 5 mM succinate (S3674, Sigma Aldrich)), along with the staining solution (20 µg ml⁻¹ oligomycin (O4876, Sigma Aldrich), 50 µg ml⁻¹ digitonin, 2 µM JC-1 (ENZ-52304, Enzo Life Sciences) and 10 mM 2-mercaptoethanol (M3148, Sigma Aldrich) in the MEB buffer), were prepared and stored at 4 °C. The BH3 peptides (or BH3 mimetics) were diluted with the staining solution to achieve twice the target concentration, producing the 2x staining solution for each peptide. The synthesized BH3 peptides (BIM (MRPEIWIAQELRRIGDEFNA, purity=98%), BAD (LWAAQRYGRELRRMSDEFEGSFKGL, purity=99%) and HRK (SSAAQLTAARLKALGDELHQY, purity=99%) all synthesized by PEPTRON (South Korea)), DMSO, carbonyl cyanide 4-(trifluorometh-oxy) phenylhydrazone (FCCP; C2920, Sigma Aldrich), ABT-199 and AZD-5991 were used for BH3 profiling. We aliquoted 50 µl of the 2x staining solution into a black-coated 96-well plate to produce a BH3 profiling plate, which was then stored at −80 °C until use. To perform BH3 profiling for primary AML cells, $10^6$ primary AML cells were rinsed with 1 ml RPMI 1640 supplemented with 10% (v/v) FBS. After centrifugation (300 $g$, 5 min), the collected cells were resuspended in 1 ml MEB buffer. Subsequently, the resuspended cells were aliquoted into each well of the BH3 profiling plate that had been equilibrated to room temperature. The BH3 profiling plate was placed in a plate reader (Synergy H1, BioTek) set at a constant temperature of 32 °C, and fluorescence

was measured every 5 min over a period of 3 h. To measure the relative fluorescence units (RFU) at a wavelength of 590 nm, a light wavelength of 545 nm was used for JC-1 excitation. On the basis of the measured RFU, the AUC was measured using the data analysis software OriginPro (2022). The depolarizations were calculated using the equation below, and the BH3 profiling results for primary AML cells can be found in Supplementary Data Set 3.

$$\text{Depolarization} = 1 - \frac{\text{AUC}_{\text{sample}} - \text{AUC}_{\text{FCCP}}}{\text{AUC}_{\text{DMSO}} - \text{AUC}_{\text{FCCP}}} \qquad (2)$$

### Quantitative flow cytometry

We employed the methods outlined in previous studies to quantify protein levels using flow cytometry[87]. Briefly, $10^6$ primary AML cells were rinsed with 1 ml RPMI 1640 supplemented with 10% (v/v) FBS and collected by centrifugation (300 $g$, 5 min). The collected cells were resuspended in 0.5 ml of 37 °C lyse fix buffer (558049, BD Biosciences). The cell suspensions were centrifuged (1,700 $g$, 5 min) and the supernatants discarded. The cell pellets were resuspended in staining wash buffer (0.5% (w/v) BSA and 0.1% (w/v) sodium azide (S2002, Sigma Aldrich) in DPBS) and aliquoted into two separate tubes at 100 µl each. Permeabilization wash buffer (PW) (900 µl, 557885, BD Biosciences) was then added to each tube, followed by a 10 min incubation at room temperature. Following the incubation, the tubes were subjected to centrifugation (1,700 $g$, 5 min) and the supernatant was subsequently reduced to 100 µl. The permeabilized cells were stained with either 0.5 µg of anti-BCL2 antibody or an isotype control antibody and incubated for 30 min. The cells were rinsed by adding 900 µl of PW and centrifuged (1,700 $g$, 5 min). The rinsed cells were then stained with 1.5 µl of PE-conjugated anti-mouse IgG and incubated for 30 min in a dark room to prevent light exposure. After rinsing to eliminate any unbound antibodies, the stained cells were resuspended in 350 µl of fix buffer (557870, BD Biosciences) for fixation and stored at 4 °C until analysis using a flow cytometer (SH800S, Sony). Quantum R-PE MESF beads (827, Bangs Laboratories) were utilized to produce calibration curves. These curves were subsequently employed, following manufacturer instructions, to quantify the levels of BCL2 in the samples.

### Western blotting

All cell extracts used for western blotting were produced following the steps described in the cell lysis section. The loading concentration of each sample was quantified on the basis of the total protein concentration. All samples were heated at 95 °C for 15 min with SDS containing sample buffer (EBA-1052, EPLIS Biotech), resolved with 6–15% gradient SDS–PAGE gels and transferred to PVDF membranes (IB401001, Thermo Fisher). PVDF membranes were blocked in 5% (w/v) skim milk in TBST buffer (20 mM Tris (93362, Sigma Aldrich) pH 7.6, 150 mM NaCl, 0.1% (w/v) Tween 20) for 1 h at room temperature. Each membrane was immunoblotted with anti-BCL2 antibody diluted at 1:1,000 in TBST buffer with 1% (w/v) BSA. After overnight incubation at 4 °C, membranes were immunoblotted with HRP-linked anti-mouse IgG diluted at 1:1,000 in TBST buffer with 1% (w/v) BSA for 1 h at room temperature. Protein bands were detected using an ECL (W3653-020, GenDEPOT) imaging system of a luminescent analyzer (ImageQuant LAS 4000 mini, Cytiba), and the band intensity was analysed with ImageJ (1.53a).

### Reporting summary

Further information on research design is available in the Nature Portfolio Reporting Summary linked to this article.

### Data availability

The data supporting the findings of this study are available within the article and its Supplementary Information. All raw data generated or

analysed during the study are available on GitHub at https://github.com/tyyoonlab-snu/Nat-Biomed-Eng-2023- (ref. 86). Source data are provided with this paper.

## Code availability

The custom MATLAB and Python codes used for the analysis of the drug-efficacy prediction model are available on GitHub at https://github.com/tyyoonlab-snu/Nat-Biomed-Eng-2023- (ref. 86).

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

## Acknowledgements

We thank S. H. Kim and J. S. H. Park from PROTEINA Co., Ltd for critical reading of the manuscript, and our colleagues, C. Kim, G. S. Eun and T. G. Kim at Seoul National University for active discussion concerning data analysis. T.-Y.Y. discloses support for the research described in this study from the National Grants for Leading Scientists of the National Research Foundation of South Korea (NRF) (grant number NRF-2021R1A3B1071354) and the Bio Medical Technology Development Program of the NRF (grant number NRF-2018M3A9E2023523). Y.K. discloses support for the research described in this study from the Korea Health Technology R&D Project of the Korea Health Industry Development Institute (KHIDI) (grant number HI14C1277).

## Author contributions

T.-Y.Y. conceived of and supervised the project. C.C., J.M.B., M.C., H.L., B.C., Y.K. and T.-Y.Y. designed the experiments. C.C. and M.C. performed the single-molecule pull-down and co-IP experiments. C.C. and H.K. measured the interchip c.v.s of the immunoassays. J.M.B. and Y.K. collected and characterized primary AML samples, and performed the in vivo ABT-199 treatment journey. C.C., Y.L. and H.P. developed the PPI probes for PBA. C.C., M.C., B.C., H.K., S.H. and Y.L. performed the PPI profiling for the primary AML samples. C.C. and Y.L. performed BH3 profiling for the primary AML samples. C.C. and S.H. performed flow cytometry for the primary AML samples. C.C. and M.C. measured the ex vivo drug efficacies of the primary AML samples. H.L. developed the ex vivo drug efficacy predicting programmes. C.C. analysed and visualized all data. C.C. and T.-Y.Y. wrote the paper with inputs from all authors.

## Competing interests

C.C., M.C., H.L., B.C., S.H., Y.L., Y.K. and T.-Y.Y. filed a patent on the findings of this work (patent number 10-2020-0157961, South Korea). The other authors declare no competing interests.

## Additional information

**Extended data** is available for this paper at https://doi.org/10.1038/s41551-024-01241-3.

**Correspondence and requests for materials** should be addressed to Youngil Koh or Tae-Young Yoon.

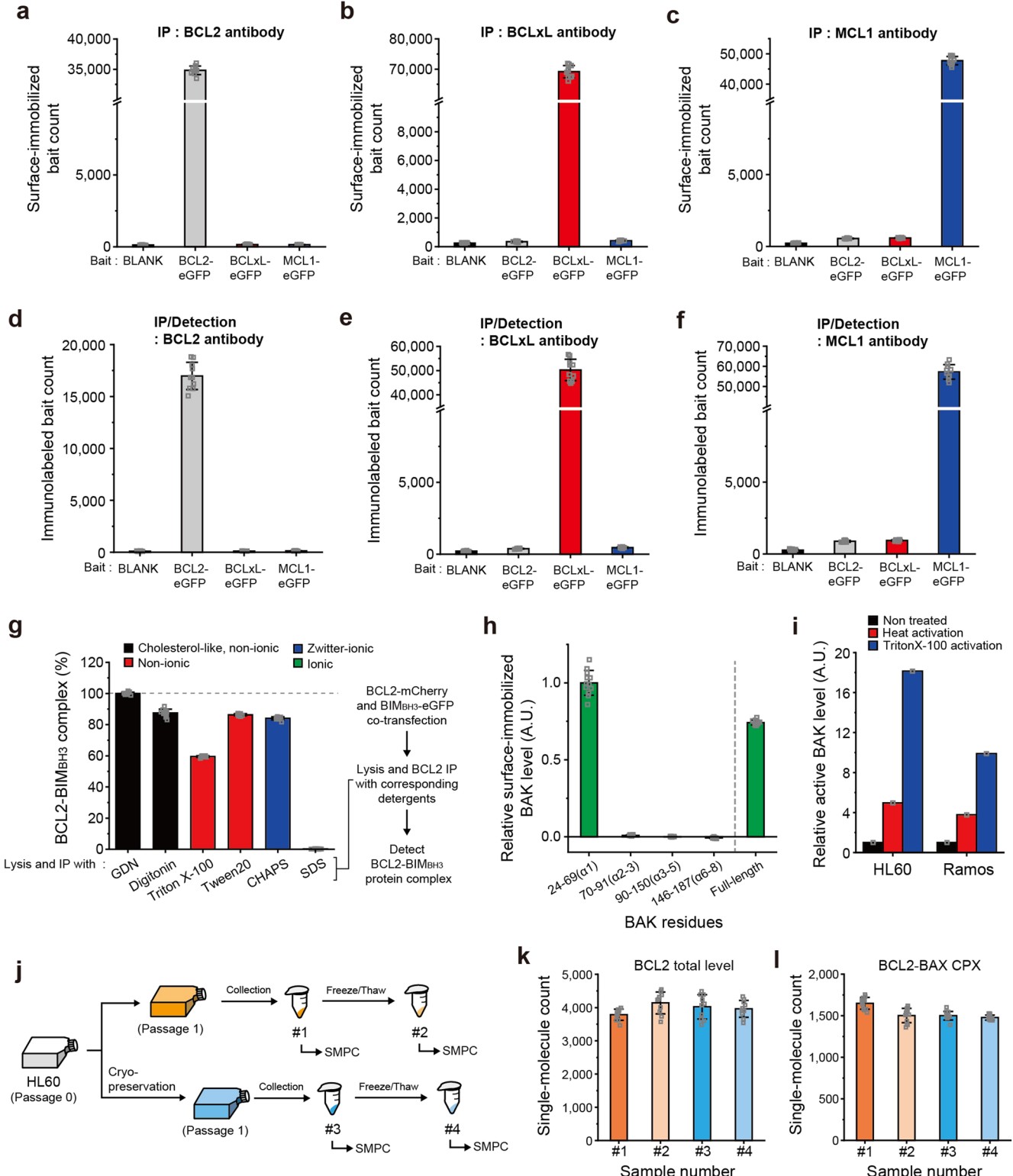

**Extended Data Fig. 1 | Development of single-molecule pull-down and co-IP (SMPC) to profile BCL2-family proteins. a-c**, Selection of monoclonal antibodies for surface immunoprecipitation (IP). Surface IP of eGFP-labeled target anti-apoptotic proteins from transfected HEK293T crude cell extracts. Crude cell extracts with 1 nM of eGFP-labeled anti-apoptotic protein were used to IP. (**a**) BCL2 IP antibody, (**b**) BCLxL IP antibody, (**c**) MCL1 IP antibody (*n* = 10 independent images). **d-f**, Selection of monoclonal antibodies for total level immunoassay of surface-immobilized target anti-apoptotic proteins. (**d**) BCL2 detection antibody, (**e**) BCLxL detection antibody, (**f**) MCL1 detection antibody (*n* = 10 independent images). **g**, Relative model BCL2-BIM$_{BH3}$ complex from BCL2-mCherry/BIM$_{BH3}$-eGFP co-transfected HEK293T cells after the lysis with

different detergent types (*n* = 10 independent images). **h**, Detection of surface-immobilized BAK fragments by using the monoclonal antibody AT38E2. Crude cell extracts with 1 nM of eGFP-labeled BAK fragments were used to IP (*n* = 10 independent images). The data were normalized to the BAK level of 24-69 (α1) fragments. **i**, Activation of BAK from HL60 and Ramos cells by heat or triton X-100 treatment to crude cell extracts. The data were normalized to the active BAK level of non-treated HL60. **j**, Schematics for the preparation of HL60 samples with different preservation conditions. **k,l**, Comparison of the PPI profiles across the sample states using SMPC platform. (**k**) BCL2 total level, (**l**) BCL2-BAX complex (*n* = 10 independent images). Error bars represent means±s.d.

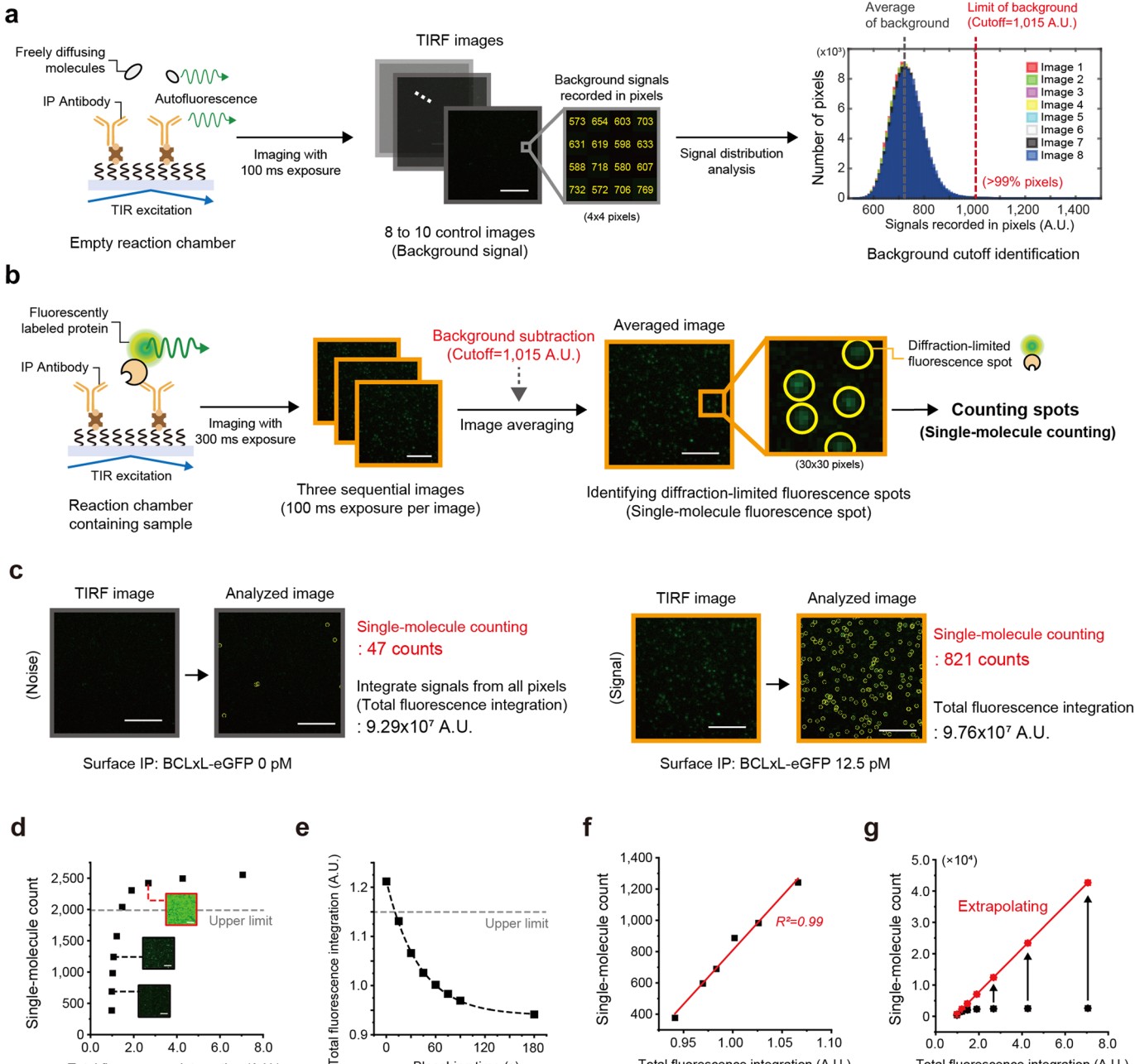

**Extended Data Fig. 2 | Post-image analysis for counting the number of single-molecule fluorescence spots. a**, Setting the cutoff to eliminate background signals for post image analysis. We used 8 to 10 control images captured with a 100 ms time resolution from empty reaction chambers to establish a background limit. The background limit (cutoff=1,015 A.U.) was determined from the signal distribution recorded in each pixel of the control images, effectively eliminating 99% of background signals. **b**, Identification of diffraction-limited fluorescence spots. Three sequential images captured with a 300 ms time frame were averaged to create a single image upon subtracting the background. From the averaged image, the number of diffraction-limited fluorescence spots were counted (single-molecule counting). **c**, Identified single-molecule counts and integration of individual pixel signals of the imaging area (total fluorescence integration)

from the analyzed TIRF images. The TIRF images were obtained by surface IP of BCLxL-eGFP expressed in HEK293T cell extracts (Scale bar: 10 μm) (**a-c**). **d-g**, Calibration of single-molecule count from the total fluorescence integration. (**d**) Single-molecule count dependent on the total fluorescence integration of surface-immobilized BCL2-mCherry obtained from the analyzed images (Scale bar: 10 μm). (**e**) Photobleaching kinetics of surface-immobilized BCL2-mCherry signals depending on the bleaching time. (**f**) The linear correlation between single-molecule count and total fluorescence integration of surface-immobilized BCL2-mCherry after photobleaching. (**g**) Calibration of single-molecule count from total fluorescence integration by extrapolating. The linear correlation was obtained in (**f**). The TIRF images were obtained by surface IP of BCL2-mCherry expressed in HEK293T cell extracts (**d-g**).

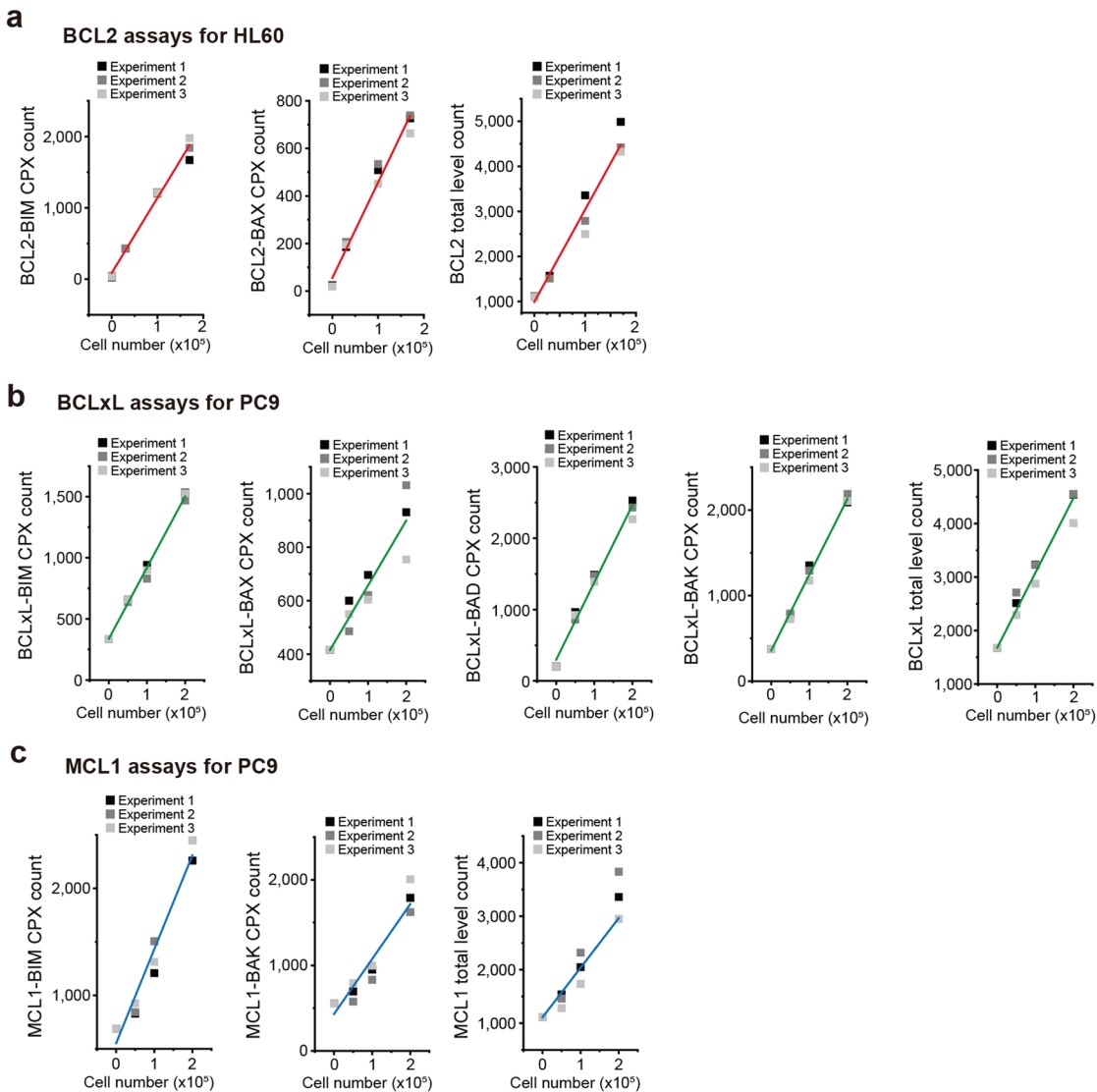

**Extended Data Fig. 3 | Attesting the stability of the SMPC platform by inter-chip measurements of immunoassays. a**, Counts of BCL2-related immunoassays (BCL2-BIM complex, BCL2-BAX complex, BCL2 total level) from the fixed numbers of HL60 cells ($n = 3$). **b**, Counts of BCLxL-related immunoassays (BCLxL-BIM complex, BCLxL-BAX complex, BCLxL-BAD complex, BCLxL-BAK complex, BCLxL total level) from the fixed numbers of PC9 cells ($n = 3$). **c**, Counts of MCL1-related immunoassays (MCL1-BIM complex,

MCL1-BAK complex, MCL1 total level) from the fixed numbers of PC9 cells ($n = 3$). The single-molecule counts were rescaled to account for the labeling efficiencies of the immunoassays calculated in Extended Data Fig. 5 as well as the specific incubation conditions for direct comparison. All data were measured from independent inter-chip experiments. CVs obtained from independent inter-chip measurement for all the immunoassays and cell numbers ($n = 3$). Individual data points shown for independent biological replicates.

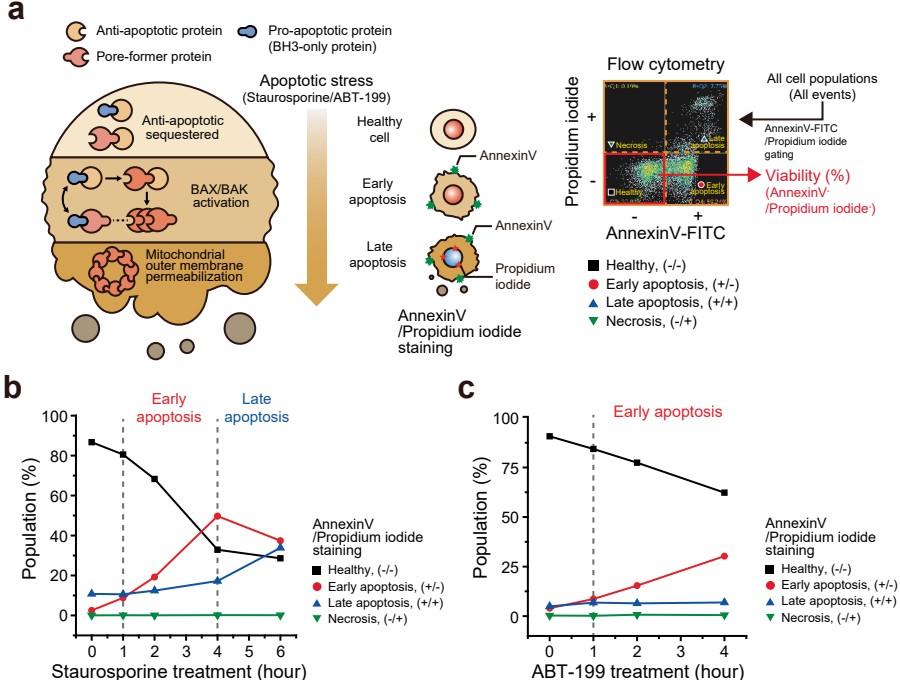

**Extended Data Fig. 4 | Tracking the response of HL60 cells under apoptotic stress. a**, Schematics for the intracellular PPI profile changes (left), and the result of flow cytometry analysis after Annexin V/Propidium iodide staining through the progression of apoptosis pathway (right). Viability represents the double negative populations after Annexin V/Propidium iodide staining and analyzed with flow cytometry. **b,c**, Population changes of HL60 cells at different time points through the treatment of drug with Annexin V/Propidium iodide staining. (**b**) 2 μM of staurosporine, (**c**) 300 nM of ABT-199.

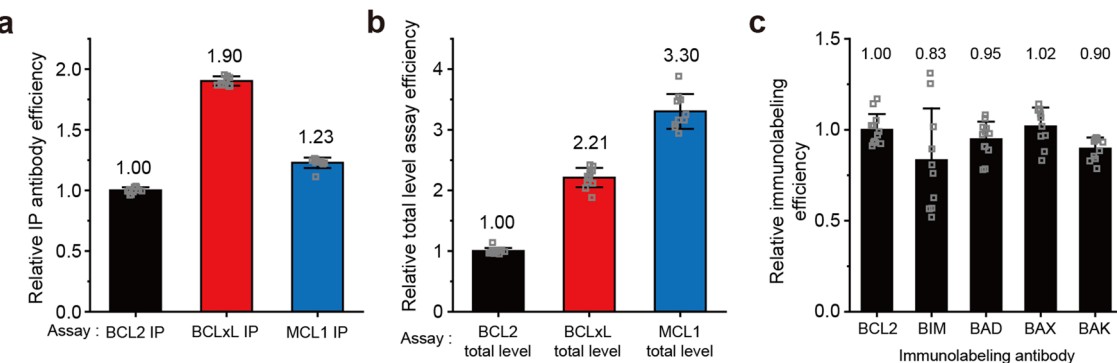

**Extended Data Fig. 5 | Relatively comparing the normalized efficiencies of different immunoassays. a**, Comparing relative efficiencies of IP antibodies for anti-apoptotic proteins. 1 nM of eGFP-labeled anti-apoptotic proteins were immobilized on the surface by the matched IP antibodies respectively, and the eGFP signals were compared ($n$ = 10 independent images). **b**, Comparing relative efficiencies of immunolabeling antibodies for anti-apoptotic proteins. The surface immobilized anti-apoptotic protein in (**a**) were immunolabeled by the matched labeling antibodies respectively, and the relative efficiencies for total level assays were compared ($n$ = 10 independent images). **c**, Comparing relative immunolabeling efficiencies of detection antibodies for BCL2 family proteins. 1 nM of eGFP-labeled BCL2 family proteins were immobilized on the surface by the anti-eGFP antibody and detected with matched detection antibodies for immunolabeling. The relative immunolabeling efficiencies of the detecting antibodies were calculated based on the ratio of labeled proteins to the total number of proteins immobilized on the surface ($n$ = 10 independent images). The data were normalized to the efficiencies of BCL2 IP assay, BCL2 total level assay, and BCL2 immunolabeling antibody, respectively (**a-c**). Error bars represent means±s.d.

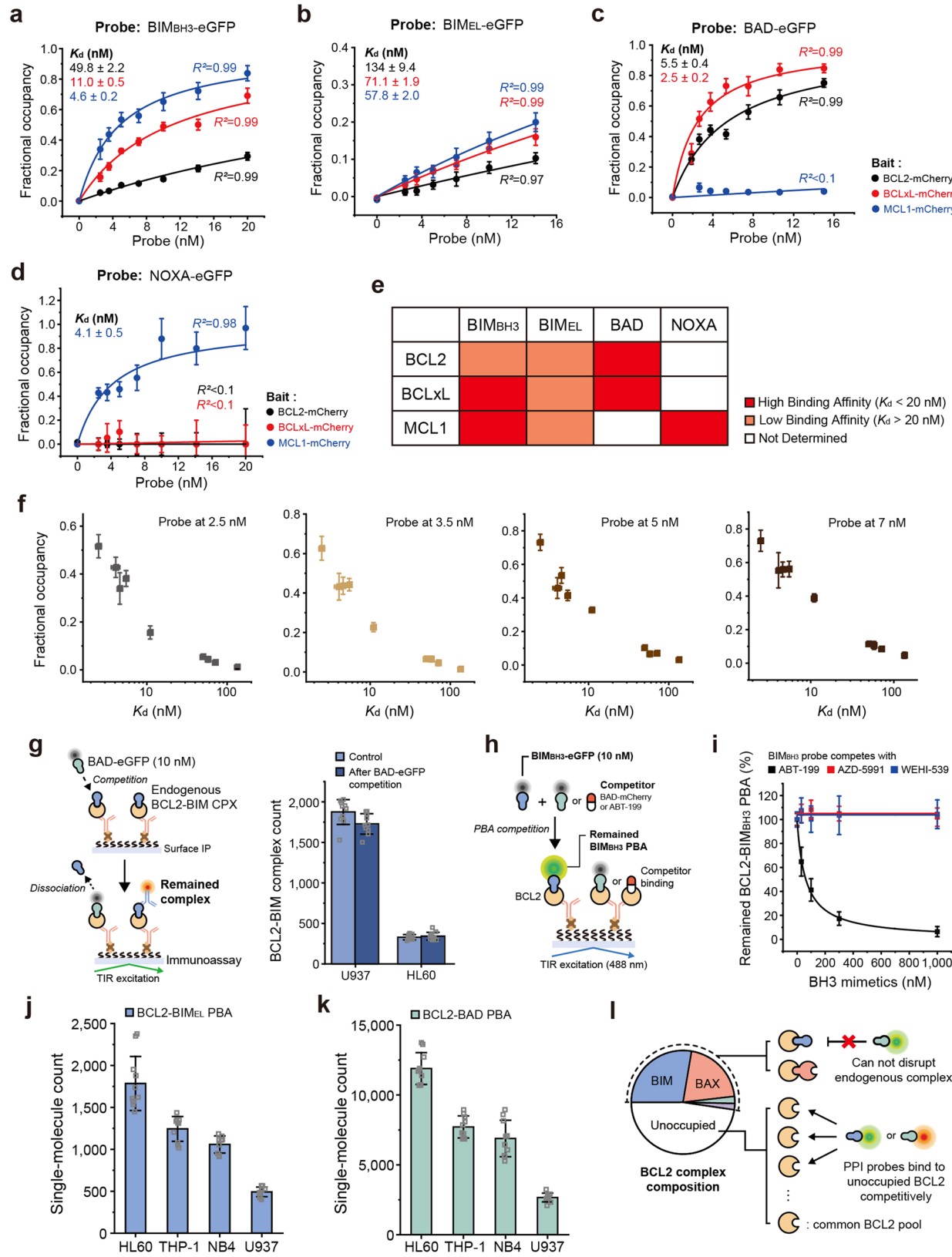

**Extended Data Fig. 6 | See next page for caption.**

**Extended Data Fig. 6 | Population profile of PPI probes binding onto unoccupied BCL2 and the cartoon schematic of the experiment. a-d**, PBA binding curves for each PPI probes (eGFP-labeled BIM$_{BH3}$, BIM$_{EL}$, BAD, NOXA) to anti-apoptotic proteins (mCherry-labeled BCL2, BCLxL, MCL1) to calculate dissociation constant ($K_d$). (**a**) BIM$_{BH3}$-PBA, (**b**) BIM$_{EL}$-PBA, (**c**) BAD-PBA, (**d**) NOXA-PBA. **e**, Comparison of the binding affinities between each binding pairs. **f**, Correlations between the calculated $K_d$ values and the occupancy values at a fixed PPI probe concentration. **g**, Dissociation of BCL2-BIM complex after *in vitro* competition by BAD-eGFP PPI probes ($n = 10$ independent images).

Protein complexes were surface-immobilized by anti-BCL2 IP antibody and detected with anti-BIM detection antibody after the competition. **h**, Schematic of *in vitro* PBA competition between BIM$_{BH3}$-eGFP PPI probe and competitors (PPI probe or BH3 mimetics) on surface-immobilized BCL2 proteins. **i**, Remained BCL2-BIM$_{BH3}$ PBA after *in vitro* binding competition with BH3 mimetics (ABT-199, AZD-5991, WEHI-539). BIM$_{BH3}$ PPI probe was presented in 10 nM. **j,k**, BCL2 PBA counts with different PPI probes from four AML cell lines. (**j**) BCL2-BIM$_{EL}$ PBA, (**k**) BCL2-BAD PBA ($n = 10$ independent images). **l**, Schematic for selective binding of PPI probes for unoccupied BCL2 proteins. Error bars represent means±s.d.

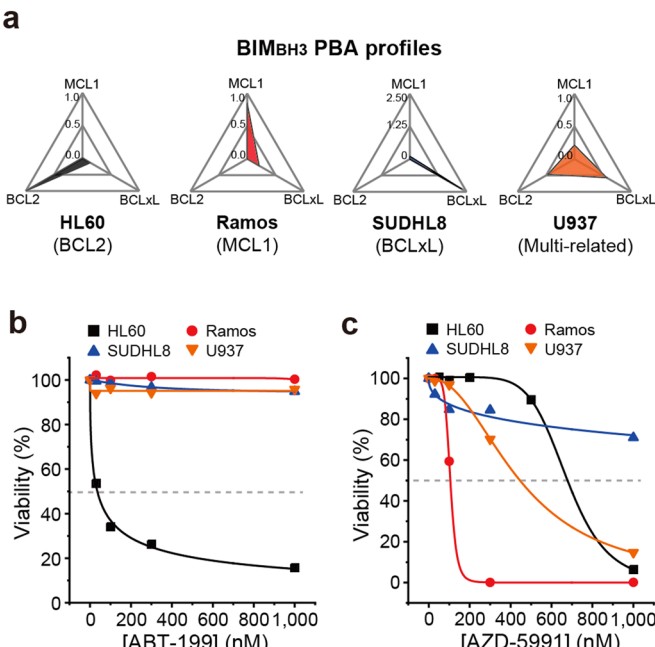

**Extended Data Fig. 7 | BIM$_{BH3}$ PBA profile predicts response of single-agent BH3 mimetics on leukaemia cell lines. a**, BIM$_{BH3}$ PBA profiles of four different leukemia cell lines. The BIM$_{BH3}$ PBA levels were rescaled to account for the relative PBAs of BIM$_{BH3}$ probe for anti-apoptotic proteins determined in Extended Data Fig. 6a. The data were normalized to the BCL2-BIM$_{BH3}$ PBA level of HL60 cell. **b,c**, Viability of leukemia cell lines after treatment of BH3 mimetics for 24 hours. (**b**) ABT-199, (**c**) AZD-5991.

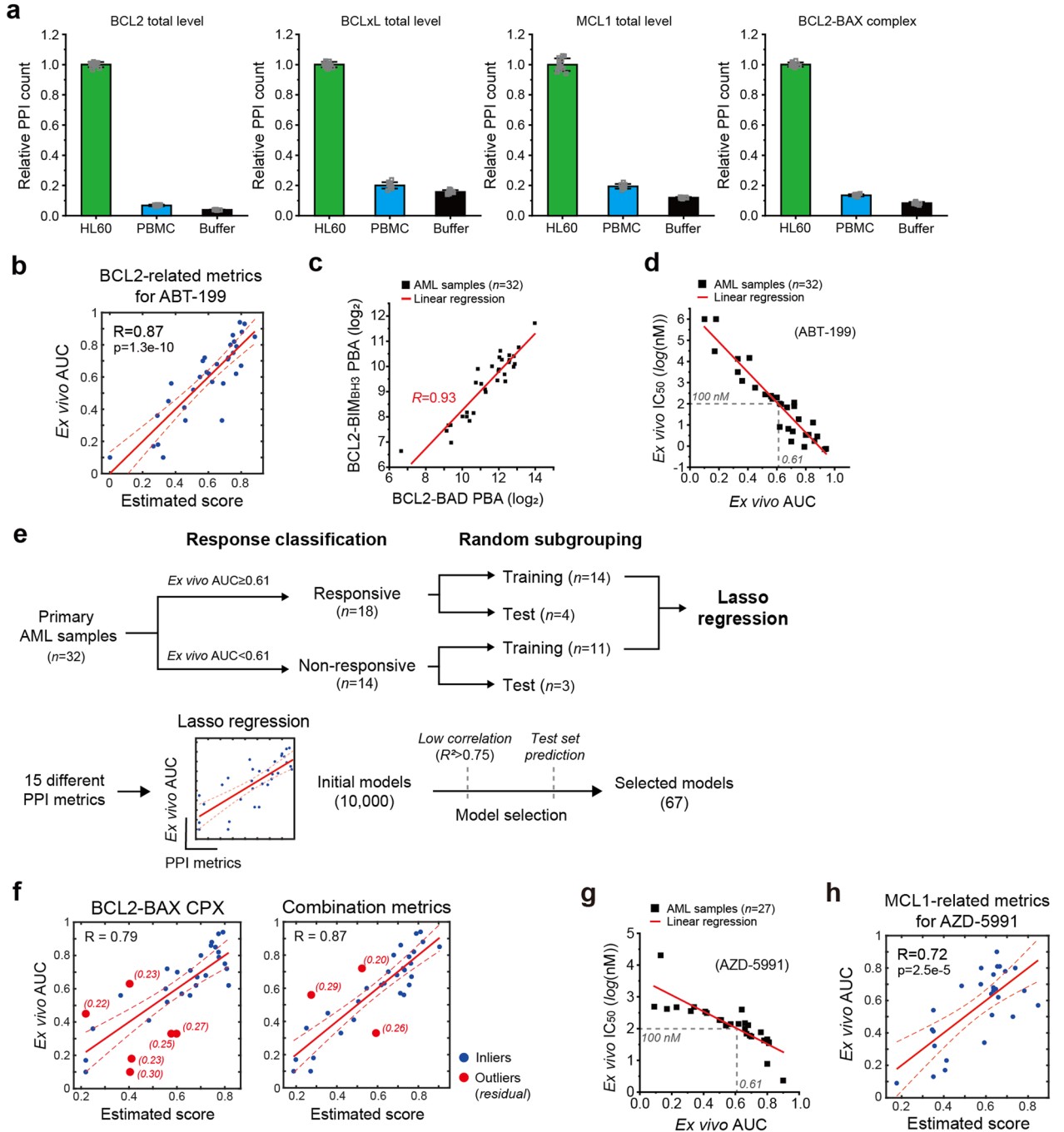

**Extended Data Fig. 8 | Development of the *ex vivo* drug-efficacy prediction model by using combinations of multiple PPI metrics. a**, Comparison of BCL2 family PPI profiles from the healthy donor PBMC sample and HL60 cells with SMPC ($n = 10$ independent images). The data were normalized to the HL60 cell. **b**, Correlation between *ex vivo* AUC for ABT-199 and combination of BCL2-related metrics (BCL2 total level, BCL2-BIM$_{BH3}$ PBA, BCL2-BAD PBA, and BCL2-BAX CPX) (One-sided *F*-test, *p*-value = 1.3e-10). **c**, Linear correlations between two different BCL2 PBA metrics. **d**, Correlations between *ex vivo* AUC and IC$_{50}$ of primary AML samples for ABT-199 ($n = 32$). **e**, Schematic of Lasso regression analysis for the selection of PPI metrics highly correlated with drug response. The training and

test groups were randomly selected from the primary AML sample cohort. The Lasso regression models were generated using the training group and evaluated based on Pearson's R as well as prediction outcomes for the test group. 67 models were selected from 10,000 different initial models. **f**, Identification of outliers in the ABT-199 drug efficacy prediction models. The residuals of each outlier in the model (*ex vivo* AUC – estimated score) were indicated. **g**, Correlations between *ex vivo* AUC and IC$_{50}$ of primary AML samples for AZD-5991 ($n = 27$). **h**, Correlation between *ex vivo* AUC for AZD-5991 and combination of MCL1-related metrics (MCL1 total level, MCL1-BIM$_{BH3}$ PBA, MCL1-NOXA PBA, and MCL1-BAK CPX) (One-sided *F*-test, *p*-value = 2.5e-5).

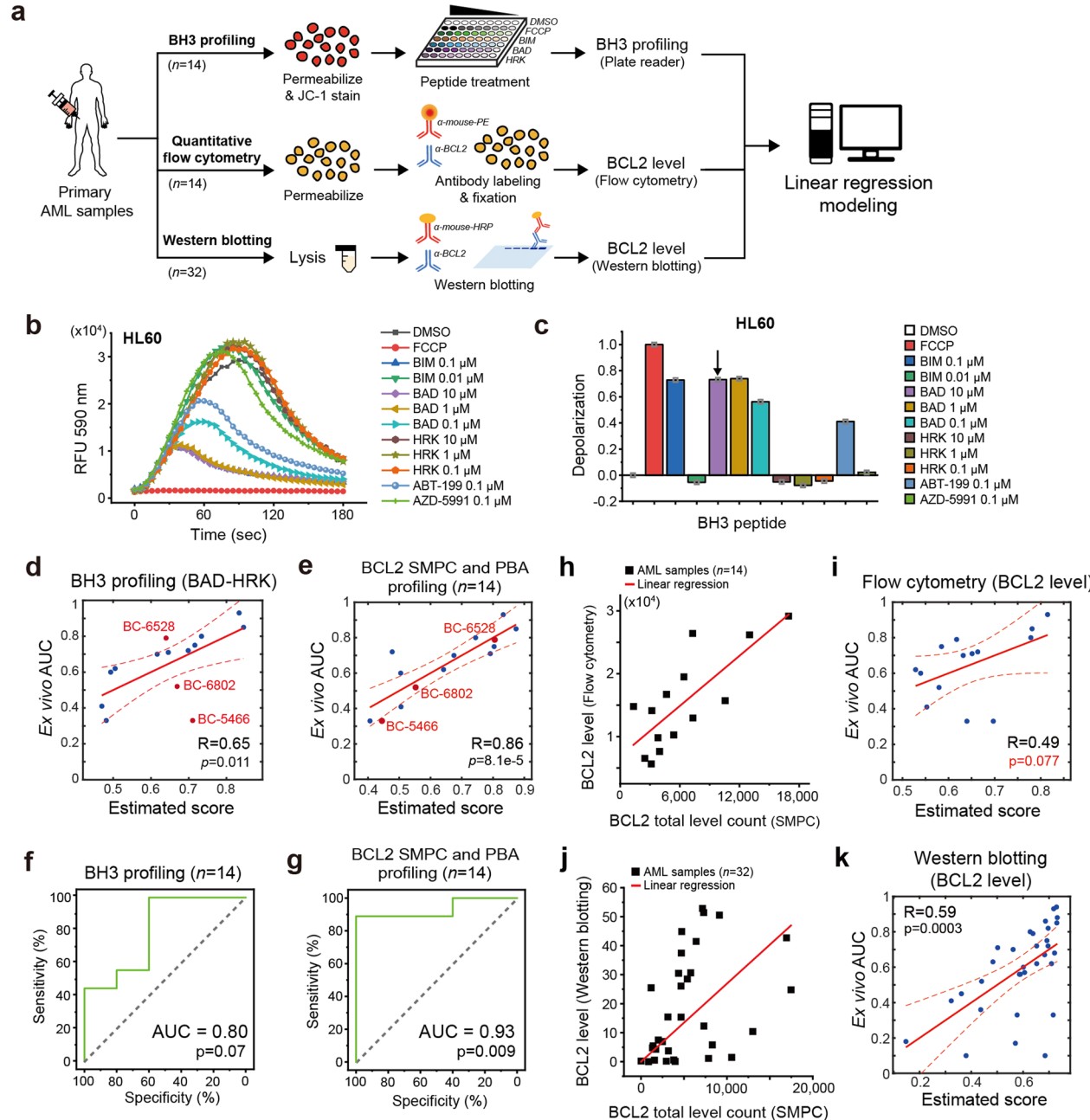

**Extended Data Fig. 9 | Comparison of BCL2 SMPC and PBA PPI profiling and different methods to predict responses to BH3 mimetics. a**, Schematic for BH3 profiling based on JC-1 staining, and measurement of BCL2 levels using flow cytometry or western blotting for primary AML samples. **b,c**, BH3 profiling on HL60 cells. (**b**) Relative fluorescence units (RFU) of 590 nm wavelength for HL60 cells through the treatment of BH3 peptides or BH3 mimetics, (**c**) Depolarization of HL60 cells through the BH3 profiling. Depolarization by 10 μM of BAD peptides was indicated. **d,e**, Correlations for *ex vivo* ABT-199 AUC with (**d**) depolarizations by BAD (10 μM) – HRK (10 μM) peptides, (**e**) combination of multiple PPI metrics (BCL2-BIM$_{BH3}$ PBA, BCL2-BAX CPX, and BCLxL-BAK CPX). The outliers identified in model (**d**) were indicated (*n* = 14)

(One-sided *F*-test, *p*-value = 0.011, 8.1e-05). **f,g**, ROC curves for *ex vivo* ABT-199 responses with (**f**) depolarizations by BAD (10 μM) – HRK (10 μM) peptides, (**g**) Combination of multiple PPI metrics (Two-sided *t*-test, *p*-value = 0.07, 0.009). **h**, Correlations between BCL2 protein levels determined by SMPC and flow cytometry for primary AML samples (n = 14). **i**, Prediction models for ABT-199 *ex vivo* AUC with BCL2 protein levels determined by flow cytometry (One-sided *F*-test, *p*-value = 0.077). **j**, Correlations between BCL2 protein levels determined by SMPC and western blotting for primary AML samples (n = 32). The raw blot image is provided as a Source Data. **k**, Prediction models for ABT-199 *ex vivo* AUC with BCL2 protein levels determined by western blotting (One-sided *F*-test, *p*-value = 0.0003).

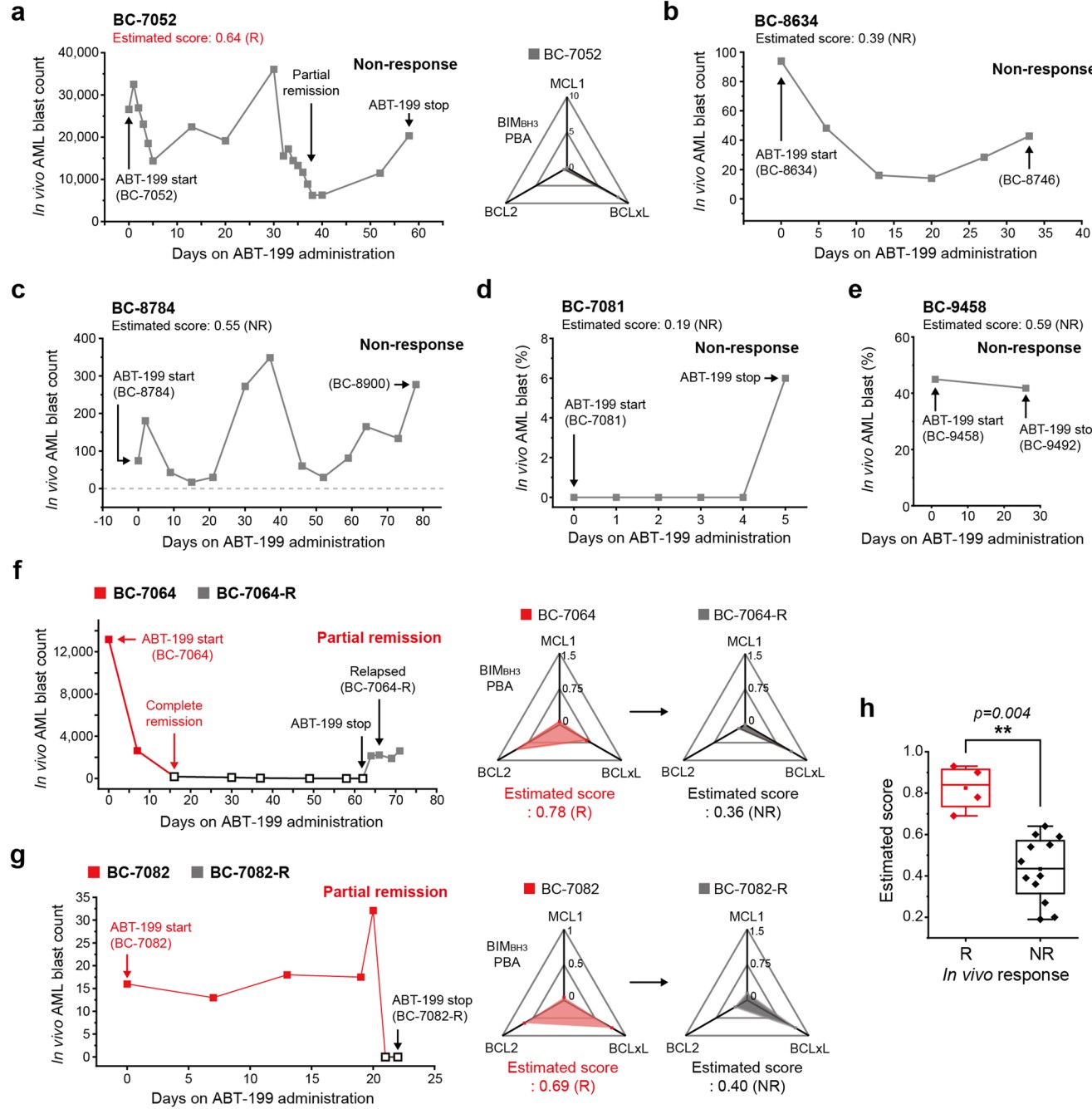

**Extended Data Fig. 10 | Longitudinal tracking of *in vivo* ABT-199 responses in patients with AML. a-g,** Changes of *in vivo* AML blast counts or AML blast ratio through the days after ABT-199 administration. The estimated scores and the PPI diagnostic results (R: Responsive, NR: Non-responsive) were obtained from the model in Fig. 4f. (**a**) BC-7052, (**b**) BC-8634, (**c**) BC-8784, (**d**) BC-7081, (**e**) BC-9458, (**f**) BC-7064, (**g**) BC-7082. The BIM$_{BH3}$ PBA profiles of each sample were presented, and The data were normalized to the BCL2-BIM$_{BH3}$ PBA level of HL60 cell. **h,** Comparison of the estimated scores of AML samples between responsive

(R, $n = 4$) and non-responsive (NR, $n = 10$) samples for *in vivo* ABT-199 administration (Two-sided Mann-Whitney test, *p*-value = 0.004). Six additional samples from the same cohort collected after relapse (BC-7064-R, BC-7082-R, BC-7107-R2, BC-8900, BC-9458, BC-9492) were included, which were categorized as the NR. The centre line represents the median, the box limits are upper and lower quartiles, and the whiskers represent 1.5x interquartile range for the box plots.

# Reporting Summary

## Statistics

For all statistical analyses, confirm that the following items are present in the figure legend, table legend, main text, or Methods section.

| n/a | Confirmed | |
|---|---|---|
| ☐ | ☒ | The exact sample size (*n*) for each experimental group/condition, given as a discrete number and unit of measurement |
| ☐ | ☒ | A statement on whether measurements were taken from distinct samples or whether the same sample was measured repeatedly |
| ☐ | ☒ | The statistical test(s) used AND whether they are one- or two-sided *Only common tests should be described solely by name; describe more complex techniques in the Methods section.* |
| ☒ | ☐ | A description of all covariates tested |
| ☐ | ☒ | A description of any assumptions or corrections, such as tests of normality and adjustment for multiple comparisons |
| ☐ | ☒ | A full description of the statistical parameters including central tendency (e.g. means) or other basic estimates (e.g. regression coefficient) AND variation (e.g. standard deviation) or associated estimates of uncertainty (e.g. confidence intervals) |
| ☐ | ☒ | For null hypothesis testing, the test statistic (e.g. *F*, *t*, *r*) with confidence intervals, effect sizes, degrees of freedom and *P* value noted *Give P values as exact values whenever suitable.* |
| ☒ | ☐ | For Bayesian analysis, information on the choice of priors and Markov chain Monte Carlo settings |
| ☒ | ☐ | For hierarchical and complex designs, identification of the appropriate level for tests and full reporting of outcomes |
| ☐ | ☒ | Estimates of effect sizes (e.g. Cohen's *d*, Pearson's *r*), indicating how they were calculated |

*Our web collection on statistics for biologists contains articles on many of the points above.*

## Software and code

Policy information about availability of computer code

| | |
|---|---|
| Data collection | Single-molecule fluorescence signals were collected using a PI-View (PROTEINA) and assessed using the PI-Analyzer (v1) provided by the manufacturer. The flow-cytometry data were collected using SH800S (Sony) and the SH800S Cell Sorter Software (v2.1.5) provided by the manufacturer. The Western blotting data were collected using ImageQuant LAS 4000 mini (Cytiva) and the ImageQuant TL software (v10.2) provided by the manufacturer. |
| Data analysis | The PI-Analyzer software (v1) by PROTEINA was used for post-image analysis of the single-molecule fluorescence images. The SH800S Cell Sorter Software (v2.1.5) by Sony was used for flow-cytometry analysis. ImageJ (v1.53a) was used for Western blotting to measure the protein-band intensity. OriginPro (v2022) was used for fitting curves and statistics. Custom graphical user interface (GUI) software written in MATLAB (2021a) and Python (3.11) were used for the analysis of the drug-efficacy prediction model. The custom MATLAB and Python codes are available on GitHub at https://github.com/tyyoonlab-snu/Nat-Biomed-Eng-2023- |

For manuscripts utilizing custom algorithms or software that are central to the research but not yet described in published literature, software must be made available to editors and reviewers. We strongly encourage code deposition in a community repository (e.g. GitHub). See the Nature Portfolio guidelines for submitting code & software for further information.

## Data

Policy information about availability of data

All manuscripts must include a data availability statement. This statement should provide the following information, where applicable:

- Accession codes, unique identifiers, or web links for publicly available datasets
- A description of any restrictions on data availability
- For clinical datasets or third party data, please ensure that the statement adheres to our policy

> The data supporting the findings of this study are available within the article and its Supplementary Information. All raw data generated or analysed during the study are available on GitHub at https://github.com/tyyoonlab-snu/Nat-Biomed-Eng-2023-

## Research involving human participants, their data, or biological material

Policy information about studies with human participants or human data. See also policy information about sex, gender (identity/presentation), and sexual orientation and race, ethnicity and racism.

| | |
|---|---|
| Reporting on sex and gender | All of the patients underwent chromosomal analysis owing to underlying acute myeloid leukemia (AML). Sex was determined on the basis of chromosomal analysis, but sex was not considered in the study design because there is no evidence that ABT-199 responses depend on sex. |
| Reporting on race, ethnicity, or other socially relevant groupings | 35 Korean patients with acute myeloid leukemia undergoing ABT-199 treatment, regardless of previous medical history. All participants were of Asian race and ethnicity, born in Korea. But ethnicity and other socially relevant groupings were not considered in the study design because there is no evidence that ABT-199 responses depend on those groupings. |
| Population characteristics | 35 Korean patients with acute myeloid leukemia undergoing ABT-199 treatment at Seoul National University Hospital were enrolled, regardless of previous medical history. Samples were collected between January 2014 and December 2019. Acute promyelocytic leukemias and biphenotypic leukemias were excluded. |
| Recruitment | All patients undergoing ABT-199 treatment were asked to participate, and upon agreement their bone-marrow and peripheral-blood samples were collected and their medical history recorded. There were no potential biases. |
| Ethics oversight | The study was conducted according to the Declaration of Helsinki and was approved by the institutional review board (IRB) at Seoul National University Hospital (IRB number: H-1910-176-107). Informed consent was taken from all patients before participating in any study-related procedure. No compensation was given for their participation. |

Note that full information on the approval of the study protocol must also be provided in the manuscript.

# Field-specific reporting

Please select the one below that is the best fit for your research. If you are not sure, read the appropriate sections before making your selection.

☒ Life sciences ☐ Behavioural & social sciences ☐ Ecological, evolutionary & environmental sciences

For a reference copy of the document with all sections, see nature.com/documents/nr-reporting-summary-flat.pdf

# Life sciences study design

All studies must disclose on these points even when the disclosure is negative.

| | |
|---|---|
| Sample size | Generally, the single-molecule fluorescence images were obtained from 10 different locations within each chamber of the imaging chip to ensure its statistical repeatability, on the basis of a previous study (Lee, 2013; Yoo, 2016; Lee, 2018). The inter-chip coefficient of variance (CV) of single-molecule co-IP were calculated from 3 independent measurements of protein complexes to ensure statistical repeatability. |
| Data exclusions | To avoid any biased analysis, the images containing either photo-bleached region or aggregated large dye cluster were excluded after the post-image analysis of PI-View system. The exclusion criteria are described in Extended Data Fig. 2 and in the previous study (Lee, 2013; Yoo, 2016; Lee, 2018). |
| Replication | Experiments to perform further analysis (including single-molecule fluorescence imaging) were repeated independently at least 2 times, and replications were successful for all experiments. The SMPC for inter-chip CVs were repeated biologically in 3 independent times (Fig. 1 and Extended Data Fig. 3), and replications were successful for all experiments. |
| Randomization | For all single-molecule fluorescence imaging, each data point was collected from separate wells within the multi-well imaging chips on separate days. The primary AML samples were collected on independent dates and the PPI profiles were measured within the independent days. |
| Blinding | For the single-molecule fluorescence imaging, blinding was not performed because of the unbiased nature of the in vitro experiments performed in this study. |

For ex vivo drug-efficacy analysis, PPI profiling, BH3 profiling and quantitative flow-cytometry analysis of primary AML samples were performed and analysed in a blind manner.

# Reporting for specific materials, systems and methods

We require information from authors about some types of materials, experimental systems and methods used in many studies. Here, indicate whether each material, system or method listed is relevant to your study. If you are not sure if a list item applies to your research, read the appropriate section before selecting a response.

## Materials & experimental systems

| n/a | Involved in the study |
|-----|----------------------|
| ☐ | ☒ Antibodies |
| ☐ | ☒ Eukaryotic cell lines |
| ☒ | ☐ Palaeontology and archaeology |
| ☒ | ☐ Animals and other organisms |
| ☒ | ☐ Clinical data |
| ☒ | ☐ Dual use research of concern |
| ☒ | ☐ Plants |

## Methods

| n/a | Involved in the study |
|-----|----------------------|
| ☒ | ☐ ChIP-seq |
| ☐ | ☒ Flow cytometry |
| ☒ | ☐ MRI-based neuroimaging |

## Antibodies

| | |
|---|---|
| Antibodies used | Anti-rabbit immunoglobulin G (IgG) with biotin conjugation (111-065-144; Jackson ImmunoResearch, 1:200) and anti-mouse IgG with biotin conjugation (715-066-151; Jackson ImmunoResearch, 1:200) were used to immobilize the antibodies for surface IP. Anti-RFP antibody with biotin conjugation (ab34771; Abcam, 1:200) and anti-GFP antibody with biotin conjugation (ab6658; Abcam, 1:200) were used to immobilize mCherry- or eGFP-labeled proteins on surface. Anti-MCL1 (94296S; Cell Signaling Technology, D2W9E, 1:100), BCLxL (MA5-15142; Thermo Fisher Scientific, C.85.1, 1:100), BCL2 (4223S; Cell Signaling Technology, D55G8, 1:100), BAX (5023S; Cell Signaling Technology, D2E11, 1:100), and BAK (ab32371; Abcam, Y164, 1:100) antibodies were used to immobilize the corresponding proteins via surface pull-down. Anti-MCL1 (MAB8825; Abnova, C9, 1:100), BCLxL (NBP1-47665; Novus Biologicals. OTI4A9, 1:100), BCL2 (sc-7382; Santa Cruz Biotechnology, C2, 1:100), BAX (MABC1176M; Sigma Aldrich, 6A7, 1:100), BAK (sc-517390; Santa Cruz Biotechnology, AT38E2, 1:100), BIM (sc-374358; Santa Cruz Biotechnology, H5, 1:100), BAD (sc-8044; Santa Cruz Biotechnology, C7, 1:100), NOXA (sc-56169; Santa Cruz Biotechnology, 114C307, 1:100) antibodies were used to detect the corresponding proteins in the protein total level and the protein complex assays. To measure BCL2-BAX complex level, anti-BAX (5023S; Cell Signaling Technology, D2E11, 1:100) and anti-BCL2 (BMS1028; Invitrogen, Bcl-2/100, 1:100) antibodies were used to immobilize and detect the BCL2-BAX complex, respectively. Anti-rabbit IgG with Cy3 conjugation (111-165-046; Jackson ImmunoResearch, 1:1,000) and anti-mouse IgG with Cy3 conjugation (715-165-151; Jackson ImmunoResearch, 1:1,000) were used to label the detection antibodies for immunoassay. Anti-BCL2 antibody (BMS1028; Invitrogen, Bcl-2/100, 0.5 µg), mouse IgG1 kappa isotype control (554121; BD Biosciences, 0.5 µg), and anti-mouse IgG with PE conjugation (715-116-150; Jackson ImmunoResearch, 1.5 µl) were used to detect the protein levels for quantitative flow cytometry. Anti-BCL2 antibody (sc-7382; Santa Cruz Biotechnology, C2, 1:1,000) and HRP-linked anti-mouse IgG (7076S; Cell Signaling Technology, 1:1,000) were used to detect the protein levels for Western blotting. |
| Validation | All antibodies are commercially validated and published previously for their corresponding applications for specific species. All antibodies were validated by immunoassay using the SMPC within our lab described in Extended Data Fig. 1, Extended Data Fig. 3 and Extended Data Fig. 5. All informations about antibodies can be found in the manufacturer's guidelines. |

## Eukaryotic cell lines

Policy information about cell lines and Sex and Gender in Research

| | |
|---|---|
| Cell line source(s) | HL60, THP-1, U937 and Ramos cells were purchased from Korean Cell Line Bank. HEK293T and SU-DHL8 cells were purchased from ATCC. NB4 cells were purchased from DSMZ. PC9 cells were provided by Y. Hayata (Kyushu University Faculty of Medicine, Japan). PC9 cells are not currently commercially available. |
| Authentication | The cell line was authenticated at source, and regularly validated by morphological analysis. |
| Mycoplasma contamination | The cell line was confirmed to be mycoplasma-negative. |
| Commonly misidentified lines (See ICLAC register) | No commonly misidentified cell lines were used. |

# Flow Cytometry

## Plots

Confirm that:

☒ The axis labels state the marker and fluorochrome used (e.g. CD4-FITC).

☒ The axis scales are clearly visible. Include numbers along axes only for bottom left plot of group (a 'group' is an analysis of identical markers).

☒ All plots are contour plots with outliers or pseudocolor plots.

☒ A numerical value for number of cells or percentage (with statistics) is provided.

## Methodology

Sample preparation

Viability analysis: All cells were rinsed with cold DPBS and collected by centrifugation. After the supernatants were discarded, the cell pellets were resuspended with cold AnnexinV-binding buffer (Biolegend). After resuspension, AnnexinV-FITC solution (Biolegend) and PI solution (Biolegend) were added into the cell suspensions. The cell suspensions were incubated for 15 minutes at room temperature avoiding exposure to light. After that, 400 µl of cold AnnexinV-binding buffer was added to avoid overstaining.

Quantitative flow cytometry: All cells were rinsed with warm DPBS and collected by centrifugation. After the supernatants were discarded, the cell pellets were resuspended with staining wash buffer. After resuspension, the cells were permeabilized by PW buffer (BD Bioscience) and stained by adding anti-BCL2 antibody or an isotype control antibody. The cell suspensions were incubated for 30 minutes at room temperature avoiding exposure to light. After that, the cell suspensions were rinsed with PW buffer and stained by adding PE-conjugated anti-Mouse IgG with 30 minutes incubation. The stained cells were rinsed with PW buffer and fixed by Fix buffer (BD Bioscience).

Instrument

Flow cytometry was performed on SH800S (Sony).

Software

Analyses were performed using dedicated software for SH800S (v2.1.5, Sony).

Cell population abundance

The purity of the analysed primary samples was confirmed by flow cytometry.

Gating strategy

Viability analysis: The proportions of negatively stained cells for both AnnexinV and PI from each primary sample were calculated and converted to viability (%) (see Extended Data Fig. 4a).

Quantitative flow cytometry: Quantum R-PE MESF beads (Bangs Laboratories) were used to produce calibration curves. These curves were subsequently employed, following the manufacturer's instructions, to quantify the levels of BCL2 in the samples (Smith, 2019).

☒ Tick this box to confirm that a figure exemplifying the gating strategy is provided in the Supplementary Information.

