## [Peer review file · Nature Biomedical Engineering]

Profiling protein–protein interactions to predict the efficacy of B-cell-lymphoma-2-homology-3 mimetics for acute myeloid leukaemia

Corresponding author: Tae-Young Yoon

Editorial note

This document includes relevant written communications between the manuscript's corresponding author and the editor and reviewers of the manuscript during peer review. It includes decision letters relaying any editorial points and peer-review reports, and the authors' replies to these (under 'Rebuttal' headings). The editorial decisions are signed by the manuscript's handling editor, yet the editorial team and ultimately the journal's Chief Editor share responsibility for all decisions.

Any relevant documents attached to the decision letters are referred to as **Appendix #**, and can be found appended to this document. Any information deemed confidential has been redacted or removed. Earlier versions of the manuscript are not published, yet the originally submitted version may be available as a preprint. Because of editorial edits and changes during peer review, the published title of the paper and the title mentioned in below correspondence may differ.

Correspondence

Fri 14 Apr 2023

Decision on Article nBME-23-0122

Dear Prof. Yoon,

Thank you again for submitting to *Nature Biomedical Engineering* your manuscript, "Predicting the efficacy of BH3 mimetics through the profiling of multiple protein complexes". The manuscript has been seen by 5 experts, whose reports you will find at the end of this message.

You will see that the reviewers appreciate the work. However, they express concerns about the degree of support for the claims, and provide useful suggestions for improvement. We hope that with significant further work you can address the criticisms and convince the reviewers of the merits of the study. In particular, we would expect that a revised version of the manuscript provides:

- * Expanded evidence of the prediction outcomes of the single-molecule pull-down and co-immunoprecipitation method in comparison with other methods for obtaining the efficacies of BH3 mimetics, and use of more established metrics (such as receiver operating characteristic curves), ideally in a larger cohort of patients.
- * Thorough characterization of the binding affinities of the probes, as per the queries of Reviewer #5.
- * Extended discussion of the applicability of the clinical applicability of the method, mentioning any necessary caveats and implementation bottlenecks.
- * Thorough reporting of the experimental methods and reagents used. Please clearly describe all the probes and antibodies used, and use consistent nomenclature.

When you are ready to resubmit your manuscript, please upload the revised files, a point-by-point rebuttal tothe comments from all reviewers, the reporting summary, and a cover letter that explains the main improvements included in the revision and responds to any points highlighted in this decision.

Please follow the following recommendations:

- * Clearly highlight any amendments to the text and figures to help the reviewers and editors find and understand the changes (yet keep in mind that excessive marking can hinder readability).
- * If you and your co-authors disagree with a criticism, provide the arguments to the reviewer (optionally, indicate the relevant points in the cover letter).
- * If a criticism or suggestion is not addressed, please indicate so in the rebuttal to the reviewer comments and explain the reason(s).
- * Consider including responses to any criticisms raised by more than one reviewer at the beginning of the rebuttal, in a section addressed to all reviewers.
- * The rebuttal should include the reviewer comments in point-by-point format (please note that we provide all reviewers will the reports as they appear at the end of this message).
- * Provide the rebuttal to the reviewer comments and the cover letter as separate files.

We hope that you will be able to resubmit the manuscript within 20 weeks from the receipt of this message. If this is the case, you will be protected against potential scooping. Otherwise, we will be happy to consider a revised manuscript as long as the significance of the work is not compromised by work published elsewhere or accepted for publication at *Nature Biomedical Engineering*.

We hope that you will find the referee reports helpful when revising the work. Please do not hesitate to contact me should you have any questions.

Best wishes,

Filipe

Dr Filipe Almeida
Associate Editor, Nature Biomedical Engineering

Reviewer #1 (Report for the authors (Required)):

Yoon et al described that by the single-molecule pull-down and co-IP, more than 20 different types of PPI complexes were quantified upon ABT-199 treatment. By comparing the obtained multi-dimensional data, they constructed an analysis model for evaluation of ABT-199 efficacy. This model was applied on acute myeloid leukemia patients. Predictive biomarkers for both initial responses and resistance to ABT-199 were obtained. This study revealed a capability for the extensive characterization of BCL2-family PPI network in clinical specimens. It is of great interest for the field of BH3 mimetics and antitumor field. However, there are some major concerns need to be addressed.

1. Fig4 c,d, and e, the score for Bcl-2/Bim complex is relative lower (0.43), while those for Bcl-2/Bax and the combination metrics are 0.79 and 0.87, respectively. If the authors believe the combination metrics is significantly more accurate than Bcl-2/Bax in the view of prediction, Please discuss it deeply. Especially, for clinical application does it deserve to measure and calculate such a complexed index instead of one pair of PPI?

2. Fig.3g, at 0 hour of ABT-199 treatment, Bax and Bak activation are much higher than those indicated in Fig 3b, which seemed rational since not any apoptosis stimuli there.

3. In Fig.3g and Fig.3h, the activated Bax is higher than total Bax at several time points. Comparably, in Extended Data Fig. 4b, Bcl-xl complex is higher than total Bcl-xl.

4. How did the authors measure the Bak activation?

5. Since an accurate quantification is one of the most advances for single-molecule pull-down, why a fold number is used in the right Y-axis in Fig.2L and Extended Data Fig. 4a rather than a single molecule count number ?

6. This reviewer suggests authors to provide details on the fitting of the drug efficacy prediction model, especially, the theoretical formula to calculate the estimated score.

Reviewer #2 (Report for the authors (Required)):

The manuscript by Chun et al. "Predicting the efficacy of BH3 mimetics through the profiling of multiple protein complexes" seeks to address current issues in BH3 mimetics treatment of hematological malignancies, and the necessity for the predictive biomarkers for both initial responses and resistance development.

BH3 mimetics are protein-protein interaction (PPI) inhibitors that saturate anti-apoptotic proteins in the BCL-2 family to induce apoptosis in cancer cells. BH3 mimetic ABT-199 (venetoclax) has been approved for the treatment of CLL and AML in the clinic. Despite the remarkable success of ABT-199 in treating of CLL and AML, a large portion of patients are not responsive to the treatment, and many patients with an initial favourable response develop a resistance to the treatment later.

The authors hypothesized that the responsiveness of individual cancers to the PPI inhibitor would be determined by protein composition of target protein complexes. They adopted and optimized a sensitive single-cell pull-down and co-IP platform to extensively profile BCL-2 family protein complexes in clinical samples obtained from acute myeloid leukemia patients and to compare the data with BH3 mimetic efficacies determined ex vivo. Obtained multidimensional data was used to construct the analytical model to assist in therapeutic decision-making for acute myeloid leukemia patients and to predict the treatment outcome. The authors successfully demonstrated that the developed highly sensitive BCL-2 PPI profiling platform and analysis model can be used to guide therapeutics decisions in a prospective manner.

The manuscript is very well written; clear and precise. The background state of research is well presented and authors' hypothesis is clearly supported. Experiments are well designed and the methodology is vast. A large amount of very solid work was involved in the study. The results are new and interesting, and are well discussed.

These results and the developed platform enabling sensitive and accurate profiling of multiple PPI complexes in clinical samples will definitely play an important role in development of future precision medicine strategies for optimized treatment for hematological malignancies. In addition, the finding and the methodologies have a great potential to be applied to other disease-causing protein complexes outside of BCL-2 family.

Minor points:

- Several figures are overloaded with the material, which makes them hard to read. The font size of some labels is very small. Authors are recommended to review and, possibly, reorganize Figures 1, 2, 4, 6 and Extended data Fig. 10. For example, the assay schematics (Fig. 2, panels e and k) could be moved to supplementary data, and some data tables under the graphs can be removed (Fig.4 c and d).
- The authors don't discuss if they applied the developed single-molecule pull-down and co-IP platform to quantify the PPI complexes of interest in the healthy control cell lines/cell material from the healthy donors, or to other hematological malignancies highly susceptible to BCL2 inhibition, such as chronic lymphocytic leukemia (CLL) or multiple myeloma. In future, such data will enrich the developed analytical model and enhance the predicting power of clinical treatment outcome.

Reviewer #3 (Report for the authors (Required)):

The authors used a single-molecule pull-down and co-immunoprecipitation (co-IP) platform to quantify multiple BH3 protein interactions. They aimed to predict sensitivity to BH3 mimetics (specifically venetoclax) with this approach. They correlate their score first with ex-vivo and finally with in-vivo response to venetoclax in a limited number of 10 patients.

The manuscript adds to a growing number of interesting techniques which allow the quantification of BH3 proteins for the use as a biomarker to predict treatment response.

The manuscript is systematically structured but individual paragraphs are sometimes difficult to understand. Specifically the final part reporting on the correlation with in-vivo response is very difficult to understand what has been done.

The authors claim that based on 6 out of 10 matches between in-vivo and ex-vivo response that their test system works. No statistical tests have been performed and conclusions made by the authors are not justified based on the presented data. This is by far the weakest part of the paper.

The paper shows a good correlation of the newly developed assay with ex-vivo response to venetoclax, which is encouraging. However, it is not illustrated why such an assay would be better ex-vivo drug response profiling of cancer cells as proposed by different consortia. Further the peptide-based BH3 profiling is a further well established assay which measures the same endpoint but to which the new assay is not compared to.

Reviewer #4 (Report for the authors (Required)):

Chun et al present a novel method for measuring protein- protein interactions via single molecule analysis via what is essentially immunofluorescent microscopy. They use this to predict responses in AML to Bh3 mimetics.

This is a very innovative and interesting paper. It is well organized and well-written. The biology of BH3 mimetic effect has always suggested that a complete measurement of all PPIs could be useful to predict and understand response. However, there has not been a good method for making the number of measurements required before this paper, at least not that I am aware of. They perform a technical tour de force, and do a great job explaining their biochemical optimization. They train two predictors, one for a BCL-2 inhibitor, one for an MCL-1 inhibitor. Overall, I think this is a fascinating paper with a technology that has many potential uses beyond the test case presented in this paper. I think that it will be of interest not only to those studying the BCL-2 family of proteins, but indeed to anyone interested in protein-protein interactions. While have several specific comments to follow, I would be very positive about this paper.

1) The overall approach - this is a great source of the novelty of the paper, and as such, we simply need to know more about it. I am sure that somewhere they have data regarding calibration of their microscopy to know that they are actually getting counts of single molecule data. It would be nice if they would refer to this if there is a published reference, or put it in the supplement if not.

2) It is not acceptable that the raw data and the code for the drug efficacy prediction model (lines 619-626) are available only "upon reasonable request". For this to be published, they must instead be available directly on line to anyone reading the paper. Moreover, the linear regressions and the specific coefficients used must also be available for immediate inspection, not just upon request.

3) If this type of assay were ever to be used more broadly, it would be important to know the impact of shipping and storing conditions. For instance, did the authors ever compare their PPI results between fresh and frozen samples, either with cell lines or primary AML samples?

4) In overexpression models in cell lines, using FLIM-FRET, the laboratory of David Andrews has examined PPI and how they are affected by BH3 mimetics. The authors may want to compare their results to these in the discussion.

5) The title and motivation of this paper is all about prediction. As such it is disappointing that we never see a quantitative study of how good a predictor they actually have here. The validation set is just a half dozen patient cases. I would suggest they need more cases, and then an objective measure of the performance of their predictor using a measure of binary prediction like a receiver operating characteristic (ROC).

6) Do I understand correctly that in Figure 4b, there are no negative correlations, only positive correlations? This is quite surprising. It is especially surprising as they apparently include the BCL-2/BAK CPX value in their predictor with a negative coefficient, despite it apparently positively correlating with sensitivity to ABT-199? This does not seem to make sense.

7) Figure 4, they show a correlation with BCL-2 levels and response. They may want to comment on previous attempts to correlate response to BCL-2 levels alone (mRNA or protein) that were apparently not as successful. Due to method of measurement?

More specific points by line in paper:

1 - Title – "In AML" would be appropriate since this is only about AML.

36 - mTOR = mammalian target of rapamycin

40 – Does not require interaction of BAX with BAK.

82 – BH3 profiling does not require the derivation or maintenance of Cancer-derived mitochondria.

402 – Not that much of a puzzle, really. Kinetics of in vivo and in vitro response already suggest that dissociation of pre-existing complexes as the answer.

Figure 6g is really not intelligible. What are the triangles showing? What are the lines at right showing? Why are high and low lines of equal length.

Reviewer #5 (Report for the authors (Required)):

This paper reports a lot of interesting observations and a model that the authors claim can be used prognostically to determine likely response of AML patients to ABT-199 (Venetoclax). They report using single molecule counting of PPI complexes changes in cell lines and patient samples in response to drugs including the clinically relevant drug ABT-199. Many of the experiments appear to be well performed and much of the data is convincing. The work reported is potentially both a mechanistic and technological advance with potential translational importance. However, there are a number of problems that make the manuscript very difficult to follow. In addition, there are multiple examples where the authors provide single interpretations of the data where other possibilities seem to be equally or more likely. Finally, the authors use terms in misleading ways that may be due to a language barrier but that further confound interpretation.

Major Technical criticisms or questions for which addressing is mandatory:

1. Fig. 1. The definition of a positive signal in the TIRF images is not clear. The two tiny images shown in panel f have very different background levels. The circled areas of images in Extended Data Fig. 3 does not make it clear how spots were selected. They report an average spot has 439 counts while the background pixels have values between 135 and 525 (I calculate an average of 352). Depending on the level of fluctuation it seems to me that the cut-off for identifying a spot is critical. Were spots identified automatically or manually? What is the actual definition of a spot? There is no information about local background correction, or other image processing. The entire project depends on appropriate identification of spots therefore this information is critical.

2. The authors examine Bim binding to BAX and BAK but there is nothing in the paper about Bid which is likely to be as important or more important than Bim given that Bim preferentially binds BAX while Bid binds both BAX and BAK similarly. This needs to be addressed either experimentally or as a caveat in the discussion.

3. I looked up the antibodies to BAX and BAK. In only one place is the BAX antibody referred to as 6A7 – the caption to figure 1. On the Santa Cruz catalogue the BAK antibody sc-517390 is not validated for recognizing active BAK - the only data shown is for western blotting. In Fig. 2 they report identical activation of BAX and BAK something we have never seen in cells. To use this antibody to quantify active BAK requires additional

experimental controls.

4. As I understand it, the model indicates that the BCL-2-BIM PBA indicates binding to available (empty) Bcl-2 proteins. But comparing panels b, c and d the total BCL2 = 2500. About 1000 of Bcl2 are occupied by BIM and 1500 are bound to BAX. There are no 'extra' Bcl-2 by 2 hours STS treatment. What about BAK? In panel J they find almost no BAK in HL60, NB4 or U937 cells. Comparing panel b and g why is the total BCL-2 around 3000 in b and >12,000 in panel g? And explanation is required.

5. Lines 202-205. The data shown do not report on relative affinities of PPIs. There are no binding curves in the manuscript for BCL-2 binding Bad compared to Bim. The competition experiment in extended Fig 7b is more determined by off-rate than affinity. Affinity is a well-defined quantitative value. Unless, the authors can justify the use of the term it should be replaced.

6. Lines 287-298. It is not clear what the authors are using as a metric. They state only that combining the top three correlations resulted in a limited improvement indicating that they are redundant. According to Fig 4 the correlation with BCL-2 level is essentially equivalent and it is likely all 4 are highly correlated with each other. Has this been tested? If all that is required is measuring BCL-2 levels by flow cytometry that would be much simpler than the method proposed here. It is not clear why the authors decided instead to add the negative correlation with BCL-XL-BAK complexes instead. Similarly in Fig. 5 the PPI metric selected is MCL1 total level, BCL-XL-BimBH3 PBA and BCL-XLBAK CPX yet the active BAK level and Bcl-XL-BAD CPX had higher Pearson's R. How was the selection made, why and what effect does it have on the prediction. These issues need to be addressed in the text and additional data justifying the selections provided.

7. It is not clear to me that the "six cases, including BC-6524 and BC-7230" showed an outstanding performance in predicting the clinical responses. According to Table 3 in the supplementary, of the 6 two of three non-responders were correctly identified and then only by including partial response, transient response and complete response as all responders – with no correlation to Ex vivo AUC and the extent of response are the others correctly reported. In addition, the methods section on Drug efficacy prediction model analysis is not clear making interpretation even more difficult. The claims made by the authors need to be more reflective of the true potential for clinical utility.

Minor technical criticisms or questions for consideration by the authors and editor:

1. Inconsistent nomenclature and incomplete information make the manuscript very hard to read.

2. In the manuscript CPX is not defined and LV is defined only in a figure.

3. More explanation is required for Extended Data Fig. 6. The description is not clear.

4. The statement: "Presumably, under the STS treatment condition used, the rate at which active BAXs were produced exceeded the rate that BCL2 sequestered BAX proteins, resulting in HL60 cells finally succumbing to apoptotic stress and initiating MOMP (Fig. 2a,d)." is confusing. I assume they mean that more BAX proteins become activated than can be bound by BCL2 (really all anti-apoptotic proteins). The rate is immaterial here as there is a large excess of BAX compared to anti-apoptotic proteins.

5. The statement on lines 192-193 misrepresents the data. BCL2 proteins did not continuously form additional BCL2-BAX complexes. According to the data in Fig. 2D complexes increased for 2 hours and then leveled off with no further increases. The text needs to be revised or the statement supported by specific data.

6. The description of the PPI probes is inadequate. "HEK293T cells expressing PPI probes (BIMBH3-eGFP, BIMEL-eGFP, BAD-eGFP, and NOXA-eGFP) and SF9 cells expressing BIMBH3-eGFP probe were lysed with Triton-X100 lysis buffer for PPI probe binding assay (PBA)." Does not even make clear whether the eGFP is fused to the amino- or carboxyl- end of the proteins, or if the fusion proteins retain authentic pro-apoptotic function or if the partner proteins (BIMEL, BAD, NOXA) are full length. What is the sequence of the BIM BH3 region being used? What is "proper" lysis buffer (Line 530)?

7. The statement on line 211 that BCL2 accounts for a dominant portion in sequestering... is misleading. The data indicates that there are more unoccupied Bcl-2s which fits with the later panels. It doesn't mean "BCL2 accounts for a dominant portion in sequestering pro-apoptotic proteins in HL60".

8. Figure 3F suggests that ABT-199 does not displace BIM from BCL-2 which fits with previous reports of

BIM binding Bcl-2 and BCL-XL not being displaced by BH3 mimetic drugs. This result is not very surprising. The text should be revised.

9. How come there are more active BAX and BAK in Fig. 3G (4000) than total BAX and BAK in Fig. 3H (3000)?

10. The cartoon in Fig. 3o does not seem to match the data. The data in panel n and the text indicate that there are many more Bcl-XL-BAK complexes formed when cells are treated with AZD-5991 than are released from MCL-1 but the cartoon shows an equal number (1). It isn't clear what the authors think happens to the released BAK.

Fri 03 Nov 2023

Decision on Article NBME-23-0122A

Dear Prof. Yoon,

Thank you for your revised manuscript, "Predicting the efficacy of BH3 mimetics through the profiling of multiple protein complexes in acute myeloid leukemia". Having consulted with the original reviewers — except Reviewer #3, who, despite our chasing efforts, has not provided a report and is unlikely to do so at this point — I am pleased to write that we shall be happy to publish the manuscript in *Nature Biomedical Engineering*.

We will now undertake a comprehensive review of your manuscript to ensure it meets our editorial and formatting requirements. In the coming weeks, you can expect to receive a checklist detailing these requirements. It is important that you adhere to these instructions when preparing your final manuscript files for upload. In the meantime, we kindly request that you prepare a detailed rebuttal letter addressing the remaining criticisms raised by Reviewers #4 and #5. You may send this letter directly to me, and I will forward your responses to the respective reviewers.

Best wishes,

Filipe

Dr Filipe Almeida
Associate Editor, Nature Biomedical Engineering

Reviewer #1 (Report for the authors (Required)):

The MS, NBME-23-0122A, contributes to therapeutic of AML and clinical application of ABT-199. The authors have corrected or explain all the concerns of this reviewer. This reviewer believes the revised MS has reach the publication request.

Reviewer #2 (Report for the authors (Required)):

The authors have addressed all the concerns I raised during my initial review, demonstrating a commendable commitment to enhancing the quality and clarity of their work.

Reviewer #4 (Report for the authors (Required)):

The responses are on the whole adequate. This is a very interesting and useful study. A few issues need to be addressed.

One of the other reviewers provoked a comparison of these results with BH3 profiling. I hope these results will be available to the reader. As will many of the responses, it was never quite clear what was being provided to the reviewer alone as opposed go going into the revised manuscript. The authors should recognize that the reviewer comments (not only this one, but all of them!) generally brought up points that should be explicated not only to the reviewers, but also to readers who are likely going to require the same explanations.

An additional point is that prior work with BH3 profiling found that sensitivity to probes of BCL-XL and MCL-1 were better negative predictors of in vivo response to venetoclax in AML than response to BAD, which is sensitive to BCL-2 as well as BCL-XL and BCL-w. This is congruent with the BCL-XL complexes found here to be a negative predictor. The authors may want to comment on this.

Figure 4b -I am not sure the authors understood my point. My point is that some of the correlations are negative, so why are all the R values positive? This could be fixed by simply labeling the y-axis "absolute value Pearson R".

Reviewer #5 (Report for the authors (Required)):

The authors are to be commended for the revisions made to the manuscript. Overall the text and figures are much clearer and the conclusions (for the most part) better supported by the data.

There is one important issue that remains. Extended data Fig. 6 reports the data used to calculate K_d values for the various interactions. To calculate a K_d the data must fit to a Hill equation - even to generate an "apparent K_d " the data must saturate. Straight lines indicate collisions not binding. In several places K_d values are reported for straight lines and for curves that do not saturate. It is not clear what value is being used by the software for saturation (the default in many packages is to use the highest value whether or not it is sensible). If the actual saturation point is known and the curve comes close to that value an estimate of an apparent K_d is reasonable but for the data presented it is not clear that these conditions apply. Furthermore the data in support for the occupancy calculations (extended Data Figure 6h) indicates occupancy (I assume it is fractional occupancy) and K_d for something called Probe at different concentrations. It is not clear what the two binding partners are and for the reasons outlined above it is not clear that the K_d values are meaningful. It is also not clear what the different colors are meant to indicate. Finally, where "Not Binding" is reported (Extended Fig. 6g) no data are provided in panels a-d and the histograms in panels e-f have no explanation as to the concentrations of the molecules.

Thu 30 May 2024

Feedback from reviewer #4 and #5 (NBME-23-0122B)

Dear Prof. Yoon,

Thank you for your efforts in reformatting and editing your study.

During this process, I consulted with Reviewers #4 and #5. Reviewer #4 is satisfied with the new version of the study; however, Reviewer #5 has raised additional concerns regarding the interpretations of some of your results. Please find Reviewer #5's feedback attached.

We hope you can address these concerns by making the necessary revisions to your study or by providing a convincing explanation to assure Reviewer #5 of the technical robustness of your work.

When you're ready, please reply to this email with the revised manuscript and a rebuttal letter.

Best regards,

Filipe

Appendix #1

Appendix 1

As far as I am concerned, the authors are not correct. I did not assume that they were using a non-equilibrium binding method like SPR.

The fit of the data to a straight line giving a low R-squared value just indicates how well the data fit the line. In an equilibrium experiment of the type they are performing, the probes can collide, particularly when one of the probes is tethered. Therefore, the data are in 2D rather than 3D.

In a concentration curve, real binding saturates. At high concentrations of the acceptor, the donor becomes saturated. Without that, in my opinion, they cannot distinguish collisions from binding.

Here is a quote from an elementary textbook on pharmacology:

"In the case of ligand-receptor complexes, K_d represents the ligand concentration and should be calculated when 50% of the receptors are bound to ligands."

If you do not know where 100% is (estimated from saturation), how do you calculate K_d when 50% of the receptors are bound to ligands? And if you use their definition of fractional occupancy, the graphs they provide in the rebuttal show fractional occupancies for Bcl-2 of approximately 25%, 10%, 70% (so this one is okay), and 0 (also okay to conclude no binding). For 2 of the 4 graphs, they do not have the data needed to properly estimate a K_d .

This is from the Vanderbilt web site:

Binding affinity simulation III

$$Q_a = Q_0 + (Q_{max} - Q_0) \frac{([P_{tot}] + [A_{tot}] + K_d) \pm \sqrt{([P_{tot}] + [A_{tot}] + K_d)^2 - 4[P_{tot}][A_{tot}]}}{2[P_{tot}]}$$

- For $[P] \gg K_d$, the binding curve will saturate at about 1:1 protein to ligand
- Here we have $[P_{tot}] = Q_0 = 0.5\text{mM}$ with a K_d of $0.25\mu\text{M}$
- Lack of curvature makes accurate fitting of the data difficult or impossible

Rebuttal 1

**Point-by-point responses to the comments on Chun *et al.*,
“Predicting the efficacy of BH3 mimetics through the profiling of
multiple protein complexes in acute myeloid leukemia”**

We deeply appreciate the insightful comments and constructive recommendations offered by the reviewers. All the comments have been meticulously addressed in a point-by-point fashion throughout the revision process. As a consequence, we are confident that the revised manuscript has been substantially improved in its scientific quality, as well as in clarity and readability.

We will first address the four points that were raised by the multiple reviewers or we considered important. These points are as follows:

1. Detailed explanation of the development for drug efficacy prediction models with combinatorial metrics.
2. Evidence for the predictive power of the combined metrics model.
3. Enhanced validation of clinical prediction outcomes.
4. In-depth characterization of PPI binding affinities.

The figure panel numbers referenced in this point-by-point response correspond to those in the revised manuscript, unless stated otherwise. In the main document, we have highlighted newly added or modified paragraphs in blue to ease the identification of revisions.

Responses to all reviewers

Detailed explanation of the development for drug efficacy prediction models with combinatory metrics

The reviewers inquired about the detailed procedures for developing our analysis model, which is based on multiple protein and protein-protein interaction (PPI) metrics. We employed linear regression to assess the performance of each model in terms of the resulting Pearson correlation value and the coefficients of PPI metrics along with their p -values. We discarded a certain model if any of the p -values became larger than 0.05, which signaled diminished statistical significance of the given model.

Rebuttal Fig. 1 | The BCL2-related metrics models based on BCL2 total level metric to predict *ex vivo* ABT-199 efficacy. Correlations between *ex vivo* AUC for ABT-199 and PPI profiles. The statistical indicators (coefficient, p -value) of each metric were presented.

Four BCL2-related metrics – BCL2-BAD PBA, BCL2-BIM_{BH3} PBA, BCL2-BAX CPX and BCL2 total level – exhibited the highest correlations with *ex vivo* efficacy (Fig. 4b). A simple combination of the four BCL2-related metrics only led to limited improvements in correlation while losing statistical significance (i.e., p -values becoming larger than 0.05) (Rebuttal Fig. 1 and Extended Data Fig. 8a).

Extended Data Fig. 8c,d

Therefore, we sought to examine more diverse combinations of metrics beyond BCL2 PPI and BCL2 total level. To this end, we divided the 32 AML samples into responsive and non-responsive groups using an *ex vivo* AUC threshold of 0.61, corresponding to an IC_{50} of 100 nM ABT-199 (Extended Data Fig. 8c)^{1,2}. We then examined approximately 10,000 different models, each with varying metric combinations and coefficients, selecting 67 models that made accurate predictions for the training set responsiveness, as validated by the remaining test samples (Extended Data Fig. 8d). Remarkably, the final 67 models consistently identified a subgroup of selected metrics rather than displaying diverse metrics randomly (Fig. 4e). Two metrics, BCLxL-BIM_{BH3} PBA and BCLxL-BAK CPXs, were negatively correlated with *ex vivo* AUC values, presumably reflecting a compensatory role for BCLxL in counteracting accumulated apoptotic pressure in AMLs.

Fig. 4e

Upon detailed examination of the 67 models, we noticed a pattern where the information from each metric displayed minimal overlap. For example, while combining the BCL2-BAX CPX and BCL2-BIM_{BH3} PBA data successfully (which represented occupied and unoccupied portions of BCL2), neither metric could be combined with the BCL2 total

level data (which included both occupied and unoccupied pools), likely due to partial overlap in the information they provided (Rebuttal Fig. 2a versus 2b,c and Extended Data Fig. 8a). Similarly, the BCLxL-BAK CPX consistently exhibited higher absolute values in its correlations to *ex vivo* AUC than BCLxL-BAX CPX, possibly because of redundancy in the information concerning BAX that was already considered by the inclusion of the BCL2-BAX CPX metric (Fig. 4e).

Rebuttal Fig. 2 | The combination metrics models based on BCL2 total level metric to predict *ex vivo* ABT-199 efficacy. Correlations between *ex vivo* AUC for ABT-199 and PPI profiles. The statistical indicators (coefficient, *p*-value) of each metric were presented.

As a result, a combination of BCL2-BIM_{BH3} PBA, BCL2-BAX CPX, and BCLxL-BAK CPX (with a negative coefficient) was selected, showing a slight but significant enhancement in the correlation to AUC values with statistical significance (*p*-value<0.05) for all metrics examined (Fig. 4f). We applied the same approach to develop an analysis model for AZD-5991 efficacy, and found that a combination of MCL1 total level, BCLxL-BIM_{BH3} PBA and BCLxL-BAK CPX gave the highest correlation while maintaining statistical significance (Fig. 5f). It is noteworthy that the principle of minimal information overlap between metrics appeared to be preserved here as well, given that the final combination encompassed occupied and unoccupied populations of both MCL1 and BCLxL with seemingly minimal overlap.

Fig. 4f (left) and Fig. 5f (right)

In the revised manuscript, we have revised the main text and the Method section to clarify the process of model analysis and to include these results of the statistical analysis. We hope this revision could help readers understand this technology clearly and apply it in actual clinical settings. Also, we have uploaded all raw data and the codes generated or analyzed during the study to GitHub.

Evidence for the predictive power of the combined metric model

The reviewers requested additional evidence to support the predictive power of the combined metric model, in particular, compared with the single-metric models. To see the differences between the models in a more specific manner, we considered mean squared error (MSE), which serve as indicators of deviations from the prediction model.

$$MSE = \frac{1}{n} \sum_{i=1}^n (Y_i - \hat{Y}_i)^2 \quad (1)$$

n = number of samples (32 in our study), Y_i = Experimentally determined AUC, \hat{Y}_i = Estimated score from the model.

Rebuttal Fig. 3 | Identification of outliers in the ABT-199 drug efficacy prediction models.

Correlations between *ex vivo* AUC for ABT-199 and PPI profiles for the BCL2-BAX CPX, (b) Combination of BCL2-BIM_{BH3} PBA, BCL2-BAX CPX, and BCLxL-BAK CPX metrics. The residuals of each outlier (*ex vivo* AUC – estimated score) were presented.

As shown in Rebuttal Fig. 3, the BCL2-BAX CPX model exhibited an MSE of 0.021, while the combined metrics model showed an MSE of 0.014. This indicates that the combined model likely generates a fewer number of outliers. To see this, we computed the residual value for each data point and performed z-score normalization for residuals. An individual data point was classified as an outlier if its z-score were larger than 1.5, which was twice as strict as the general criterion for an outlier (typically, larger than 3.0). According to this reinforced criterion, the model that considered only BCL2-BAX CPX identified six outliers, whereas the combined model identified only three outliers (Rebuttal Fig. 3).

Rebuttal Fig. 4 | The combination metrics model contains fewer outliers for drug response prediction. Correlations between *ex vivo* AUC for ABT-199 and PPI profiles. (a) BCL2-BAX CPX, (b) Combination of BCL2-BIM_{BH3} PBA, BCL2-BAX CPX, and BCLxL-BAK CPX metrics.

Furthermore, the combined model provides a valuable mechanistic insight into how the anti-apoptotic proteins in the BCL2 family function to counter apoptotic pressure. For instance, BC-7052 exhibited a relatively low count of BCL2-BAX complexes among the 32 cohort samples, resulting in a modest score of 0.40 (Rebuttal Fig. 4a). However, BC-7052 had a relatively higher count of BCL2-BIM_{BH3} PBA, which increased its score to 0.63 in the combined model, exactly matching the experimentally determined AUC of 0.63 (Rebuttal Fig. 4b). Thus, comprehensive profiling of both current complex populations (i.e., BCL2-BAX CPX) and potential PPI populations (BCL2-BIM_{BH3} PBA) would provide a better measure for dependence of the given cancer on BCL2 than considering only one type of populations. Another example is BC-6063. In the BCL2-BAX CPX model, it exhibited an estimated score of 0.60, which was significantly higher than the experimentally determined AUC value of 0.33 (Rebuttal Fig. 4a). However, this discrepancy was substantially reduced in the combined model, where a large BCLxL-BAK CPX count determined for BC-6063 decreased the score to 0.42 (Rebuttal Fig. 4b).

Next, we quantitatively defined the predictive power with receiver operating characteristic (ROC) analysis. For binary prediction between responsive and non-responsive samples determined by *ex vivo* ABT-199 AUC (Extended Data Fig. 8c), the resulting AUC showed the highest values for the combined model compared to single-metric models (Fig.

4g,h). Moreover, the MSE was also determined to be the smallest for the combined model, indicating higher accuracy in estimating ABT-199 efficacy (Fig. 4h).

Fig. 4g,h

Together, the combined model not only performs better in predicting drug responsiveness for individual data points (i.e., smaller MSEs overall and fewer outliers), but also offers important mechanistic insights into the modes of action of ABT-199 in achieving efficacy in the studied AML samples.

Enhanced validation of clinical prediction outcomes

The reviewers concerned about the clarity of the manuscript's previous version, particularly regarding how BCL2 PPI profiling was used to make predictions and correlate with clinical outcomes. To address this, we would like to highlight that we conducted BCL2 PPI profiling on primary AML samples for a total of ten patients, including three new patient cases added during manuscript revision (Supplementary Table 4). Our aim was to examine whether these profiling results could inform therapeutic decisions related to ABT-199 treatment.

Specifically, we utilized the combined metric model depicted in Fig. 4f, applying a threshold score of 0.61 that corresponds to an IC₅₀ of 100 nM ABT-199 (Extended Data Fig. 8c), to stratify patients into potential responders and non-responders^{1,2}. Responses to *in vivo* ABT-199 treatment were assessed according to the 2017 ELN guidelines³; patients achieving complete remission (CR), CR with incomplete hematologic recovery (CRi), or partial remission (PR) were classified as responders (R), while those who showed no response were classified as non-responders (NR) (Supplementary Table 4).

Patient No.	Sample No.	Estimated score	In vivo response	Note
1	BC-6524	0.90	R (CRi)	
2	BC-7084	0.78	R (PR)	+Azacitidine
3	BC-7082	0.69	R (PR)	
4	BC-7107-R	0.93	R (CRi)	+LDAC
5	BC-7052	0.64	NR	+LDAC
6	BC-7081	0.19	NR	+Decitabine
7	BC-7230	0.27	NR	
8	BC-8634	0.39	NR	+Dacogen
9	BC-8746	0.54	NR	+Azacitidine
10	BC-8784	0.55	NR	+Dacogen
	BC-7064-R	0.36	n.a.	Relapsed (BC-7064)
	BC-7082-R	0.40	n.a.	Relapsed (BC-7082)
	BC-7107-R2	0.20	n.a.	Relapsed (BC-7107-R)
	BC-8900	0.47	n.a.	After treatment (BC-8634)
	BC-9458	0.59	n.a.	After treatment (BC-8746)
	BC-9492	0.60	n.a.	After treatment (BC-8784)

Supplementary Table 4

Remarkably, our model's predictions for responders and non-responders closely aligned with actual clinical outcomes for nine out of the ten patients, achieving a sensitivity of 100% and a specificity of 83% (Fig. 6e). BC-7052 represents an interesting exception; our model projected a favorable response with a score of 0.64. While the patient initially showed a short-term response and entered partial remission, they subsequently

e

		Estimated score	
		NR (<0.61)	R (≥0.61)
In vivo response	NR	5	1
	R	0	4

(n=10)

Sensitivity = 100%
Specificity = 83.3%

Fig. 6e

ceased ABT-199 treatment (Extended Data Fig. 10a). Interestingly, *ex vivo* efficacy tests showed an AUC value of 0.63 for this patient's sample, mirroring our model's prediction (Supplementary Table 4). This concordance highlights a limitation of the current model, which relies solely on *ex vivo* drug efficacy data. It suggests that incorporating factors specific to *in vivo* cancer evolution may further refine the model's predictive abilities. Notably, BC-7052's sample displayed a high count in the BCLxL PBA assay, which is not currently accounted for in our model. This suggests that BCLxL protein activity might become upregulated as ABT-199 treatment continues, warranting further investigation (Extended Data Fig. 10a).

Fig. 6f and Extended Data Fig. 10h

In response to the reviewer's comment, we also performed a statistical analysis. Given the limited number of patient cases, we employed the Mann-Whitney test to assess the data. We observed a significant difference in the initial response scores between the responder (R) and non-responder (NR) groups, comprising a total of ten patients (two-sided, $p=0.014$) (Fig. 6f). When we included six additional samples from the same cohort – collected after relapse and therefore added to the NR group – the contrast between the two groups intensified, further reducing the p -value to 0.004 (Extended Data Fig. 10h).

We acknowledge that our study is based on a small cohort size. In the revised manuscript, we have clarified our methodology for making predictions, and how these predictions correlate with clinical outcomes (in the main text, the Methods section and Supplementary Table 4). We also included the results of the statistical analysis to provide further context.

In-depth characterization of PPI binding affinities

We're grateful to the reviewers for their insightful comment. We would like to highlight that our experimental setup allows us to conduct binding and competition experiments, such as those between BAD and BIM, under conditions that closely approximate equilibrium (Fig. 2k). Given our platform's capability for single-molecule counting, we can accurately determine the number of surface baits and bound preys. This leads directly to occupancy values, which at varying prey concentrations can be used to calculate the dissociation constant (K_d) for a given binding reaction, as specified by Equation (2).

$$\text{Occupancy} = \frac{[\text{PPI probe}]}{K_d + [\text{PPI probe}]} \quad (2)$$

To validate the feasibility of this approach, we conducted a series of binding reactions that covered all possible combinations between three anti-apoptotic proteins (BCL2, BCLxL, and MCL1) and four different PPI probes (BIM_{EL}, BIM_{BH3}, BAD, and NOXA) (Extended Data Fig. 6a-d). Specifically, we fused these anti-apoptotic proteins with red fluorescent proteins (mCherry) and performed surface pull-downs to count the surface baits. Subsequently, we introduced eGFP-labeled PPI probes at increasing concentrations and allowed more than 10 minutes for the binding reactions to reach equilibrium. This enabled us to ascertain the resulting occupancy values, which represent the proportions of surface-immobilized anti-apoptotic proteins occupied by the PPI probes (Extended Data Fig. 6h).

Extended Data Fig. 6a-d, and 6h

Our results showed that while the full-length form of BIM (BIM_{EL}) bound to all three types of anti-apoptotic proteins with K_d values ranging from 50 to 150 nM, the BIM_{BH3} probe, containing only the BH3 domain, exhibited considerably stronger binding affinities that ranged from 49.8 ± 2.2 to 4.6 ± 0.2 nM (Extended Data Fig. 6a,b). The BAD probe demonstrated nM-range K_d values for BCL2 and BCLxL but exhibited negligible binding to MCL1 (Extended Data Fig. 6c). In contrast, the NOXA probe bound strongly only to MCL1 and showed no affinity for BCL2 or BCLxL (Extended Data Fig. 6d). These results align with previously reported findings using fluorescence polarization⁴. Moreover, our experiments indicate that by determining the occupancy value at a specific prey concentration, K_d values can be estimated in a single experiment with reasonable accuracy (Extended Data Fig. 6h). Using a PPI probe concentration of 5 nM, for instance, we were able to cover K_d values spanning almost two orders of magnitude (from 1 to 100 nM).

In summary, we have demonstrated that our PBA experiments are effective for determining the K_d values of specific PPI reactions. We have included these new datasets in Extended Data Fig. 6a-h of the revised manuscript.

Reviewer #1

Comment 1) Fig4 c,d, and e, the score for Bcl-2/Bim complex is relative lower (0.43), while those for Bcl-2/Bax and the combination metrics are 0.79 and 0.87, respectively. If the authors believe the combination metrics is significantly more accurate than Bcl-2/Bax in the view of prediction, please discuss it deeply. Especially, for clinical application does it deserve to measure and calculate such a complexed index instead of one pair of PPI?

Response: We appreciate the reviewer's comment regarding the importance of combining different metrics when the correlation value only slightly increases. We would like to emphasize that although the increase in the correlation value may seem small, the combined model significantly reduces the number of outliers in the prediction model. Please see the section titled "Evidence for the predictive power of the combined metrics model" in our response to all reviewers.

Comment 2) Fig.3g, at 0 hour of ABT-199 treatment, Bax and Bak activation are much higher than those indicated in Fig 3b, which seemed rational since not any apoptosis stimuli there.

Comment 3) In Fig.3g and Fig.3h, the activated Bax is higher than total Bax at several time points. Comparably, in Extended Data Fig. 4b, Bcl-xl complex is higher than total Bcl-xl.

Response: We appreciate the reviewer's meticulous attention to these details. For different sets of experiments, we used different lysate concentrations and incubation times for immunoprecipitation of these BCL2 family proteins on surface, accounting for why we obtained different count numbers even for the same assays.

Fig. 3g,h

We agree with the reviewer's concern that these different count numbers may induce unnecessary confusion to readers. We carried out experiments in Fig. 3g again, now using the same experimental conditions as those in Fig. 3b,h. This enabled direct comparison of the resulting single-molecule counts between the two figures (Fig. 3g,h). The updated data revealed that the activated BAX population continuously rose, while the total BAX population remained relatively stable throughout the course of ABT-199 treatment. After 4 hours of ABT-199 treatment, the active BAX population amounted to about two thirds of the total BAX population.

In the revised manuscript, we also present recalibrated data sets for Extended Data Fig. 3 that enable direct comparison between one another.

Comment 4) How did the authors measure the Bak activation?

Response: To form pores in the mitochondrial outer membrane, individual BAK proteins undergo conformational changes to activated states that allow themselves to oligomerize with other BAX and BAK proteins. What is the most distinct in this conformational change is exposure of the BH4 domain ($\alpha 1$ helix), which is otherwise buried in the core structure (composed of the $\alpha 3$ to $\alpha 5$ helix) in the inactive state⁵. Thus, antibody binding to this exposed BH4 domain provides a way to distinguish active BAK proteins out of the entire BAK population. Indeed, for another pore-forming protein, BAX, which exhibits similar conformational changes in the activated state, it is well known that activated BAX can be

detected using a monoclonal antibody (6A7 clone) that detects the N-terminus region of BAX⁶.

The monoclonal antibody (AT38E2 clone) we used for active BAK detection was raised against a recombinant protein corresponding to amino acids 29-187 of BAK proteins, virtually encompassing the whole amino acids of BAK. We thus investigated whether AT38E2 selectively binds to the $\alpha 1$ helix, but not to other regions of BAK. Specifically, we divided BAK into four fragments: $\alpha 1$ helix, $\alpha 2-3$ helices, $\alpha 3-5$ helices, and $\alpha 6-8$ helices respectively. We labeled each of these fragments with eGFP at the C-terminus and examined its binding to AT38E2 using single-molecule pull-down. The resulting data clearly indicated that AT38E2 selectively bound to the $\alpha 1$ helix of BAK (Extended Data Fig. 1h).

Extended Data Fig. 1h,i

Next, we verified whether the antibody could detect endogenous, activated BAK proteins. It was shown that BAK proteins could be activated by modest heat as well as by triton X-100 treatment under *in vitro* conditions^{7,8}. We applied either heat or triton X-100 to crude extracts of HL60 and Ramos cells and determined the active BAK populations through single-molecule pull-down and co-IP. The heat treatment increased the active BAK counts by four times for both HL60 and Ramos cells (Extended Data Fig. 1i). Moreover, the triton X-100 treatment induced an order-of-magnitude change in the active BAK level, indicating that most of the BAK proteins in lysates were converted to an activated form. Based on these data, we determined the population of active BAK proteins using the AT38E2-clone antibody without involving an additional treatment step.

Comment 5) Since an accurate quantification is one of the most advances for single-molecule pull-down, why a fold number is used in the right Y-axis in Fig.2L and Extended Data Fig. 4a rather than a single molecule count number?

Response: In response to the reviewer's comment, we have updated the Y-axis data in both Fig. 2k and Extended Data Fig. 3a to reflect the corresponding single-molecule counts. To achieve this, we re-conducted the competition experiments within the probe binding assays. This provides more quantitative insights into the competition between the two probes, BIM_{BH3} and BAD, when binding to BCL2.

Fig. 2k and Extended Data Fig. 3a

Comment 6) This reviewer suggests authors to provide details on the fitting of the drug efficacy prediction model, especially, the theoretical formula to calculate the estimated score.

Response: We appreciate this important comment by the reviewer. We employed linear regression to assess the performance of each model in terms of the resulting Pearson correlation value and the coefficients of PPI metrics along with their *p*-values. For more detailed accounts of developing drug efficacy prediction models, please see the section titled “Detailed explanation of the development for drug efficacy prediction models with combinatory metrics” in our response to all reviewers.

Reviewer #2

Minor points:

Comment 1) Several figures are overloaded with the material, which makes them hard to read. The font size of some labels is very small. Authors are recommended to review and, possibly, reorganize Figures 1, 2 4, 6 and Extended data Fig. 10. For example, the assay schematics (Fig. 2, panels e and k) could be moved to supplementary data, and some data tables under the graphs can be removed (Fig.4 c and d).

Response: In response to the reviewer comment, we have revised the figures according to the comments. The revisions include 1) making experimental scheme in Fig. 1a-c more concise, 2) moving Fig. 2k to Extended Data Fig. 6 (Fig. 2e retained but modified to be more concise), 3) removal of the data tables in Fig. 4c and d, as well as Fig. 5b-d, 4) making PPI diagnostic results in Fig. 6b and g more intelligible, and 5) reorganization the clinical data in Fig. 6 and Extended data Fig. 10.

Comment 2) The authors don't discuss if they applied the developed single-molecule pull-down and co-IP platform to quantify the PPI complexes of interest in the healthy control cell lines/cell material from the healthy donors, or to other hematological malignancies highly susceptible to BCL2 inhibition, such as chronic lymphocytic leukemia (CLL) or multiple myeloma. In future, such data will enrich the developed analytical model and enhance the predicting power of clinical treatment outcome.

Response: We thank the reviewer for this insightful comment. In accordance with the comment, we determined the expression levels of three major anti-apoptotic proteins (BCL2, BCLxL and MCL1) as well as active BAX and BAK populations in commercially available peripheral blood mononuclear cells (PBMCs) from a healthy donor.

Unlike cancer cells, which usually show upregulation of at least one anti-apoptotic protein, the PBMC sample showed minimal counts for all three anti-apoptotic proteins, indistinguishable from those of the control experiments with blank buffer (referred to as buffer) (Rebuttal Fig. 5a-c). Interestingly, the PBMC sample showed considerably high active

BAX and BAK populations, comparable to HL60 cells (Rebuttal Fig. 5d,e). This indicated predisposition of PBMCs to apoptotic death, consistent with the known labile, short-lived characteristics of these cells. At the same time, the count of BCL2-BAX complexes was determined to be low for the same PBMC sample (Rebuttal Fig. 5f), indicating that the active BAX and BAK populations were not sequestered by BCL2 (presumably, nor by BCLxL and MCL1 as well). These data also importantly suggest that normal cells make negligible contributions to BCL2-family PPI signals, and that any active signals would be almost exclusively attributed to cancer cells.

Rebuttal Fig. 5 | Comparison of BCL2 family PPI profiles from healthy PBMC sample and HL60 cells with single-molecule co-IP. a-c, Total levels of anti-apoptotic proteins. **d-e**, Active BAX/BAK level. **f**, BCL2-BAX complexes.

Next, we attempted to apply our platform to multiple myeloma (MM). As a preliminary study, we determined the single-molecule counts for the BCL2-BIM_{BH3} PBA and BCL2 level assays for four different MM cell lines of RPMI8226, U266, NCI-H929, and KMS-12-PE. We chose these two assays as they were ranked highest in terms of the correlation with the *ex vivo* ABT-199 efficacy amongst the 22 assays we examined. Remarkably, only KMS-12-PE cells showed significant counts for the two assays examined (Rebuttal Fig. 6a-c). Moreover, this contrast in the single-molecule counts was closely paralleled by the responses of these cell lines to ABT-199, where only KMS-12-PE cells responded to ABT-199 while other cells

remained refractory even at 1 μ M ABT-199 (Rebuttal Fig. 6d). These data, though preliminary, suggest a possibility of extending application of our platform to other hematologic malignancies like MM beyond AML.

Rebuttal Fig. 6 | Application of the BCL2 PPI profiling to MM cell lines. a-c, Comparison of PPI profiles among the MM cell lines. (a) BCL2 total level, (b) BCL2-BIM_{BH3} PBA, (c) BCL2-BIM complex. d, Viability assessment of MM cell lines following a 24-hour treatment with ABT-199. Viability is indicated by the double negative populations after Annexin V/PI staining, which were analyzed using flow cytometry. IC₅₀ values were obtained from the fitted curve (right panel).

Reviewer #3

Comment 1) The final part reporting on the correlation with in-vivo response is very difficult to understand what has been done. The authors claim that based on 6 out of 10 matches between in-vivo and ex-vivo response that their test system works. No statistical tests have been performed and conclusions made by the authors are not justified based on the presented data.

Response: We appreciate the reviewer's concerns about the clarity of the manuscript's previous version, particularly regarding how BCL2 PPI profiling was used to make predictions and correlate with clinical outcomes. To address this, we would like to highlight that we conducted BCL2 PPI profiling on primary AML samples for a total of ten patients, including three new patient cases added during manuscript revision. We have revised the Fig. 6 and Extended Data Fig. 10 to include more comprehensive details how our BCL2 PPI profiling correlates with *in vivo* drug response. For more detailed accounts of these analysis, please see the section titled "Enhanced validation of clinical prediction outcomes" in our response to all reviewers.

Comment 2) The paper shows a good correlation of the newly developed assay with ex-vivo response to venetoclax, which is encouraging. However, it is not illustrated why such an assay would be better ex-vivo drug response profiling of cancer cells as proposed by different consortia. Further the peptide-based BH3 profiling is a further well established assay which measures the same endpoint but to which the new assay is not compared to.

Response: We appreciate the reviewer's insightful comment, as an objective comparison with other technical platforms will be crucial for eventual clinical applications. In response, we conducted BH3 profiling on 14 primary AML samples from the 32 cases that were used to create the predictive model depicted in Figure 4. This was due to the limited availability of samples.

Extended Data Fig. 9b,c

BH3 profiling involved the addition of BH3 peptides from BIM, BAD, and HRK to individual AMLs (after cell permeabilization), and we monitored mitochondria depolarization using JC-1, as described in previous studies^{9,10}. We used DMSO as a negative control, which preserved mitochondrial membrane potential, and FCCP as a positive control, inducing total depolarization of the mitochondria membrane. Notably, HL60 cells exhibited robust depolarization signals in response to the BAD peptide but not to the HRK peptide, consistent with previous reports that correlated the difference between the BAD- and HRK-peptide responses with the responsiveness to ABT-199 (Extended Data Fig. 9b,c)^{4,11}.

Rebuttal Fig. 7 | BH3 profiling on primary AML cells. **a**, Relative fluorescence units (RFU) measured at the wavelength of 590 nm for primary AML cells through the treatment of BH3 peptides (BIM, BAD, and HRK) or BH3 mimetics (ABT-199, AZD-5991). **b**, Depolarization of primary AML cells through the BH3 profiling. Depolarizations by 10 μM of BAD peptides were indicated.

Having confirmed the reliability of the BH3 profiling assay in our hands, we proceeded to examine the 14 primary AML samples (Supplementary Table 3). BC-6933, which showed

a marginal *ex vivo* response to ABT-199, displayed consistently low depolarization signals in response to varying concentrations of the BAD peptide (Rebuttal Fig. 7a,b). Conversely, BC-5034 and BC-6524, which exhibited favorable *ex vivo* responses to ABT-199, showed considerably higher levels of depolarization induced by the BAD peptide (Rebuttal Fig. 7a,b). These data confirmed the promising predictive power of BH3 profiling in assessing ABT-199 responsiveness.

Next, we quantitatively compared the predictive power between two methods: BH3 profiling and BCL2 PPI profiling. We combined the depolarization signals obtained with BAD and HRK peptides (at different concentrations) and explored their various combinations to compare with the *ex vivo* ABT-199 efficacy determined for individual samples. Among the combinations, BAD-HRK (both measured at 10 μ M) showed the highest Pearson correlation value of 0.65 with the *ex vivo* ABT-199 efficacy (Extended Data Fig. 9d).

Importantly, for this smaller cohort consisting of 14 AML samples, the BCL2 PPI profiling essentially maintained the performance parameters examined, including correlation, MSE, and AUC from the ROC analysis, which were superior to those determined for the BH3 profiling (Extended Data Fig. 9d-g and 9j). Moreover, all the three outliers identified in the BH3 profiling model (BC-5466, 6528 and 6802) were found to be inliers in the model based on BCL2 PPI profiling (Extended Data Fig. 9d,e).

Extended Data Fig. 9d-g, and 9j

We also attempted to use the BCL2 level measured by flow cytometry as a predictive biomarker, which only resulted in muted performance across all evaluated parameters compared to BH3 profiling and BCL2 PPI profiling (Extended Data Fig. 9h-j).

Extended Data Fig. 9h,i

Reviewer #4

Comment 1) The overall approach - this is a great source of the novelty of the paper, and as such, we simply need to know more about it. I am sure that somewhere they have data regarding calibration of their microscopy to know that they are actually getting counts of single molecule data. It would be nice if they would refer to this if there is a published reference, or put it in the supplement if not.

Response: We thank the reviewer for this comment regarding our image processing. To clarify, each image was processed after capturing an average of fluorescence signals over a time resolution of 100 ms, using total internal reflection (TIR) for excitation. During the TIR imaging, the fluctuations in background fluorescence exist, which can be attributed to both autofluorescence from the imaging chip and freely diffusing, fluorescently labeled prey proteins. To address this, we captured 8 to 10 control images from empty reaction chambers to establish a limit of background signals, identified from the distribution of signals recorded in each pixel of the control images. Our analysis software then employed a cutoff identified from the limit of background value (1,015 A.U. for 488 nm wavelength excitation) to eliminate 99% of these background signals (Extended Data Fig. 2a).

Extended Data Fig. 2a,b

Upon background subtraction, three sequential images were averaged, corresponding to a 300 ms time frame. This enabled us to identify diffraction-limited fluorescence spots and accurately count surface-immobilized, single-fluorescent molecules (Extended Data Fig. 2b). We verified the reliability of this approach for all the assays employed in this study, observing that the coefficients of variation were typically less than 10% and never exceeding 15% in any instance (Fig. 1g-i, and Extended Data Fig. 3). This high level of reproducibility was facilitated by the low autofluorescence of our imaging chip, which enhanced the distinction between background and genuine fluorescence signals, and by the almost complete suppression of non-specific adsorption of extraneous proteins onto the surface.

Extended Data Fig. 2c

Using the method described above, we have achieved a resolution that enables us to identify and count up to 2,000 single-molecule fluorescence spots within a $50 \times 100 \mu\text{m}^2$ field of view (Extended Data Fig. 2c). This establishes an upper limit for the identification of diffraction-limited spots. When the number of such spots exceeds this limit, overlapping begins to occur, compromising our ability to identify individual spots (Extended Data Fig. 2d). To address this limitation, we switched to integrate the total fluorescence signals captured across all pixels within the field of view. Simultaneously, by recording time-traces of photobleaching, we established a relationship between the number of single-molecule spots and the total fluorescence intensity, particularly, in the range that permitted the identification of individual spots (i.e., below the upper limit), which is typically linear (Extended Data Fig. 2e,f). This linear relationship ultimately allows us to extrapolate total fluorescence intensity to corresponding single-molecule counts in regions exceeding the upper limit (Extended Data Fig. 2g).

Extended Data Fig. 2d-g

Recently, we have quadrupled the field of view of our SMPC platform, thus achieving a resolution capable of distinguishing up to 12,000 single-molecule counts. We have revised the Methods section, as well as Fig. 1f and Extended Data Fig. 2, to include more comprehensive details on our image processing approach with references^{12,13}. Additionally, all relevant analysis codes have been uploaded to GitHub for public access.

Comment 2) It is not acceptable that the raw data and the code for the drug efficacy prediction model (lines 619-626) are available only "upon reasonable request". For this to be published, they must instead be available directly on line to anyone reading the paper. Moreover, the linear regressions and the specific coefficients used must also be available for immediate inspection, not just upon request.

Response: In accordance with the comment, we have uploaded all raw data and code generated during this study to GitHub. The data is now publicly available via the following GitHub link: <https://github.com/tyyoonlab-snu/Nat-Biomed-Eng-2023->. We have also updated the Data Availability and Code Availability sections of the manuscript to reflect this change.

Comment 3) If this type of assay were ever to be used more broadly, it would be important to know the impact of shipping and storing conditions. For instance, did the authors ever compare their PPI results between fresh and frozen samples, either with cell lines or primary AML samples?

Response: We appreciate the reviewer's comment, which is highly relevant for the practical implementation of our assays. To address this point, we conducted a series of operations using a single HL60 cell sample (Extended Data Fig. 1j). Specifically, we divided one HL60 cell culture (passage 0) into two batches (passage 1). Cells from the first batch were directly harvested for measurement, creating Sample #1. The remaining cells from this batch were frozen using liquid nitrogen and later thawed for a second measurement, constituting Sample #2. The entirety of the second batch was cryopreserved with a commercial freezing medium (Recovery™ Cell Culture Freezing Medium; Gibco 12648010). After thawing and culturing these cells for an additional two days, a portion was harvested for measurement as Sample #3. The remaining cells were subjected to a second freeze-thaw cycle using liquid nitrogen before measurement, resulting in Sample #4.

Extended Data Fig. 1j-l

To assess potential damage or alterations to the cells through these various treatments,

we performed two assays measuring both total BCL2 levels and BCL2-BAX complexes (Extended Data Fig. 1k,l). Remarkably, we found that the total BCL2 levels remained essentially consistent across all four samples. Although the BCL2-BAX complexes showed slight dissociation in the samples that underwent at least one freeze-thaw cycle, the degree of dissociation was less than 10%, even in Sample #4. Therefore, we concluded that the freeze-thaw cycles had minimal impact on our assay results. This information has been included in the revised manuscript as Extended Data Fig. 1j-l.

Comment 4) In overexpression models in cell lines, using FLIM-FRET, the laboratory of David Andrews has examined PPI and how they are affected by BH3 mimetics. The authors may want to compare their results to these in the discussion.

Response: We wish to acknowledge the significant contributions made by David Andrew's laboratory in this area. We find strong parallels between their work and our own observations. For instance, their previous FLIM-FRET study explored the interactions between BCLxL (or BCL2) and various pro-apoptotic proteins, including BH3 mimetics, within living cells¹⁴. This study identified that BIM possesses an additional binding motif, referred to as the CTS region in its C-terminus, allowing it to tightly bind with anti-apoptotic proteins. Notably, the study showed that ABT-263, a dual inhibitor of BCLxL and BCL2, readily released BAD and BID but was less effective in releasing BIM, indicating that BIM likely engages with BCLxL and BCL2 through multiple binding sites. This finding raises the possibility that BIM could contribute to resistance against BH3 mimetics that target only the BH3 domain.

In our study, we observed similar behavior. Specifically, ABT-199 was unable to dissociate the BCL2-BIM complex during the early stages of apoptosis, whereas the BCL2-BAX complex was more readily disrupted, initiating apoptosis (Fig. 3f). Additionally, our analysis of a primary AML cohort found that the BCL2-BIM complex had a lower correlation with ABT-199 efficacy compared to the BCL2-BAX complex (Fig. 4b-d). These observations corroborate the findings from the Andrew group, suggesting that BCL2-BIM complexes are more stable than BCL2-BAX complexes.

In another study from the Andrew group using FLIM-FRET, it was shown that BAD could displace both BID and BAX from BCLxL¹⁵. We have replicated this competition between BAD and the BH3 domain of BIM for binding to BCL2 in our PBA experiment. Our data also underscore its importance for predicting the efficacy of BCL2 inhibitors (Fig. 2k and Fig. 4f). Additional observations from their work, which indicated that BAD disrupts BCLxL-BAX complexes through an allosteric effect, provide further support for our hypothesis that the BAX complex is a key driver of BH3-mimetic efficacy. These discussions and citations have been included in our revised manuscript.

Comment 5) The title and motivation of this paper is all about prediction. As such it is disappointing that we never see a quantitative study of how good a predictor they actually have here. The validation set is just a half dozen patient cases. I would suggest they need more cases, and then an objective measure of the performance of their predictor using a measure of binary prediction like a receiver operating characteristic (ROC).

Response: We appreciate the reviewer's concerns about the clarity of the manuscript's previous version, particularly regarding how BCL2 PPI profiling was used to make predictions and correlate with clinical outcomes. To address this, we would like to highlight that we conducted BCL2 PPI profiling on primary AML samples for a total of ten patients, including three new patient cases added during manuscript revision. Please see the section titled "Enhanced validation of clinical prediction outcomes" in our response to all reviewers.

We also evaluated the performance of the drug efficacy prediction models through the established statistical methods, MSE and ROC curve analysis. For more detailed accounts of these analysis, please refer to the section titled "Evidence for the predictive power of the combined metrics model" in our response to all reviewers.

Comment 6) Do I understand correctly that in Figure 4b, there are no negative correlations, only positive correlations? This is quite surprising. it is especially surprising as they apparently include the BCLxL-BAk CPX value in their predictor with a negative coefficient, despite it apparently positively correlating with sensitivity to ABT-199?

Response: We apologize for confusion the reviewer might have experienced and would like to provide an additional explanation. In making Figures 4b, we used linear regression to determine the correlation between the *ex vivo* ABT-199 efficacy and each PPI profile. Each generated model provided *p*-values for the linear regression itself, as well as for the intercept. To main statistical significance of the model, we accepted a model only when all the *p*-values pertinent to the model were lower than 0.05. For example, both BCL2-BAX and BCL2-BIM CPX data showed statistical significance despite their widely different correlation values (0.79 versus 0.43). On the other hand, we had to reject the single-metric model based on the BCLxL-BAK CPX data because their *p*-values for the model and intercept were 0.80 and 0.17, respectively and failed to meet the criteria, which was now indicted in the revised Fig. 4b.

Fig. 4b

While the BCLxL-BAK CPX data did not show statistical significance with the ABT-199 drug efficacy as a single factor, it acted as a negative correlator for drug efficacy with its own *p*-value of 0.044 when combined with BCL2-BIM_{BH3} PBA and BCL2-BAX CPX for our chosen model (*p*-value of the model: 1.19e-10) (Fig. 4f). For more detailed accounts of constructing the drug efficacy prediction models, please see the section titled “Detailed explanation of the development for drug efficacy prediction models with combinatory metrics” in our responses to all reviewers.

In the revised manuscript, we have revised the main text and the Method section to clarify the process of model analysis and to include these results of the statistical analysis.

Also, to avoid unnecessary confusion of readers, in Fig. 4b of the revised manuscript, we displayed the statistical significances for all of PPI profiles respectively.

Comment 7) Figure 4, they show a correlation with BCL-2 levels and response. They may want to comment on previous attempts to correlate response to BCL-2 levels alone (mRNA or protein) that were apparently not as successful. Due to method of measurement?

Response: We appreciate this comment. We found that the BCL2 protein level itself showed a correlation with the *ex vivo* ABT-199 efficacy though the correlation value was much lower than those determined for the PPI assays. We thus reasoned that the failures of previous approaches, even those based on protein levels, were due to lack of resolution in their respective methods.

For example, when we compared the single-molecule pull-down and co-IP (SMPC) platform with Western blotting, we found the SMPC platform maintained sensitivity and data quantitiveness over a target protein concentration range that was 50 times wider than the Western blotting (Rebuttal Fig. 8a). Specifically, we expressed eGFP-labeled BCL2 proteins in HEK293T cells and attempted to measure the amounts of these BCL2 proteins using the two methods. While the Western blotting reached a limit of detection around 1 ng of the target protein, the SMPC platform maintained a signal-to-noise ratio higher than 10 even at 10^{-2} ng of target protein. We then used the Western blotting to determine the BCL2 protein levels for the same primary AML sample cohort we used to develop the predictive model in Figure 4 (Rebuttal Fig. 8b). While there was a general correlation between the Western blotting and the SMPC data sets (Rebuttal Fig. 8c), the Western blotting indicated a lower Pearson correlation value (0.59 vs. 0.76) and a larger mean-squared error (0.036 vs. 0.023) than the SMPC platform when contrasted with the ABT-199 *ex vivo* efficacy (Rebuttal Fig. 8d,e).

Rebuttal Fig. 8 | Comparison of the Western blotting and SMPC for drug efficacy prediction. a, Comparison of signal to noise ratios for BCL2-eGFP detection between Western blotting and SMPC. **b,** BCL2 protein levels in primary AML samples determined by Western blotting. **c,** Correlations between BCL2 protein levels determined by Western blotting and SMPC for primary AML samples ($n=32$). **d,e,** Prediction models for ABT-199 *ex vivo* AUC with BCL2 protein levels determined by **(d)** Western blotting, **(e)** SMPC.

Next, we sought to use flow cytometry to measure the BCL2 protein level and compare the resulting data with the SMPC platform. Due to limited sample availability, we used 14 patient samples (out of total 32 samples) as a subset of the cohort. To reduce fluctuations across measurements, we used PE-conjugated MESF beads (QuantumTM R-PE MESF, 827; Bangs Laboratories, Inc.) as a reference following the previous protocol¹⁶. The flow cytometry data also exhibited overall correspondence with the SMPC data set (Rebuttal Fig. 9a). However, when compared with the *ex vivo* ABT-199 efficacy, the flow cytometry data reflected a lower correlation value and a larger number of outliers, as the Western blotting data did, compared with the SMPC data set (Rebuttal Fig. 9b,c). Moreover, the fitting with the flow cytometry data indicated a p -value larger than 0.05, failing to show statistical significance. In summary, the SMPC platform provides higher resolution and data quantitiveness than the conventional Western blotting and flow cytometry, leading to higher, robust predictive power.

Rebuttal Fig. 9 | Comparison of the flow cytometry and SMPC for drug efficacy prediction. a, Correlations between BCL2 protein levels determined by flow cytometry and SMPC for primary AML samples ($n=14$). **b,c,** Prediction models for ABT-199 *ex vivo* AUC with BCL2 protein levels determined by **(b)** SMPC, **(c)** flow cytometry.

More specific points by line in paper:

1 - Title – "In AML" would be appropriate since this is only about AML.

: We changed the title as “Predicting the efficacy of BH3 mimetics through the profiling of multiple protein complexes in acute myeloid leukemia”.

36 - mTOR = mammalian target of rapamycin

: Edited as suggested.

40 –Does not require interaction of BAX with BAK.

: Edited as suggested.

82 – BH3 profiling does not require the derivation or maintenance of Cancer-derived mitochondria.

: Edited as suggested.

402 – Not that much of a puzzle, really. Kinetics of in vivo and in vitro response already suggest that dissociation of pre-existing complexes as the answer.

: Edited as suggested. We would like to highlight that our data point to the BCL2-BAX complex, rather than BCL2-BIM complex, as a main driver mediating the ABT-199 effectiveness.

Figure 6g is really not intelligible. What are the triangles showing? What are the lines at right showing? Why are high and low lines of equal length.

: The triangle-shaped plots were designed to help the direct comparison of PBA signals among three anti-apoptotic proteins, with an aim of providing a visualized insight into the distribution and balance of the anti-apoptotic proteins in the sample. We have identified the correlations between triangle plots and drug responses in four different types of hematological malignance cell lines (Extended Data Fig. 7).

Extended Data Fig. 7

The lines in the Figure 6 were intended to indicate the counts of the three assays included in our analysis model. In the revised manuscript, we removed these unintelligible lines in Fig. 6c and 6g. We also reorganized the presentation of the clinical data in Fig. 6 and Extended data Fig. 10.

Reviewer #5

Comment 1) Fig. 1. The definition of a positive signal in the TIRF images is not clear. The two tiny images shown in panel f have very different background levels. The circled areas of images in Extended Data Fig. 3 does not make it clear how spots were selected. They report an average spot has 439 counts while the background pixels have values between 135 and 525 (I calculate an average of 352). Depending on the level of fluctuation it seems to me that the cut-off for identifying a spot is critical. Were spots identified automatically or manually? What is the actual definition of a spot? There is no information about local background correction, or other image processing. The entire project depends on appropriate identification of spots therefore this information is critical.

Response: We thank the reviewer for this comment regarding our image processing. To clarify, each image was processed after capturing an average of fluorescence signals over a time resolution of 100 ms, using total internal reflection (TIR) for excitation. The reviewer astutely noted the fluctuations in background fluorescence, which can be attributed to both autofluorescence from the imaging chip and freely diffusing, fluorescently labeled prey proteins. To address this, we captured 8 to 10 control images from empty reaction chambers to establish a limit of background signals, identified from the distribution of signals recorded in each pixel of the control images. Our analysis software then employed a cutoff identified from the limit of background value (1,015 A.U. for 488 nm wavelength excitation) to eliminate 99% of these background signals (Extended Data Fig. 2a).

Extended Data Fig. 2a,b

Upon background subtraction, three sequential images were averaged, corresponding to a 300 ms time frame. This enabled us to identify diffraction-limited fluorescence spots and accurately count surface-immobilized, single-fluorescent molecules (Extended Data Fig. 2b). We verified the reliability of this approach for all the assays employed in this study, observing that the coefficients of variation were typically less than 10% and never exceeding 15% in any instance (Fig. 1f,h, and Extended Data Fig. 3). This high level of reproducibility was facilitated by the low autofluorescence of our imaging chip, which enhanced the distinction between background and genuine fluorescence signals, and by the almost complete suppression of non-specific adsorption of extraneous proteins onto the surface.

To enhance clarity, we have revised the Methods section, as well as Fig. 1f and Extended Data Fig. 2, to include more comprehensive details on our image processing approach with references^{12,13}. Additionally, all relevant analysis codes have been uploaded to GitHub for public access.

Fig. 1f

Comment 2) The authors examine Bim binding to BAX and BAK but there is nothing in the paper about Bid which is likely to be as important or more important than Bim given that Bim preferentially binds BAX while Bid binds both BAX and BAK similarly. This needs to be addressed either experimentally or as a caveat in the discussion.

Response: We are grateful for the reviewer's insightful comment. We made extensive efforts to evaluate commercially available antibodies against BID and identified the 3E8-1B10 clone

as having the highest affinity appropriate for our single-molecule co-IP platform (Rebuttal Fig. 10a). Unfortunately, even this optimized antibody showed only background levels of surface pull-down when it came to truncated BID (tBID) (Rebuttal Fig. 10b).

Rebuttal Fig. 10 | Selection of monoclonal antibodies for surface IP of BID protein. **a**, Surface IP of eGFP-labeled BID and tBID proteins from transfected HEK293T crude cell extracts. **b**, The surface IP levels of eGFP-labeled BID and tBID proteins were assessed using the anti-BID monoclonal antibody (3E8-1B10) in relation to varying BID concentrations.

We next opted to monitor global BID levels within HL60 cells using the same 3E8-1B10 clone antibody. As anticipated, we noticed a reduction in the quantity of full-length BID when HL60 cells were treated with ABT-199 or AZD-5591, which suggests increased apoptotic activity (Rebuttal Fig. 11a). However, we were unable to detect any complexes within the BCL2 family that included tBID (Rebuttal Fig. 11b). This outcome aligns with our earlier observation regarding the antibody's limited capability to capture the truncated form.

Rebuttal Fig. 11 | Development of immunoassays to detect BID protein complex. **a**, Changes in the total levels of full-length BID within HL60 cells after treatment with ABT-199 (0.5 μM) or AZD-5591 (0.5 μM). **b**, Detection of BID complexes with BCL2 family proteins in HL60 cells using the anti-BID monoclonal antibody (3E8-1B10).

We fully acknowledge the significance of BID in BCL2 biology and apoptosis regulation. We will address this current shortcoming by developing improved assays in future research. This limitation and its potential impact have been discussed openly in the revised version of our manuscript.

Line 453, Main text: “It is important to note that our current suite of assays lacks the capability to detect certain anti- and pro-apoptotic proteins in the BCL2 family, including BFL1 and BID. By extending our assay to incorporate these proteins – something achievable through the screening of appropriate antibodies for IP and detection – we anticipate a more comprehensive profiling of the BCL2-family PPI network, as well as enhanced accuracy in our analysis model.”

Comment 3) I looked up the antibodies to BAX and BAK. In only one place is the BAX antibody referred to as 6A7 – the caption to figure 1. On the Santa Cruz catalogue the BAK antibody sc-517390 is not validated for recognizing active BAK - the only data shown is for western blotting.

Response: The activation of BAK proteins involves specific conformational changes that enable pore formation in the mitochondrial outer membrane through oligomerization with other BAX and BAK molecules. A key feature of this conformational shift is the exposure of the BH4 domain (α 1 helix), which remains concealed within the core structure in its inactive state⁵. Antibodies targeting this exposed BH4 domain will offer a selective method to identify activated BAK proteins. This concept is also supported by a similar approach to detecting activated BAX proteins using the monoclonal antibody 6A7 clone, which targets the N-terminus region of BAX⁶.

To probe active BAK, we employed the monoclonal antibody AT38E2 (catalog number: sc-517390), developed against a recombinant protein that spans amino acids 29-187 of BAK. To confirm the selectivity of AT38E2 for the α 1 helix, we dissected BAK into four distinct fragments, each labeled with eGFP at the C-terminus, and assessed their affinity to the antibody using single-molecule pull-down. Our data unambiguously show that AT38E2

has a specific affinity for the $\alpha 1$ helix (Extended Data Fig. 1h).

Extended Data Fig. 1h,i

Subsequently, we evaluated the antibody's efficacy in detecting endogenous, activated BAK. In vitro studies have shown that heat or Triton X-100 treatment can activate BAK proteins^{7,8}. Subjecting HL60 and Ramos cell extracts to these treatments resulted in a marked increase in activated BAK levels, as confirmed by single-molecule pull-down and co-IP assays (Extended Data Fig. 1i). This indicates the antibody's ability to discriminate between activated and total BAK levels, which has been clarified in the revised Method section of our manuscript included as Extended Data Fig. 1h,i.

Comment continued) In Fig. 2 they report identical activation of BAX and BAK something we have never seen in cells. To use this antibody to quantify active BAK requires additional experimental controls.

Response: It's worth noting that the simultaneous activation of BAX and BAK, as shown in Fig. 2, appear to be specific to HL60 cells, where both BAX and BAK activation increased comparably in response to ABT-199 or STS treatment in these cells. In contrast, OCI-LY3 cells, a type of Diffuse Large B-Cell Lymphoma (DLBCL), exhibited a significant rise in active BAX levels under the same treatment conditions, but not in active BAK levels (Rebuttal Fig. 12). These results suggest that the degree of BAX and BAK activation may differ depending on cell types and contexts, and even among primary patient samples.

Rebuttal Fig. 12 | Changes in the active BAX and BAK levels after ABT-199 treatment. a,b, Changes in the levels of active BAX/BAK after the treatment with ABT-199 (0.5 μ M, 6 hours) in both (a) HL60 cells and (b) OCI-LY3 cells.

Comment 4) As I understand it, the model indicates that the BCL-2-BIM PBA indicates binding to available (empty) Bcl-2 proteins. But comparing panels b, c and d the total BCL2 = 2500. About 1000 of Bcl2 are occupied by BIM and 1500 are bound to BAX. There are no 'extra' Bcl-2 by 2 hours STS treatment. What about BAK? In panel J they find almost no BAK in HL60, NB4 or U937 cells. Comparing panel b and g why is the total BCL-2 around 3000 in b and >12,000 in panel g? And explanation is required.

Response: The primary aim of our STS experiment was to investigate changes in BCL2 proteins and their associated complexes in HL60 cells during the progression of apoptosis. We noted an uptick in both the total BCL2 levels and the BCL2-BAX complex, suggesting a reliance on BCL2 proteins in HL60 cells to counteract apoptotic stresses. We acknowledge that the single-molecule data counts in the original Fig. 2 were relative rather than directly comparable due to varying experimental conditions.

Fig. 2a-d

To address this, we rescaled the single-molecule counts in Fig. 2a-d based on labeling efficiencies detailed in Extended Data Fig. 5 as well as the specific incubation conditions. The rescaled counts showed that pre-STS treatment levels of total BCL2 were consistently around 12,000, whereas the counts for BCL2-BIM and BCL2-BAX complexes were approximately 1,000 and 700, respectively. Importantly, their ratios corroborate the data in Fig. 2j that approximately 12.3% of BCL2 in naive HL60 cells is occupied by complexes, suggesting that increased BCL2-BAX levels may not surpass the total available BCL2 during STS treatment. We also observed that counts for the BCL2-BAK and BCL2-BAD complexes remained at background levels throughout the STS treatment (Rebuttal Fig. 13).

Rebuttal Fig. 13 | Changes of BCL2 family protein complexes of HL60 cells through the apoptosis progression by 2 μ M of STS. (a) BAK complexes with anti-apoptotic proteins, (b) BAD complexes with anti-apoptotic proteins.

During manuscript revision, we further converted PBA counts into populations of unoccupied BCL2 proteins. Interestingly, for HL60 cells, about 50% of the total BCL2 proteins were found to be unoccupied when probed with the BIM_{BH3}, even though complexes like BCL2-BIM and BCL2-BAX occupied about 12.3% of total BCL2 (Fig. 2j). This implies that about one third of BCL2 proteins were engaged with unidentified interacting partners, which were not part of our current assay set. For NB4 and U937 cells, these proportions were lower, at 24% and 21%, respectively (Fig. 2j). Including additional interactors like BID and truncated BID in our profiling could provide a more comprehensive understanding of BCL2 complex compositions.

To enhance clarity, we've included rescaling details in the figure legend and expanded the Method section to outline each assay protocol. We hope these revisions adequately address the reviewer's queries and offer a more thorough understanding of our study.

Comment 5) Lines 202-205. The data shown do not report on relative affinities of PPIs. There are no binding curves in the manuscript for BCL-2 binding Bad compared to Bim. The competition experiment in extended Fig 7b is more determined by off-rate than affinity. Affinity is a well-defined quantitative value. Unless, the authors can justify the use of the term it should be replaced.

Response: We're grateful to the reviewer for their insightful comment. We would like to highlight that our experimental setup allows us to conduct binding and competition experiments under conditions that closely approximate equilibrium. We determined the dissociation constants (K_d) for all possible combinations between anti-apoptotic proteins and PPI probes developed in this study. Please see the section titled "In-depth characterization of PPI binding affinities." in our response to all reviewers.

Comment 6) Lines 287-298. It is not clear what the authors are using as a metric. They state only that combining the top three correlations resulted in a limited improvement indicating that they are redundant. According to Fig 4 the correlation with BCL-2 level is essentially equivalent and it is likely all 4 are highly correlated with each other. Has this been tested?

It is not clear why the authors decided instead to add the negative correlation with BCL-XL-BAK complexes instead. Similarly in Fig. 5 the PPI metric selected is MCL1 total level, BCL-XL-BimBH3 PBA and BCL-XLBAK CPX yet the active BAK level and Bcl-XL-BAD CPX had higher Pearson's R. How was the selection made, why and what effect does it have on the prediction. These issues need to be addressed in the text and additional data justifying the selections provided.

Response: We appreciate this important comment by the reviewer. We employed linear regression to assess the performance of each model in terms of the resulting Pearson correlation value and the coefficients of PPI metrics along with their *p*-values. For more detailed accounts of developing drug efficacy prediction models, please see the section titled "Detailed explanation of the development for drug efficacy prediction models with combinatorial metrics" in our response to all reviewers.

Furthermore, we evaluated the performance of the drug efficacy prediction models through the established statistical methods, MSE and ROC curve analysis. For more detailed accounts of these analysis, please refer to the section titled "Evidence for the predictive power of the combined metrics model" in our response to all reviewers.

Comment 6 continued) If all that is required is measuring BCL-2 levels by flow cytometry that would be much simpler than the method proposed here.

Response: We appreciate this critical comment that would be important when considering practical utility of our method compared to other methods. Following the reviewer comment, we sought to use flow cytometry to measure the BCL2 protein level and compare the resulting data with the single-molecule co-IP platform. Due to the limit in sample availability, we used 14 patient samples (out of total 32 samples) as a subset of the original cohort. To reduce fluctuations across measurements, we used PE-conjugated MESF beads (Quantum™ R-PE MESF, 827; Bangs Laboratories, Inc.) as a reference following the previous report¹⁶. The flow cytometry data also exhibited overall correspondence with the single-molecule co-IP data set (Rebuttal Fig. 14a). However, when compared with the *ex vivo* ABT-199 efficacy, the flow cytometry data reflected a lower correlation value and a larger number of outliers

compared with the single-molecule co-IP data set (Rebuttal Fig. 14b,c). Moreover, the fitting with the flow cytometry data indicated a p -value larger than 0.05, failing to show statistical significance (Rebuttal Fig. 14c). In summary, the single-molecule co-IP platform provides higher resolution and data quantitiveness than the conventional flow cytometry, leading to higher, robust predictive power.

Rebuttal Fig. 14 | Comparison of the predictive powers for ABT-199 drug responses across flow cytometry and single-molecule co-IP. **a**, Correlations between BCL2 protein levels determined by single-molecule co-IP and flow cytometry for primary AML samples ($n=14$). **b,c**, Prediction models for ABT-199 *ex vivo* AUC with BCL2 protein levels determined by **(b)** single-molecule co-IP, **(c)** flow cytometry.

Comment 7) It is not clear to me that the “six cases, including BC-6524 and BC-7230” showed an outstanding performance in predicting the clinical responses. According to Table 3 in the supplementary, of the 6 two of three non-responders were correctly identified and then only by including partial response, transient response and complete response as all responders – with no correlation to *ex vivo* AUC and the extent of response are the others correctly reported. In addition, the methods section on Drug efficacy prediction model analysis is not clear making interpretation even more difficult. The claims made by the authors need to be more reflective of the true potential for clinical utility.

Response: We appreciate the reviewer's concerns about the clarity of the manuscript's previous version, particularly regarding how BCL2 PPI profiling was used to make predictions and correlate with clinical outcomes. To address this, we would like to highlight that we conducted BCL2 PPI profiling on primary AML samples for a total of ten patients,

including three new patient cases added during manuscript revision. For more detailed accounts of these analysis, please see the section titled “Enhanced validation of clinical prediction outcomes” in our response to all reviewers.

More specific points by line in paper:

1. Inconsistent nomenclature and incomplete information make the manuscript very hard to read.

: With aids from a professional editing service, we have extensively edited the manuscript to improve its readability.

2. In the manuscript CPX is not defined and LV is defined only in a figure.

: We defined these abbreviations when they first appeared in the revised manuscript.

3. More explanation is required for Extended Data Fig. 6. The description is not clear.

: We have revised the figure legend.

4. The statement: “Presumably, under the STS treatment condition used, the rate at which active BAXs were produced excelled the rate that BCL2 sequestered BAX proteins, resulting in HL60 cells finally succumbing to apoptotic stress and initiating MOMP (Fig. 2a,d).” is confusing. I assume they mean that more BAX proteins become activated than can be bound by BCL2 (really all anti-apoptotic proteins). The rate is immaterial here as there is a large excess of BAX compared to anti-apoptotic proteins.

: We have revised the sentence accordingly which reads as follows: “Presumably, under the applied STS treatment conditions, the active BAX outnumbered the BCL2's sequestration capacity, leading to the HL60 cells succumbing to apoptotic stress and initiating MOMP (Fig. 2a,d).”

5. The statement on lines 192-193 misrepresents the data. BCL2 proteins did not continuously form additional BCL2-BAX complexes. According to the data in Fig. 2D complexes increased for 2 hours and then leveled off with no further increases. The text

needs to be revised or the statement supported by specific data.

: We have revised the sentence following reviewer's comment: "While the counts of the BCL2-BIM complex gradually decreased over time, the BCL2-BAX complex counts surged for the first 2 hours following STS treatment (Fig. 2c,d)."

6. The description of the PPI probes is inadequate. "HEK293T cells expressing PPI probes (BIMBH3-eGFP, BIMEL-eGFP, BAD-eGFP, and NOXA-eGFP) and SF9 cells expressing BIMBH3-eGFP probe were lysed with Triton-X100 lysis buffer for PPI probe binding assay (PBA)." Does not even make clear whether the eGFP is fused to the amino- or carboxyl- end of the proteins, or if the fusion proteins retain authentic pro-apoptotic function or if the partner proteins (BIMEL, BAD, NOXA) are full length. What is the sequence of the BIM BH3 region being used? What is "proper" lysis buffer (Line 530)?

: The eGFP tag was fused to each C-terminal end of the full-length pro-apoptotic proteins. We confirmed that pro-apoptotic function of these fusion proteins by observing active instigation of apoptosis when overexpressed in HEK293T cells. The BIM sequence and the composition of Triton X-100 lysis buffer have been added to the Fluorescence-labeled protein constructions and Cell lysis sub-sections of the Method section.

7. The statement on line 211 that BCL2 accounts for a dominant portion in sequestering... is misleading. The data indicates that there are more unoccupied Bcl-2s which fits with the later panels. It doesn't mean "BCL2 accounts for a dominant portion in sequestering pro-apoptotic proteins in HL60".

: We have revised the sentence as follows: "Given the higher affinities of the BIM_{BH3} domain for BCLxL and MCL1 than for BCL2 (Extended Data Fig. 6b), this elevated PBA count suggests that BCL2 works as a larger reservoir in sequestering pro-apoptotic proteins compared to BCLxL or MCL1 in HL60 (Fig. 2f)."

8. Figure 3F suggests that ABT-199 does not displace BIM from BCL-2 which fits with

previous reports of BIM binding Bcl-2 and BCL-XL not being displaced by BH3 mimetic drugs. This result is not very surprising. The text should be revised.

: We have revised the sentence as follows: “The two major BCL2 complexes showed opposing results. The BCL2-BAX complex began to unravel instantly after ABT-199 administration, while the number of BCL2-BIM complexes largely persisted (Fig. 3f).”

9. How come there are more active BAX and BAK in Fig. 3G (4000) than total BAX and BAK in Fig. 3H (3000)?

: We appreciate the reviewer’s meticulous attention to these details. As we also detailed in our response to your Comment 4, we used different lysate concentrations and incubation times for different sets of experiments, accounting for why we obtained incompatible count numbers.

Fig. 3g,h

During the revision of the manuscript, we repeated the experiments shown in Fig. 3g using the same experimental conditions as those in Fig. 3h. This enabled a direct comparison of the resulting single-molecule counts between the two figures (Fig. 3g,h). The updated data reveal that the population of activated BAX steadily increased, whereas the overall BAX population remained relatively stable throughout the course of ABT-199 treatment. After 4 hours of treatment, the active BAX population constituted approximately two-thirds of the total BAX population.

10. The cartoon in Fig. 3o does not seem to match the data. The data in panel n and the text indicate that there are many more Bcl-XL-BAK complexes formed when cells are treated with AZD-5991 than are released from MCL-1 but the cartoon shows an equal number (1). It isn't clear what the authors think happens to the released BAK.

: As noted by the reviewer, we have updated Fig. 2o to differentiate between the released and nascent BAK populations, and we have also adjusted the corresponding counts of depicted BAKs for a more accurate representation in the model.

References

- 1 Matulis, S. M. *et al.* Functional profiling of venetoclax sensitivity can predict clinical response in multiple myeloma. *Leukemia* **33**, 1291-1296, doi:10.1038/s41375-018-0374-8 (2019).
- 2 Gupta, V. A. *et al.* Venetoclax ex vivo functional profiling predicts improved progression-free survival. *Blood Cancer J* **12**, 115, doi:10.1038/s41408-022-00710-9 (2022).
- 3 Dohner, H. *et al.* Diagnosis and management of AML in adults: 2017 ELN recommendations from an international expert panel. *Blood* **129**, 424-447, doi:10.1182/blood-2016-08-733196 (2017).
- 4 Bhatt, S. *et al.* Reduced Mitochondrial Apoptotic Priming Drives Resistance to BH3 Mimetics in Acute Myeloid Leukemia. *Cancer Cell* **38**, 872-890 e876, doi:10.1016/j.ccell.2020.10.010 (2020).
- 5 Li, M. X. *et al.* BAK alpha6 permits activation by BH3-only proteins and homooligomerization via the canonical hydrophobic groove. *Proc Natl Acad Sci U S A* **114**, 7629-7634, doi:10.1073/pnas.1702453114 (2017).
- 6 Peyerl, F. W. *et al.* Elucidation of some Bax conformational changes through crystallization of an antibody-peptide complex. *Cell Death Differ* **14**, 447-452, doi:10.1038/sj.cdd.4402025 (2007).
- 7 Pagliari, L. J. *et al.* The multidomain proapoptotic molecules Bax and Bak are directly activated by heat. *P Natl Acad Sci USA* **102**, 17975-17980, doi:10.1073/pnas.0506712102 (2005).
- 8 Dewson, G. *et al.* Bak activation for apoptosis involves oligomerization of dimers via their alpha6 helices. *Mol Cell* **36**, 696-703, doi:10.1016/j.molcel.2009.11.008 (2009).
- 9 Del Gaizo Moore, V. & Letai, A. BH3 profiling--measuring integrated function of the mitochondrial apoptotic pathway to predict cell fate decisions. *Cancer Lett* **332**, 202-205, doi:10.1016/j.canlet.2011.12.021 (2013).
- 10 Fraser, C., Ryan, J. & Sarosiek, K. BH3 Profiling: A Functional Assay to Measure Apoptotic Priming and Dependencies. *Methods Mol Biol* **1877**, 61-76, doi:10.1007/978-1-4939-8861-7_4 (2019).
- 11 Seyfried, F. *et al.* Prediction of venetoclax activity in precursor B-ALL by functional

- assessment of apoptosis signaling. *Cell Death Dis* **10**, 571, doi:10.1038/s41419-019-1801-0 (2019).
- 12 Lee, H. W. *et al.* Real-time single-molecule coimmunoprecipitation of weak protein-protein interactions. *Nat Protoc* **8**, 2045-2060, doi:10.1038/nprot.2013.116 (2013).
 - 13 Yoo, J., Lee, T. S., Choi, B., Shon, M. J. & Yoon, T. Y. Observing Extremely Weak Protein-Protein Interactions with Conventional Single-Molecule Fluorescence Microscopy. *J Am Chem Soc* **138**, 14238-14241, doi:10.1021/jacs.6b09542 (2016).
 - 14 Liu, Q. *et al.* Bim escapes displacement by BH3-mimetic anti-cancer drugs by double-bolt locking both Bcl-XL and Bcl-2. *Elife* **8**, doi:10.7554/eLife.37689 (2019).
 - 15 Bogner, C. *et al.* Allosteric Regulation of BH3 Proteins in Bcl-x(L) Complexes Enables Switch-like Activation of Bax. *Mol Cell* **77**, 901-912 e909, doi:10.1016/j.molcel.2019.12.025 (2020).
 - 16 Smith, M. L. & Tahir, S. K. Quantification of BCL-2 Family Members by Flow Cytometry. *Methods Mol Biol* **1877**, 163-172, doi:10.1007/978-1-4939-8861-7_11 (2019).

Rebuttal 2

Reviewer #4

The responses are on the whole adequate. This is a very interesting and useful study.

Comment 1) One of the other reviewers provoked a comparison of these results with BH3 profiling. I hope these results will be available to the reader. As will many of the responses, it was never quite clear what was being provided to the reviewer alone as opposed going into the revised manuscript. The authors should recognize that the reviewer comments (not only this one, but all of them!) generally brought up points that should be explicated not only to the reviewers, but also to readers who are likely going to require the same explanations.

Extended Data Fig. 9

Response: We sincerely thank the reviewer for the positive reception of our revised manuscript. Regarding the reviewer request for comparisons between our SMPC profiling

and other established methods such as BH3 profiling and flow cytometry, we have already incorporated these comparative analyses as Extended Data Figure 9 in the revised manuscript. This addition not only highlights the consistency and correlation of drug efficacy predictions between our SMPC platform and these established methodologies (Extended Data Fig. 9h, j), but it also demonstrates the superior correlations of our platform with *ex vivo* ABT-199 efficacy compared to both BH3 profiling and flow cytometry (Extended Data Fig. 9l). Furthermore, in the second revised version of the manuscript, we have included an in-depth comparative analysis between western blotting and SMPC in Extended Data Figure 9 (panel j to l). This comparative study serves to underscore the reliability and efficacy of our SMPC platform in characterizing protein-protein interactions.

In response to the reviewer's suggestion that newly generated results should be made readily available to readers, we have taken the initiative to incorporate additional datasets obtained during the final revision of the manuscript. These supplementary datasets have been integrated into the Extended Data Figures, ensuring their accessibility to all potential readers in the final version of the manuscript. Among these new additions, Extended Data Figure 8a now presents data illustrating that BCL2 family protein and complex levels are minimal in peripheral blood mononuclear cells (PBMCs) from healthy donors. This dataset serves as a crucial negative control, further strengthening the robustness of our PPI profiling. In addition, we have included an outlier analysis, comparing the single-metric and combinatory metrics models, as depicted in Extended Data Figure 8f. This analysis provides valuable insights into the performance and effectiveness of our metrics models.

Extended Data Fig. 8a

Extended Data Fig. 8f

The extensive data that cannot not be accommodated within the manuscript, such as statistical coefficients from the numerous prediction models, have been uploaded to GitHub (<https://github.com/tyyoonlab-snu/Nat-Biomed-Eng-2023->).

Comment 2) An additional point is that prior work with BH3 profiling found that sensitivity to probes of BCL-XL and MCL-1 were better negative predictors of in vivo response to venetoclax in AML than response to BAD, which is sensitive to BCL-2 as well as BCL-XL and BCL-w. This is congruent with the BCL-XL complexes found here to be a negative predictor. The authors may want to comment on this.

Response: We thank the reviewer for this thoughtful comment based on deep understanding of the field. In the previous studies on BH3 profiling, the highest predictive power was demonstrated when the priming degree by the BAD peptide was subtracted from that by the HRK peptide^{1,2}. This approach is logically sound as BAD primes both the anti-apoptotic proteins BCL2 and BCLxL. By subtracting the priming effect induced by HRK, which specifically targets BCLxL, the anti-apoptotic contribution of BCL2 can be distinguished from that of BCLxL. In addition, it is shown that the levels of MCL1 and its complexes increase in AML-patient derived xenograft models resistant to ABT-199¹.

These observations align well with our SMPC data and the resulting prediction model in Figure 4f, where the BCLxL-involving complexes (epitomized by BCLxL-BAK complex) operate as a negative factor for the ABT199 efficacy in the given cancers. Importantly, the same BCLxL-BAK complex also emerges as a key negative factor in the efficacy prediction model for MCL1 inhibitors, as demonstrated in Figure 5f. These findings strongly suggest that BCLxL steps in to compensate for the anti-apoptotic pathway when BCL2 or MCL1 becomes saturated by BH3 mimetics. In the Discussion section, we consolidate these shared observations between our SMPC and BH3 profiling methods as follows:

Line 453, Main text: These resistance complexes, displaying negative correlations, have also been consistently noted in the BH3 profiling method, where the priming effect triggered by the HRK peptide, a specific binder to BCLxL, acts as a detractor for ABT-199 efficacy. Similarly, the BCLxL-BAK complex has been identified as a critical negative factor in our efficacy prediction model for MCL1 inhibitors. These observations robustly imply that BCLxL compensates within the anti-apoptotic pathway when either BCL2 or MCL1 is overwhelmed by BH3 mimetics.

Comment 3) Figure 4b - I am not sure the authors understood my point. My point is that some of the correlations are negative, so why are all the R values positive? This could be fixed by simply labeling the y-axis “absolute value Pearson R”.

Response: In response to the comment, we have revised the label of the y-axis in Figure 4b (as well as in Figure 5a) as “absolute value Pearson R.”

Fig. 4b

Fig. 5a

Reviewer #5

The authors are to be commended for the revisions made to the manuscript. Overall the text and figures are much clearer and the conclusions (for the most part) better supported by the data.

Response: We thank the reviewer for positive reception of our revised manuscript.

Comment 1) Extended data Fig. 6 reports the data used to calculate K_d values for the various interactions. To calculate a K_d the data must fit to a Hill equation - even to generate an "apparent K_d " the data must saturate. Straight lines indicate collisions not binding. In several places K_d values are reported for straight lines and for curves that do not saturate. It is not clear what value is being used by the software for saturation (the default in many packages is to use the highest value whether or not it is sensible). If the actual saturation point is known and the curve comes close to that value an estimate of an apparent K_d is reasonable but for the data presented it is not clear that these conditions apply. Furthermore the data in support for the occupancy calculations (extended Data Figure 6h) indicates occupancy (I assume it is fractional occupancy) and K_d for something called Probe at different concentrations. It is not clear what the two binding partners are and for the reasons outlined above it is not clear that the K_d values are meaningful.

Response: We are grateful for the reviewer's insightful observations and would like to clarify some distinctions between our SMPC method and conventional techniques such as Surface Plasmon Resonance (SPR) and Bio-Layer Interferometry (BLI). Indeed, kinetic methods like SPR and BLI focus on observing the saturation of kinetic data to accurately determine rate constants^{3,4}. These techniques capture the initial increase in optical signals as binding events outnumber unbinding events, reflecting primarily on binding reaction rate constants (Rebuttal Fig. 1). Conversely, during buffer exchange when unbinding events predominate, the signals predominantly report on unbinding rate constants.

Rebuttal Fig. 1 | Typical set-up and binding cycle with an SPR biosensor³

Rebuttal Fig. 2 | Schematic for the SMPC-based probe binding assay (PBA) to measure fractional occupancy

In contrast, our SMPC platform is designed to measure fractional occupancy at equilibrium for each given prey concentration (Rebuttal Fig. 2). For example, in assessing the BCL2-BIM_{BH3} interaction, we immobilize mCherry-labeled BCL2 proteins on a surface and quantify these surface baits in a set field of view (100×100 μm² in our case) using mCherry fluorescence. Upon introducing eGFP-labeled BIM_{BH3} proteins, we allow sufficient time (typically over 10 minutes) to reach equilibrium where binding and unbinding reactions balance. The number of bound probe proteins is then determined by counting individual eGFP fluorescence spots, enabling us to calculate fractional occupancy (Equation 1).

$$\text{Fractional occupancy} = \frac{\text{Number of bound probes (eGFP counts)}}{\text{Number of baits (mCherry counts)}} \quad (1)$$

$$\text{Fractional occupancy} = \frac{[\text{PPI Probe}]}{K_d + [\text{PPI Probe}]} \quad (2)$$

By measuring occupancy values at various probe concentrations and fitting these data to Equation 2, we can ascertain K_d values. Thus, our approach captures occupancy values at equilibrium, not just collisional interactions. Unlike the kinetic methods, we do not require saturation of the occupancy curve for reliable K_d determination, as evidenced by the high R-squared values in our curve-fitting results (Extended Data Fig. 6a-d).

Extended Data Fig. 6a-d

Regarding binding cooperativity, we initially assumed a Hill coefficient of one, consistent with the established one-to-one complex formation between BH3-only and anti-apoptotic proteins^{5,6}. The reviewer provided an insightful comment, inquiring whether we had examined the binding cooperativity by employing a generalized Hill equation (Equation 3), which is presented as follows:

$$\text{Fractional occupancy} = \frac{[PPI\ Probe]^n}{K_d + [PPI\ Probe]^n} \quad (3),$$

where n is the Hill coefficient.

Upon applying the generalized Hill equation to our data, we found Hill coefficients approximating one within the error range, corroborating the K_d values obtained under the assumption of $n=1$ (with one exception of the BCL2-BIM_{EL} reaction, presumably because of its relatively large K_d value) (Extended Data Fig. 6e). This validates our SMPC-based PBA in accurately reflecting the biological characteristics of PPIs within the BCL2 family. While traditional methods excel in extracting binding and unbinding rate constants, our SMPC platform offers a complementary and efficient approach for determining equilibrium-based interaction strengths.

e

PBA	n	K_d (nM)	K_d (nM) (fixed $n=1$)
BCL2-BIM _{BH3}	0.92 ± 0.07	42.3 ± 5.9	49.8 ± 2.2
BCLxL-BIM _{BH3}	1.02 ± 0.09	11.4 ± 2.1	11.0 ± 0.5
MCL1-BIM _{BH3}	0.97 ± 0.08	4.6 ± 0.7	4.6 ± 0.2
BCL2-BIM _{EL}	1.29 ± 0.08	260 ± 47	134 ± 9.4
BCLxL-BIM _{EL}	1.02 ± 0.05	74.0 ± 8.1	71.1 ± 1.9
MCL1-BIM _{EL}	0.99 ± 0.07	57.0 ± 8.0	57.8 ± 2.0
BCL2-BAD	0.96 ± 0.12	5.1 ± 1.0	5.5 ± 0.4
BCLxL-BAD	1.15 ± 0.14	3.2 ± 0.8	2.5 ± 0.2
MCL1-BAD	Not Determined		Not Determined
BCL2-NOXA	Not Determined		Not Determined
BCLxL-NOXA	Not Determined		Not Determined
MCL1-NOXA	0.85 ± 0.24	3.4 ± 1.1	4.1 ± 0.5

Extended Data Fig. 6e

We have revised the Methods section to give more details on how we determined the dissociation constant for the given PBA reaction.

Line 638, Methods section: To ascertain the dissociation constants (K_d), we quantified the fractional occupancy of surface baits at equilibrium. This was calculated as the ratio of bound probes to the number of surface baits within our designated field of view. We then measured these fractional occupancy values across a range of probe concentrations and fitted them to the following equation, which enabled the determination of the dissociation constant for a specific PBA reaction.

$$\text{Fractional occupancy} = \frac{[PPI\ probe]}{K_d + [PPI\ probe]}$$

Continued for comment 1) It is also not clear what the different colors are meant to indicate.

Finally, where "Not Binding" is reported (Extended Fig. 6g) no data are provided in panels a-d and the histograms in panels e-f have no explanation as to the concentrations of the molecules.

Response: In line with the aforementioned explanation, we have refined the definitions presented in Extended Data Figure 6 for enhanced clarity. We specified the binding pairs for individual reactions as well as included binding curves that we failed to ascertain K_d values (labeled "Not determined" in Extended Data Fig. 6f). We also tried to improve the readability of Extended Data Figure 6 by revising the color code and the text labeling.

f

	BIM _{BH3}	BIM _{EL}	BAD	NOXA
BCL2	Low Binding Affinity ($K_d > 20$ nM)	Low Binding Affinity ($K_d > 20$ nM)	High Binding Affinity ($K_d < 20$ nM)	Not Determined
BCLxL	High Binding Affinity ($K_d < 20$ nM)	Low Binding Affinity ($K_d > 20$ nM)	High Binding Affinity ($K_d < 20$ nM)	Not Determined
MCL1	High Binding Affinity ($K_d < 20$ nM)	Low Binding Affinity ($K_d > 20$ nM)	Not Determined	High Binding Affinity ($K_d < 20$ nM)

■ High Binding Affinity ($K_d < 20$ nM)
■ Low Binding Affinity ($K_d > 20$ nM)
□ Not Determined

Extended Data Fig. 6f

References

- 1 Bhatt, S. *et al.* Reduced Mitochondrial Apoptotic Priming Drives Resistance to BH3 Mimetics in Acute Myeloid Leukemia. *Cancer Cell* **38**, 872-890 e876, doi:10.1016/j.ccell.2020.10.010 (2020).
- 2 Seyfried, F. *et al.* Prediction of venetoclax activity in precursor B-ALL by functional assessment of apoptosis signaling. *Cell Death Dis* **10**, 571, doi:10.1038/s41419-019-1801-0 (2019).
- 3 Cooper, M. A. Optical biosensors in drug discovery. *Nat Rev Drug Discov* **1**, 515-528, doi:10.1038/nrd838 (2002).
- 4 Muller-Esparza, H., Osorio-Valeriano, M., Steube, N., Thanbichler, M. & Randau, L. Bio-Layer Interferometry Analysis of the Target Binding Activity of CRISPR-Cas Effector Complexes. *Front Mol Biosci* **7**, 98, doi:10.3389/fmolb.2020.00098 (2020).
- 5 Petros, A. M., Olejniczak, E. T. & Fesik, S. W. Structural biology of the Bcl-2 family of proteins. *Bba-Mol Cell Res* **1644**, 83-94, doi:10.1016/j.bbamcr.2003.08.012 (2004).
- 6 Kale, J., Osterlund, E. J. & Andrews, D. W. BCL-2 family proteins: changing partners in the dance towards death. *Cell Death and Differentiation* **25**, 65-80, doi:10.1038/cdd.2017.186 (2018).

Rebuttal 3

Response to Reviewer #5's comment

Comment) As far as I am concerned, the authors are not correct. I did not assume that they were using a non-equilibrium binding method like SPR.

The fit of the data to a straight line giving a low R-squared value just indicates how well the data fit the line. In an equilibrium experiment of the type they are performing, the probes can collide, particularly when one of the probes is tethered. Therefore, the data are in 2D rather than 3D.

In a concentration curve, real binding saturates. At high concentrations of the acceptor, the donor becomes saturated. Without that, in my opinion, they cannot distinguish collisions from binding.

Here is a quote from an elementary textbook on pharmacology:

"In the case of ligand-receptor complexes, K_d represents the ligand concentration and should be calculated when 50% of the receptors are bound to ligands."

If you do not know where 100% is (estimated from saturation) I know, how do you calculate K_d when 50% of the receptors are bound to ligands? And if you use their definition of fractional occupancy, the graphs they provide in the rebuttal show fractional occupancies for Bcl-2 of approximately 25%, 10%, 70% (so this one is okay), and 0 (also okay to conclude no binding). For 2 of the 4 graphs, they do not have the data needed to properly estimate a K_d .

Response: We appreciate the reviewer's concern. We wish to highlight that the determination of PPI affinity does not influence the central conclusions of our manuscript. For actual data collection, the probe binding assay (PBA) was consistently performed with the total protein concentration of the samples maintained at a constant level, such as 1 mg/ml. By averaging the time-resolved images from the PBA over 300 ms, we only counted PPI complexes that were stable for at least this duration. The PBA counts obtained were directly utilized to train the predictive models, as illustrated in Figures 4 and 5, without referencing specific PPI affinity values.

In response to the reviewer's feedback, we have removed two sentences from the main text that referred to affinity values, along with the previous Extended Data Fig. 6b-f and Supplementary Table 1. We assert that since the PBA counts, obtained under controlled conditions, were directly applied in model training, the elimination of these sentences and the associated supplementary data do not undermine the validity of our conclusions.

Modifications to the main text and the Extended Data Figures have been highlighted using the track changes feature in MS Word.